



# Assessing uncertainty in past ice and climate evolution : overview, stepping-stones, and challenges

Lev Tarasov[1] and  Michael Goldstein[2]

[1]Department of Physics and Physical Oceanography, Memorial University of Newfoundland and Labrador, St. John's, Canada, A1B 3X7
[2]Durham University, Durham, England

**Correspondence:** Lev Tarasov (lev@mun.ca)

**Abstract.**

In the geosciences, complex computational models have become a common tool for making statements about past earth system evolution. However, the relationship between model output and the actual earth system (or component thereof) is generally poorly specified and even more poorly assessed. This is especially challenging for the paleo sciences for which data constraints are sparse and have large uncertainties. Bayesian inference offers, in principle, a self-consistent and rigorous framework for assessing this relationship as well as a coherent approach to combining data constraints with computational modelling. Though "Bayesian" is becoming more common in paleoclimate and paleo ice sheet publications, our impression is that most scientists in these fields have little understanding of what this actually means nor are they able to evaluate the quality of such inference. This is especially unfortunate given the correspondence between Bayesian inference and the classical concept of the scientific method.

Herein, we examine the relationship between a complex model and a system of interest, or in equivalent words (from a statistical perspective), how uncertainties describing this relationship can be assessed and accounted for in a principled and coherent manner. By way of a simple example, we show how inference, whether Bayesian or not, can be severely broken if uncertainties are erroneously assessed. We explain and decompose Bayes Rule (more commonly known as Bayes Theorem), examine key components of Bayesian inference, offer some more robust and easier to attain stepping stones, and provide suggestions on implementation and how the community can move forward. This overview is intended for all interested in making and/or evaluating inferences about the past evolution of the Earth system (or any of its components), with a nominal focus on past ice sheet and climate evolution during the Quaternary.



## 1 Introduction

"*No one trusts a model except the [wo]man who wrote it; everyone trusts an observation except the [wo]man who made it*", Harlow Shapely.

For those studying the past evolution of the earth/climate system, a primary goal is to improve our understanding of the past on both phenomenological and process levels. However, beyond inferences closely tied to well-constrained proxies and/or to
core physics of the ice/climate/earth system, "what do we really know?" and especially "how certain are we?" are a challenge to answer.

Given the relatively sparse (in both time and space) data from the past along with uncertainties in their interpretation and dating, large-scale inferences nominally based solely on paleo data often arguably rely heavily on storytelling with varying combinations of proxy models relating a datum to a physical characteristic of interest, age calibration models, and basic
physical reasoning for self-consistency. Evolving computational models of various components of the Earth and climate system permit data-rich stories that incorporate the embodied physical equations (or more accurately approximations thereof). But when used on their own, they have limited retrodictive confidence. They therefore also arguably fall within the storytelling and self-consistency rubric. It is also our own experience that modellers who know their models well are often the most skeptical about their model results, contrary to Harlow Shapely's quip quoted above.

The exponential growth of accessible computer power over the last few decades has permitted the ongoing development of a synthesis of the above two approaches: inference based on rigorously combining computational modelling with paleo observations. This potentially offers detailed pictures of ice sheet and climate system evolution that can go confidently well beyond storytelling but, as detailed below, only when all uncertainties are rigorously addressed and assessed. Such assessment is to date not prevalent within the community (though relevant communities are making calls for such assessment, *e.g.,*
Whitehouse and Tarasov, 2014; Stokes et al., 2015) and is the central theme of this overview.

Story-telling, or in more usual terminology, hypothesis creation/elaboration, is a central part of science. This is a characteristic common to most (all?) disciplines that endeavour to explain the world around us, including history and theology. What distinguishes the scientific enterprise? Though the philosophy of science offers nuanced alternatives, we posit most scientists would assign a central role to the rigorous testing of falsifiable hypotheses (and continuous testing of theories) as one dis-
tinguishing feature. However, without rigorous measures to quantify the correspondence between model-based realizations of hypotheses and the "real world", such testing becomes vague and ill-defined. Such correspondence quantification, aka uncertainty assessment, enables meaningful statements about the world around us instead of statements that only describe the behaviour of our models..

Given the above, rigorous quantification of uncertainty is, we submit, another distinguishing feature of the natural sciences.
It is also a feature that deserves more attention in relevant "culture wars" and in dealing with anti-science memes. However, too often, uncertainties are either ignored, or only partially assessed. Furthermore, as we'll elucidate below, the assumptions underlying commonly utilized statistics are generally broken for the given context. Another tendency is to provide various



provisos such as "subject to unaccounted uncertainties in the climate forcing" that somehow the average reader is supposed to be better able to assess/interpret the impact thereof than the modeller who should be most intimately familiar with their model's foibles.

As we will show below, incomplete uncertainty assessment will result in inferences that are incorrect with, for example, "reality" well outside of the conventional two sigma, uncertainty bounds. Rigorous uncertainty assessment is a challenging task for many modelling contexts, especially all dealing with complex environmental systems. It has attracted significant attention from philosophers of science (*e.g.,* Frigg et al., 2015b). It is also a task that can't be ignored as any natural scientist needs to be able to assess the extent and rigour of uncertainty assessment as they review and assimilate literature. However most in the natural sciences are not provided with the appropriate conceptual tools for such assessment during their training.

A related issue in the context of data model comparison is the tendency to focus attention on simulations that best fit observational/paleo data with little or no attention to what may be gleaned from other simulations, both within and outside of inferential bounds. By more clearly and credibly defining the uncertainty bounds of one's inferences, one can better delineate what is not known and where future efforts might offer the most scientific return. Accepting this premise would require much more attention on, for example, the max/min bounds of ice sheet evolution during the last deglaciation. Yet major international endeavours such as the paleo model intercomparison project (PMIP) have, to date, only based their intercomparisons on using "best-fitting" ice sheet chronologies as their boundary conditions, ignoring the uncertainties in these chronologies (some of which have no assessed uncertainties).

To provide some motivation for this overview, and the chosen focus on a paleo context, it is worth considering glaciological model reconstructions of past ice sheet evolution (table 1). To date, none have adequately addressed relevant uncertainties. Furthermore, the number of ensemble model parameters varies widely (from 2 to 39), as does the range and quantity of data constraints. No published studies after 2014 have used more than 5 ensemble model parameters, raising concerns about methodological progress in assessing uncertainties within the paleo ice sheet community.

Methodological progress is much more evident in recent work on modelling near future (order 100 years) evolution of ice or climate (table 2). Near-future ice sheet evolution is a much more tractable problem than past ice sheet evolution given the relevant time-scales (limiting not only the possible ice sheet evolution phase space and model parameters requiring statistical calibration but also computational cost) and the availability of state-of-the-art climate histories from climate models.

As a step towards addressing the above challenges for the paleo community, below we unpack and provide some suggestions towards operationalization of the following self-evident statement: In order to use a model to make meaningful inferences about a physical system, the relationship between the model and the actual system must be meaningfully specified.

Our aim is to improve our understanding of the physical system itself. Models are a tool to do this, but not the fundamental aim. In order to use a model to make meaningful inferences about a physical system, the relationship between the model and the actual system must be rigorously specified. By rigorously we mean: spell out what it means, why it has that meaning, and why that meaning is relevant to the problem at hand. This is a high-level aim, to which we give a general structure as well as pointers towards implementation.



**Table 1.** Near complete list of paleo ice sheet reconstructions (restricted to within last 2 glacial cycles) from glaciological modelling published from 2010 to 2021. Third to sixth columns are respectively : 3: structural and initialization model uncertainty assessment, "MME" indicates a multi-model ensemble was used 4: number of glaciological model runs, 5: number of model parameters that varied between model runs (ensemble parameter dimension, E.P.D.), 6: whether a statistical model ("emulator") of the glaciological model was used to enable adequate statistical sample of model parameters ("GPE" is a Gaussian process emulator, "BANN" is a Bayesian artificial neural network emulator). Other terminology is explained in the next sections.

| study | context/description | str./init. uncert. assess? | E.P.D. | ens. param. dim. | emulat. | constraint data |
|---|---|---|---|---|---|---|
| Tarasov et al. (2012) | deglacial calibration of a glaciological model for North American ice sheet complex with MCMC sampling | limited, yes | $> 50,000$ | 39 | BANN | deglacial: RSL, marine limits, uplift, strand-lines, margins |
| Whitehouse et al. (2012) | geologically-constrained Antarctic deglacial model | no, no | hand-tuned | 9 discrete | no | geological constraints of deglacial ice extent and thickness |
| Kleman et al. (2013) | MIS5b to MIS4 N. Hemispheric ice sheet topography | no, no | hand-tuned | 2 | no | regional extent for two time-slices |
| Golledge et al. (2013) | LGM near-equilibrium Antarctic ice sheet | no, no | hand-tuned | ? | no | unclear |
| Briggs et al. (2014) | last 2 glacial cycle Antarctic glaciological model ensemble | no, yes | 2929 | 31 | no | PD geometry and deglacial: RSL, marine and terrestrial ages |
| Pollard et al. (2016) and Chang et al. (2016) | deglacial West Ant. ice sheet | yes, no | 625 | 4 | GPE | as above & inferred past grounding lines & vertical velocities |
| Patton et al. (2016) | Eurasian ice complex build-up to LGM | no, no | hand-tuned | 5 | no | various (not tabulated) geological constraints |
| Patton et al. (2017) | last deglacial history of Iceland | no, no | hand-tuned | unclear | no | as previous & tabulated marine ages |
| Salter et al. (2022) | North American deglaciation history matching with emulated climate forcing | no, no | ? | ? | GPE | geologically inferred ice extent and ice volume from Tarasov et al. (2012) |
| Albrecht et al. (2020) | last glacial cycle Antarctic ice sheet | no, no | 256 | 4 | no | grounding-line locations, elevation–age data, ice thickness, surface velocities and uplift rates |



**Table 2.** Examples of present to future ice sheet and climate modelling that at least partially address uncertainty quantification. Third to sixth columns are respectively (as per previous table) : 3: structural and initialization model uncertainty assessment, 4: number of model runs, 5: ensemble parameter dimension, 6: whether a statistical model ("emulator") of the glaciological model was used to enable adequate statistical sample of model parameters ("GPE" is a Gaussian process emulator, "RBF" is a radial basis function). "AIS" is Antarctic ice sheet and "GCM" General Circulation climate model. Other terminology is explained in the next sections.

| study | context/description | str./init. uncert. assess? | E.P.D. | emulator | constraint data | distinguishing features |
|---|---|---|---|---|---|---|
| Sexton and Murphy (2011) | Bayesian GCM calibration for present-day | from MME | 31 | GPE | 6 eigenvector × 12 climate fields | |
| Edwards et al. (2021) | Future land ice mass loss | partial from MME subsampling | 1-3 | GPE | none | emulation of multi-model ensemble and extensive sensitivity/robustness tests |
| Gilford et al. (2020) | future AIS contribution to sealevel | no, no | 2 | GPE | Eemian sealevel | Bayesian updating against last interglacial mass loss |
| Wernecke et al. (2020) | future 50 year mass loss from Ant. Amundsen Sea Embayment | from history matching, no | 3 + 2 binary | GPE | 1992-2015 surface elevation | history matching & Bayesian calibration |
| Lowry et al. (2021) | Future AIS mass loss | no, no | 5 | GPE with RBF kernel | historical sealevel | emulator has time dependence |

This overview is intended for all who are concerned with either inferring past (on glacial cycle timescales) ice sheet and/or climate system evolution or using the results of such inferences (be they modellers, data gatherers, or mongrels). Our goal is that after a careful read, you will at least be able to more critically evaluate uncertainty assessment for such modelling contexts. This includes gaining some awareness of some common inferential errors underlying virtually all paleo modelling studies to date that purport to infer past system evolution. Much of the discussion should also be relevant to those interested in inferring other aspects of the paleo environment for contexts that can benefit from rigorous integration of modelling and paleo data. Modellers of geophysical systems working on present-day or future time-frames may benefit from a selective reading keeping in mind the context dependency of much of the implementation discussion. The presentation is largely conceptual given the breadth of the intended audience. However, clearly delineated sections offer some more specific and technical guidance for those interested in actual implementation and further development of relevant methodologies.

Below, we lay out a broad framework for assessing uncertainty (Sect. 2), consider a few stepping-stones (Sect. 3), provide guidance on implementation (appendix A), and suggest some key concrete steps we believe are necessary for the community to move forward (Sect. 4). The framework starts with a consideration of the Bayesian approach. Unlike classical frequentist statistics, this approach to uncertainty quantification can rigorously determine a meaningful assessment of probability of, for





**Table 3.** Road-map for various audiences

| Audience | Sections to read |
|---|---|
| Paleo modeller interested in implementation | all |
| Mathematically competent paleo researcher not interested in implementation | all but sections 5.6-5.7 and appendices A2-A6 |
| Mathematically limited paleo researcher | "" and also skip the technical parts of section 2.4 and 2.7-2.8 |
| Any paleo researcher with a working understanding of Bayes Rule | can skip all of section 2 except 2.5, 2.6, and 2.9 |
| modeller in other geophysical contexts | selective reading as per competency (noting above) and interest |

instance, mean sea level having a specific value at last glacial maximum. The road-map (table 3) provides targeted guidance to the main types of intended readers. Given the still prevalent lack of understanding of Bayesian inference among much of the paleo science community, the presentation begins with a high-level introduction to Bayesian inference for the paleo context.

## 2 the Bayesian framework

This section provides the conceptual framework for Bayesian inference by considering each term in Bayes theorem in turn along with an examination of how distinct sources of uncertainty can be meaningfully assessed. It also provides a simple example of how inference can be broken when uncertainties are not appropriately addressed. The correspondence between Bayesian inference and the scientific method is underlined and should aid the reader in the understanding of Bayes theorem. None of these issues can be well addressed without consideration of what "probability" means, to which we first turn.

### 2.1 "What is Probability?"


In addressing uncertainty, the issue of what is probability is unavoidable. Is it purely an individual's judgment, subject to the rules of probability theory, about how likely some inference or event is? This is crudely the personalistic interpretation of probability. Or does it reflect something more objective? If so, what does "objective" mean? To what extent are probabilities independent properties of the world? When you make a probabilistic statement concerning your own research, do you believe 115 your statement (and not just in the sense "this is what follows if I apply this statistical tool")? What does your probabilistic statement mean to you? How does it actually apply to the world around us? Or perhaps to be more concrete, how much of one week's salary would you bet that your assertion is true with the odds that follow from your probabilistic statement?

No matter what interpretation of probability one chooses, the assignment of probabilities require judgements. To be testable and potentially falsifiable, these judgements must be made and treated in a rigorous and self-consistent way. A natural way to 120 achieve this is to require that probabilities follow the rules of probability theory (*e.g.,* Kolmogorov, 1933). Specifically :

1. The probability of an event is a non-negative real number.





2. An impossible event has 0 probability.

3. A certain event has probability of 1.

4. Probability of events A or B is equal to the sum of their individual probabilities (P(A or B)=P(A)+P(B)) when A and B
are disjoint, *i.e.* one can happen only when the other can't happen.

Adherence to the rules of probability ensures that once required judgements are transparently made, all ensuing probabilistic calculations are acceptable to all competent practitioners given the premises (even if the subsequent interpretations may vary). A key consequence of the rules of probability is Bayes rule.

## 2.2 Bayesian inference for the paleo community: the scientific method expressed in probability theory

Bayesian inference is a rigorous formalism for evidence-based reasoning within a probabilistic framework. This contrasts with traditional frequentist statistics which does not directly determine probabilities and necessarily assumes repeated experiments. Bayesian inference is encapsulated in **Bayes rule** itself:

$$\textbf{Posterior probability} \quad = \quad \frac{\text{Likelihood} \times \text{Prior probability}}{\text{normalization constant}} \tag{1}$$

| **Prior probability** | : | Our probability that the hypothesis is true before seeing the new data |
| **Likelihood** | : | Likelihood of the new data if the hypothesis were true |
| **normalization constant** | : | A term to ensure that the probability that the hypothesis is either true or false equals one |
| **Posterior probability** | : | Revised probability of our hypothesis given the new data |

The unpacking of what each of the terms mean and how they are determined will proceed below with dedicated subsections for each term on the right-hand side. As a start, understanding of some basic notation is required. The expression $P(A = a \mid B = b)$ **denotes the conditional probability of the variable A having some specific value "a" if the statement that the variable**
**B has some value "b" were true**. For example, $B$ could have value 1 if a random African citizen had height greater than 2 m and $A$ could be their gender. $P(A = \text{male} \mid B = 1)$ would then be the conditional probability that a random African citizen were male, if we only knew that their height was greater than 2 m.

Conditional probability is usually abbreviated to exclude explicit mention of of actual values (*e.g.,* $P(A \mid B)$ ) when consid-
ering the probabilistic relationships independent of any specific value a or b (from now on a lower case letter will refer to a specific value of the corresponding upper case variable). One further fundamental probabilistic quantity is the **joint probability of $A$ and $B$: $P(A, B)$. This is the probability that $A = a$ AND $B = b$**. To be consistent under various sets of natural axioms, the conditional probability $P(A \mid B)$ can be defined as the quotient of the joint probability of $A$ and $B$, $P(A, B)$, and the probability of $B$, $P(B)$:

$$P(A \mid B) = \frac{P(A, B)}{P(B)} \tag{2}$$





To concretize these equations, consider $A$ having two values: rain ($A$ =rain) or no rain ($A$ =no rain) tomorrow afternoon. Let $B$ also have two values: cloudy ($B$ =cloudy) or clear sky ($B$ =clear sky) conditions tomorrow morning. Then the conditional probability that tomorrow afternoon will have rain if tomorrow morning is cloudy, $P(A =\text{rain}|B =\text{cloudy})$, is equal to the joint probability of rain tomorrow afternoon and cloudy skies tomorrow morning, $\text{P}(A =\text{rain}, B =\text{cloudy})$, divided by the probability

of cloudy skies tomorrow morning ($P(B =\text{cloudy})$. The crucial difference between joint and conditional probability is that the former has no input condition, while the later has the given condition (*i.e.* that the statement $B$ =cloudy is true in our example).

Multiplying both sides of equation 2 by $P(B)$ gives the **multiplication rule for probabilities:**

$$P(A, B) = P(A|B)\, P(B) \tag{3}$$

The above may also provide a more understandable implicit definition for conditional probability, *i.e.* $P(A|B)$ being that

quantity which when multiplied by $P(B)$ provides the joint probability $P(A, B)$. [1]

To place Bayes rule in a more useful formulaic representation, consider a set of variables that collectively defines the current configuration of a model. This is will be referred to as the **parameter vector** and denoted by $C_M$ (Configuration of Model). For a computational model, $C_M$ includes model parameters (such as basal drag coefficients for sliding ice in an ice sheet model, coefficients controlling cloud formation in climate models,...). $C_M$ can also define the selection of external forcings

or inputs (*e.g.,* to define parametrized aspects of the climate history that drives a stand-alone glaciological model). We are interested in determining a probability distribution for the most appropriate value of $C_M$ ("value" actually refers to the set of component values). Prior to testing the model against new data, one's initial (or previously derived) judgement for the probability distribution of $C_M$ is denoted by $P(C_M)$. One then compares new data against model results to determine a revised (posterior) probability distribution for $C_M$ given comparison of new data ($D$) against model results is given by Bayes rule (or

conceptually in above eq. 2):

$$P(C_M|D) = \frac{P(D|C_M)\, P(C_M)}{P(D)} \tag{7}$$

Bayes rule follows from the given definition for conditional probability (eq. 2) or equivalently from the axiomatic multiplication rule):

$$
\begin{aligned}
P(C_M|D) &= \frac{P(C_M, D)}{P(D)} \\
&= \frac{P(D|C_M)\, P(C_M)}{P(D)}
\end{aligned}
\tag{8}
$$


---

[1]An important self-consistency feature that we will use below is that the multiplication rule holds even if an additional conditional variable is added to both sides of the above equation, as follows by repeated use of the definition of conditional probability (equation 1), sic:

$$
\begin{aligned}
P(A, B|C) &= \frac{P(A, B, C)}{P(C)} \tag{4}\\
&= \frac{P(A, B, C)}{P(B, C)}\, \frac{P(B, C)}{P(C)} \tag{5}\\
&= P(A|B, C)\, P(B|C) \tag{6}
\end{aligned}
$$



From Bayes rule, the posterior probability for a single parameter vector $C_M$ given data $D$ is proportional to the product of the likelihood ($P(D|C_M)$) and prior probability ($P(C_M)$). It should be noted that there is also implicit dependence on all the assumptions and judgements underlying one's use of Bayes rule as we'll further explore below[2].

To make the above concrete, consider our previous rain and cloud example. Let $C_M$ represent our "model" for whether there will be rain in the afternoon and that it can have only two values (Rain or noRain). Let $D$ be the datum of whether the sky is clear or cloudy when we look out the window after waking up. Suppose based on climatology, and the fact that we had no rain yesterday, our prior, *i.e.* before looking out the window, is $P(C_M = Rain) = 0.3$ (and therefore $P(C_M = NoRain) = 0.7$). Further suppose our likelihood has the following values:

$$P(D = clear | C_M = Rain) = 0.1, \tag{9}$$

$$P(D = clear | C_M = NoRain) = 0.8. \tag{10}$$

The remaining two likelihood values are given by the difference from one (*e.g.,* $P(D = cloudy | C_M = Rain) = 1 - P(D = clear | C_M = Rain) = 1 - 0.1 = 0.9$). After we look out the window in the morning and see that the sky is clear, our posterior for rain this afternoon is given by Bayes rule :

$$P(C_M = Rain | D = clear) \quad = \quad \frac{P(D = clear | C_M = Rain) \times P(C_M = Rain)}{P(D)} \tag{11}$$

$$= \quad \frac{0.1 \times 0.3}{0.1 \times 0.3 + 0.8 \times 0.7} = 0.05 \tag{12}$$

The value used for denominator $P(D)$ follows from requiring that that sum of posterior probabilities for rain and no rain ($P(C_M = Rain | D = clear) + P(C_M = NoRain | D = clear)$) must $= 1$.

To facilitate a broader interpretation of Bayes rule, consider the idealized form of the scientific method as taught in primary school science or described in Wikipedia. In detail, start with a hypothesis based on current understanding, carry out an experiment, and thereby test the hypothesis and revise our understanding. Bayes rule translates to this description of the scientific method as follows. **Express initial understanding** about the appropriate choice of model configuration in the form of a prior probability distribution ($P(C_M)$). **Get new data** $D$ (from a data archive, laboratory experiment, or fieldwork). **Compare this data to model predictions via the likelihood** function ($P(D|C_M)$), which is the conditional probability of the data given the parameter vector (and therefore given the model, as the parameter vector has no meaning without an associated model). Hence **revise one's understanding** (update the probability distribution for model parameters and/or competing models) of the phenomena under consideration **as expressed in the posterior**, $P(C_M|D)$, *i.e.* the conditional probability of the parameter vector (and model) given the new data. The likelihood comparison embodies the experimental test of the model.

"Model" is any approximate representation of some aspect of the world around us (past/present/future), and thereby encompasses not only computer-based models, but also conceptual models including scientific hypotheses and theories. For nonmodellers, the evolving sedimentation rate of a marine or lake core and associated chronology is perhaps a conceptually closer example of a model. Where relevant, we will adopt the convention of specifically denoting an implemented computational model by "simulator" (to differentiate from the set of physical equations that the simulator is approximating).

---

[2]In the above and throughout, if variables are continuous, probabilities would be replaced by corresponding probability density functions.



Given this correspondence to the scientific method, Bayes rule provides a core probabilistic underpinning for the natural sciences. It should therefore arguably be understood by all scientists whose work involves comparison between data and models
(in the previously presented most general sense of "model"). To pursue this further, each term in Bayes rule is detailed below. Those interested in a much more thorough introduction are referred to one or both of the following texts: Sivia and Skilling (2006); Gelman et al. (2013). Rougier (2007) provides a complementary presentation to much of this section for a general climate modelling context. Kennedy and O'Hagan (2001) is a good technical starting point for a standard Bayesian approach to uncertainty in computer models.

## 2.3   Defining the prior

The prior ($P(C_M)$) embodies one's initial understanding of the system in question. Initially, before the first comparison between simulator prediction and data, the prior will be formulated on the basis of explicit user/expert judgement, likely in combination with results from other studies. The initial prior for a single model parameter, for instance, may be judged to be a trapezoidal distribution over a parameter range encompassing physically plausible values, an example of which is shown by the example 1
prior in green in Fig. 1. For the example of a marine core, the prior for the sedimentation rate would require expert judgement taking into account inferred rates from proximal cores, possible range of sedimentation variations given plausible ranges on biological productivity,...

This judgement aspect has often been a target by critics of Bayesian approaches, with a usual focus on the specification of the prior. This focus has no clear justification as judgements are required for all aspects of the inferential process and not just the
initial specification of the prior. But this holds true for any statistical inference including those by frequentist approaches. For instance, many scientists will rely on standard frequentist statistics such as $\chi^2$, confidence intervals from standard deviations, p-value tests, ... These often presume an underlying Gaussian structure to the uncertainties which all too frequently is not explicitly stated let alone tested. Judgements in the choice of constraint data, simulator, and which simulator parameters will be calibrated are required under any inferential approach. More fundamentally, traditional frequentist approaches do not offer
a route to a rigorous probabilistic inference about past ice or climate (in part because the frequency of occurrence of a single possible non-repeated event has no meaning within the frequentist framework). Nor does the frequentist approach allow a direct inference about a model given the data. It instead makes inferences about the data, assuming that a model is true.

Bayesian approaches tend to force the user to make all such assumptions explicit. Furthermore, for a sufficiently wide prior, the impact of the prior on the posterior diminishes as the specificity of the likelihood increases (*i.e.* as the constraint value of
the data increases). For the example in Fig. 1, this would translate into reduced posterior dependence on the choice between the two indicated priors (green and purple lines) with increased specificity (narrowness in the plot) of the likelihood (red and blue lines respectively).

Prior specification should therefore err on a clearly ample range over which the quantity of interest has non-negligible probability. A prior that is set narrow without good reason may exclude a high probability region of the posterior that would
result from a more carefully specified prior. The trade-off is that excessively wide priors increase the computational cost of




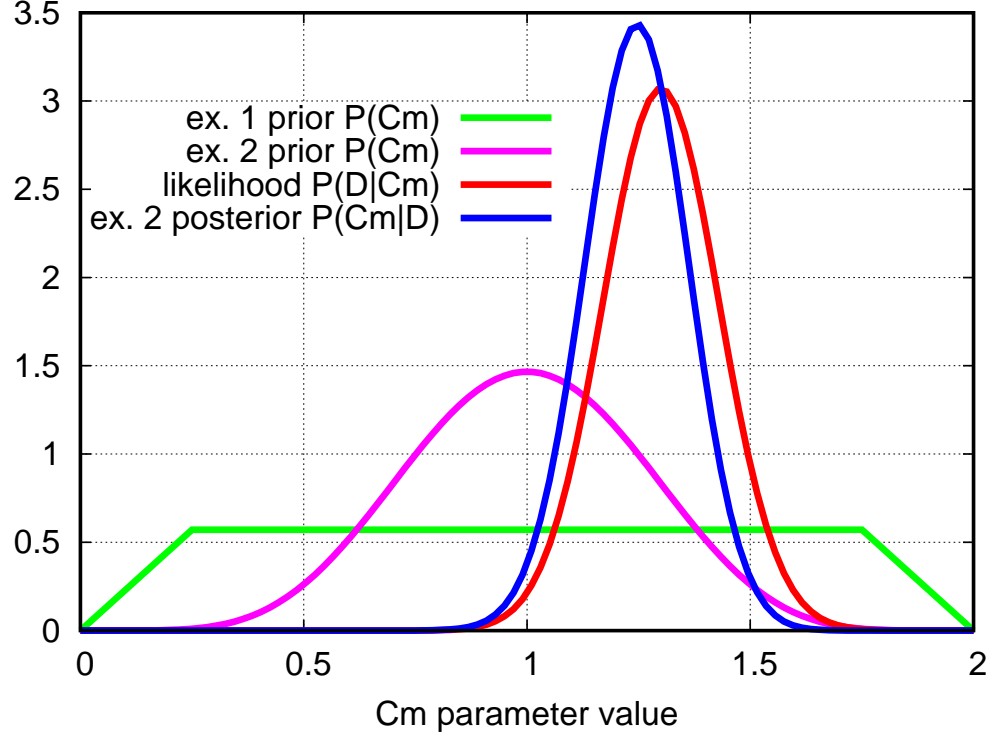

**Figure 1.** Example prior and posterior probability distributions for a parameter vector consisting of a single parameter ($C_M$) with maximum physically plausible range of 0 to 2. The likelihood (red) is also the posterior for prior example 1. Prior example 2 (purple) subject to the red likelihood results in the blue posterior.

the inference procedure. Thus the effort put into prior specification should be proportional to the computational expense of the simulator, and inversely proportional to the amount and quality of constraint data.

A key self-consistent feature of Bayes rule is that it can be applied sequentially, with the posterior from an initial application of the rule becoming the prior for a subsequent application with new data. Thus, after initial formulation of the prior, its

subsequent value for the same simulator for use with new data is rigorously set by the Bayesian inferential process. For our sedimentation example, an existing posterior inference for sedimentation rates could become the prior for a revised posterior from likelihood application of new C14 dates.

The judgement aspect of prior (and likelihood) specification does not imply all choices of prior specification are equivalent. Prior specification must obey the rules of probability. It must also be transparent, respect physical bounds, and be defensible to

the relevant community of experts (all normative criteria for "good" science). "Defensible" does not mean that everyone will make the same assignment of the prior. But it does mean that after reasoned argument, a contextually knowledgeable reviewer,



for instance, will not reject the assignment[3]. Like any other aspect of the scientific specification, sensitivity to judgements should be assessed by consideration of sensitivity and robustness to scientifically reasonable variations of the specification.

## 2.4   The likelihood function and error model

The likelihood, $P(D|C_M)$ embodies the comparison of model predictions to new data, *i.e.* data that was not used in the specification of the prior $P(C_M)$. If there were no uncertainty in the system and an adequate amount of data to uniquely distinguish each model configuration was available, then $P(D|C_M)$ would simply take the value 1 for the correct ($C_M$) model, and 0 for all other model configurations. It is the presence of uncertainty[4] which necessitates non-trivial probability assessment.

This uncertainty is usually first broken down into at least two broad categories. The most familiar category is observational

uncertainties accounting for discrepancies between observational data and the physical system. This includes measurement uncertainties, such as in measuring the elevation of a fossil mollusc shell, and uncertainties in the relationship between the measured value and quantity of interest. The latter are often referred to as proxy or indicative meaning uncertainties, an example of which is the difference between the present-day elevation of a fossil mollusc shell sample and past relative sea level.

The category that is often ignored is simulator uncertainties due to limitations of the simulator in representing "reality". This includes limitations in simulator structure, initialization, and inputs.

It is useful to further break down simulator uncertainties into those that are reducible (via an improved choice of simulator inputs and/or parameters), and those that are irreducible (for at least the near future, *e.g.,* due to climate models not resolving individual clouds). The irreducible errors are often referred to as the structural uncertainty or structural discrepancy.

The relationship between observational and simulator errors, simulator prediction, and observations can be conceptually summarized as follows:

*observation = Reality + observational error*

and on the simulator side:

*Reality = simulator prediction + simulator error*

Substituting for Reality into the first word equation :

*observation = simulator prediction + simulator error + observation error*

rearranging =>

*observation − simulator prediction = simulator error + observational error*

However, we do not know the actual errors, which would require apriori accurate identification of the state of reality. We therefore replace errors in the above relationships with a statistical error model for the distribution of these errors. For instance,

---

[3]Defensibility and transparency distinguishes the Scientific Bayesian interpretation, achieved by considering the spectrum of defensible expert judgement within the field (Goldstein, 2006).

[4]The impossibility of exact inversion for non-trivial simulators also leads to this necessity.



the most common error model for the measurement error component of observational error is a Gaussian distribution with 0 mean (assuming the measurement device has been properly calibrated) and specified standard deviation.

The error model not only includes the chosen statistical distribution, but also the chosen dependencies. For measurement
error, a common assumption is that the measurement error at one time and/or location is independent of the measurement error at another time and/or location. In actuality, most measurements, at least in a paleo context are likely to have some error correlation in both space and time. Example sources of this correlation include the local reference error for field measurements of sample elevation as well as sample preparation and mass-spectrometer calibration contributions to errors in radiocarbon age extraction. However, for most paleo contexts, inferential errors introduced from the assumption of measurement error
independence are likely of relatively minor impact. More substantial correlations over space and time in the error structure are often introduced, for instance, in the age calibration of the sample from uncertainties in the C14 reservoir age for a marine region.

Depending on the measurement device, the measurement error may or may not be independent of the measured value. For the dependent case, this often occurs when the measurement uncertainty is specified to be a constant percentage of the measured
value. In this case, a logarithmic transformation can still provide independence, which aids computational tractability.

Converting our above conceptual equation between an observation and model prediction to a probabilistic statement, we then get:

*P(observation − simulator prediction | simulator, configuration)*

*= P(observational error + simulator error | simulator, configuration).*

Symbolically, with "R" referring to the inaccessible reality (*e.g.,* ice sheet topography during last glacial maximum), the observational error is $D - R$. With "M" referring to model simulator and associated simulator output, the simulator error is $(R - M(C_M)$. The above word relation then becomes algebraically self-evident:

$$P(D - M(c_M) | M, C_M = c_M) = P((D - R) + (R - M(C_M)) | M, C_M = c_M), \tag{13}$$

with "$| M, C_M = c_M)$" indicating "conditioning" on the choice of model ($M$, though usually this is only implicitly assumed)
and the choice of model configuration ($c_M$). In line with the previous definition of conditional probability, "conditioning" here means that the probability of the term on the left-side of "|" is dependent on the assumed truth of what the terms to the right of "|" represent: *i.e.* that the choice of model and model configuration is appropriate. The right-hand side of the above equation provides the tractable form for specifying the likelihood as a function of error models for both observations (*i.e.* $D - R$ residuals) and the simulator ($R - M(C_M)$).

A key point to note is that as we are seeking a posterior distribution for the model configuration ($c_M$) to have only irreducible structural errors, the likelihood also needs to be conditioned on the model configuration having no reducible error. Therefore only the irreducible structural error model is used for the simulator error in eq. 13.





Given that the above relationship (eq. 13) is conditioned on $C_M$ (*i.e.* predicated on $C_M$ having a value $c_M$), the left-hand side of the above equation then equates to the likelihood function $(P(D = d \,|\, M, C_M = c_M))$[5] since:

$$P(D - M(c_M) \,|\, M, C_M = c_M) = P(D - M(c_M) = d - M(c_M) \,|\, M, C_M = c_M)$$

$$= P(D = d \,|\, M, C_M = c_M) \tag{14}$$

The first line just expands what $P(D - M(c_M) \,|\, M, C_M = c_M)$ means while the second line uses the conditioning on $M(c_M)$, *i.e.* the probability distribution for $D$ will be determined by the distribution for $D - M(c_M)$ as $M(c_M)$ is a constant given the conditioning.

Consideration of the given definitions may aid intuitive understanding. The likelihood function computes the likelihood that the simulator output is consistent with the data. This consistency is specified by the error model, *i.e.* the uncertainty distribution for combined discrepancies between data and reality and simulator and reality (together this gives the distribution of discrepancies between data and simulator).

The simulator error model specifies the structural error distribution (*i.e.* the errors that are irreducible over the range of model configurations ($C_M$)), so that Bayes rule then provides the posterior distribution for the simulator configuration with irreducible structural error. Other configurations will tend to have lower posterior probabilities, usually (but not always) associated with larger overall values of data model residuals ($d - M(c_M)$).

To make this more this concrete, consider a simple example of the likelihood for a single observational datum ($D = d$) assuming that both observational and structural error models are Gaussian distributions with 0 means and respective variances $\sigma_{obs}^2$ and $\sigma_{mod}^2$. Also assume that the two error models are independent of each other and of the model configuration. Denoting the corresponding model output as $m(c_M)$, then the likelihood is

$$P(D = d \,|\, m(c_M)) = P(d - m(c_M) \,|\, m(c_M)) \tag{15}$$

$$= P(d - m(c_M)) \tag{16}$$

$$= N(d - m(c_M); 0, \sigma_{obs}^2 + \sigma_{mod}^2) \tag{17}$$

where $N(\cdot)$ is the Gaussian density function with specified mean (0 in the above) and variance. The second line above used the assumed independence of the error distributions on $c_M$. The third line used the standard result that the sum of two independent Gaussian distributions is still Gaussian with variance given by the sum of the individual variances and mean given by the sum of individual means.

In summary, the likelihood is fully specified by the chosen error model. A major challenge is specifying the structural discrepancy component of the error model. Within most (if not all) scientific disciplines, the tendency to date has been to effectively ignore this source of uncertainty. This can result in large inferential errors as we now show by example.

---

[5]The likelihood function should not be viewed as a conditional probability function because $D$ is held fixed while the input $c_M$ is varied.





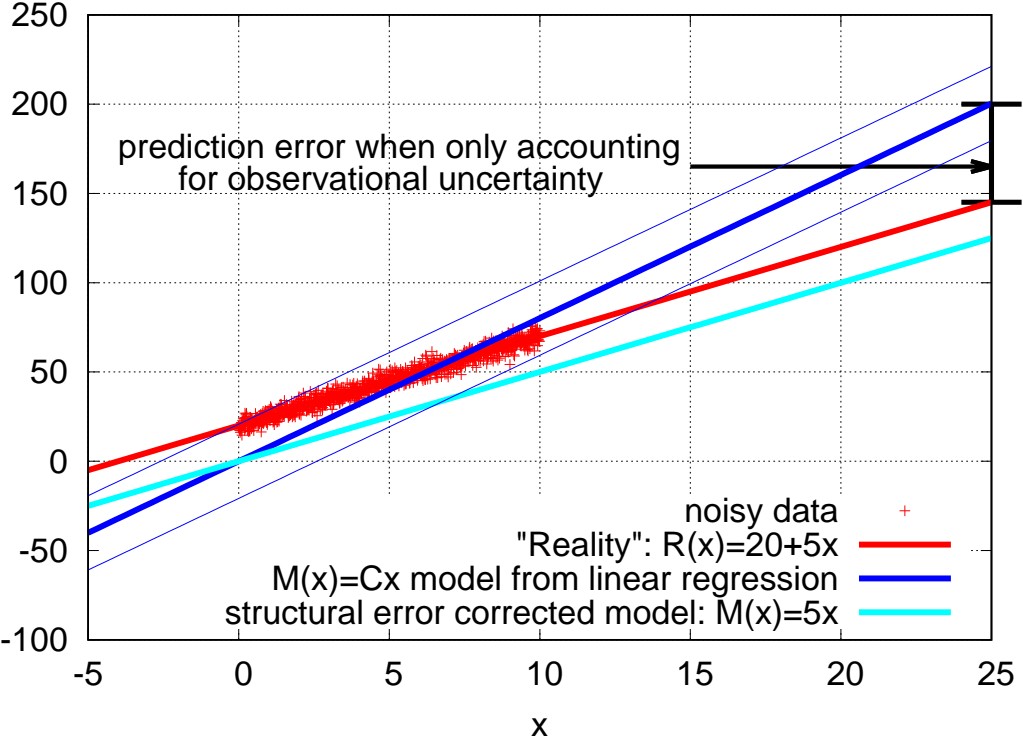

**Figure 2.** Linear regression example of inferential error that can occur when structural discrepancy is ignored. The red line ($R(x)$) is the underlying "reality" to which Gaussian noise is added to generate the 1000 sample data-points. The dark blue lines are the linear regression results (mean and $2\sigma$ confidence interval) for the structural incomplete approximation with 0 y-intercept.

## 2.5 A simple inference example illustrating the importance of accounting for structural discrepancy

As perhaps the simplest illustration of the importance of addressing structural discrepancy (irreducible model errors), consider the following toy model. Let reality be the simple linear relationship $R(x) = 20 + 5x$ (red line in Fig. 2), and given that our

models are always simpler than reality, let our model be $M(x) = Cx$, *i.e.* our model approximation of reality is forced to a 0 y-intercept. If we ignore the structural error of a forced 0 y-intercept, and blindly carry out standard linear regression against the data (with additive Gaussian noise) indicated in Fig. 2, then $M(x)$ and its $2\sigma$ confidence interval are the blue lines. The computed confidence intervals capture the relevant fraction of "observational" values, as expected for linear regression. However, the inferential value of this linear regression beyond the data range quickly becomes negligible. Not only does $M(x)$

diverge from $R(x)$ for larger x, but even the standard $2\sigma$ confidence interval fails to capture the true R(x) beyond the data range. Furthermore, an increase in the density of data-points limited to the indicated range of data will not improve the inference.

Given that models in the Earth Systems context are of interest for their predictions beyond the observational data range (across both space and time), this example shows the problematic consequences of ignoring structural discrepancy. Predictions



from simulations that "best fit the data" can quickly diverge from reality and, more critically, estimated uncertainty bounds can
significantly under-represent actual model-data differences.

Brynjarsdottir and OHagan (2014) offers a complete example of Bayesian calibration of a slightly more complicated toy
model that quantitatively elucidates how the lack of accounting for structural discrepancy will bias inferences. In both their
toy model and ours, it is also clear that when structural discrepancy is ignored although the addition of more data within the
existing data range will tend to reduce posterior variance, it will not reduce the inferential error: "with more and more data ...
we become more and more sure about the wrong value for" the model parameter (Brynjarsdottir and OHagan, 2014).

For less idealized examples that examine inferential errors when structural discrepancy is ignored or over-simplified we must
turn to the literature in other fields. For the context of a conceptual rainfall-runoff model, Schoups and Vrugt (2010) consider a
mis-specified error model that lumps together observational uncertainty and structural discrepancy into a Gaussian distribution
with 0 mean and constant standard deviation (a not uncommon practice). They show this results in a posterior distribution
for model parameter vectors that has no overlap with those obtained when the error model is more appropriately specified.
Tavassoli et al. (2004) also provides an applied example in the context of geological petroleum reservoir estimation with no
structural uncertainty assessment of how the models that "best fit the data" can have "bad" inferential value.

Given the structural limitation imposed on our toy model, the useful inference would give the model $M(x) = 5x$ (light-blue
line in Fig. 2). Though this model has a poorer fit to the observational data, its confidence interval captures $R(x)$ beyond the
data range. As such, the model has clear predictive value.

Consideration of even this simple illustration raises the difficult challenge of how to account for the structural uncertainty.
Conceptually, having access to "reality" ($R(x)$, red line in Fig. 2), we can easily determine the structural error as a constant
bias: $R(x) - M(x) = 20$. However, for realistic contexts, complete knowledge of reality ($R(x)$) is inaccessible. Dropping such
omniscience for our toy example, as long we have confidence in our observational uncertainty (especially that observational
uncertainties do not grow at the ends of the data range), one could test a range of values for the slope C, and then note the
simplification in the structure of the residuals as the test value approached the actual value of the underlying reality[6] (Fig.
3). But in considering models of earth system components in four (space-time) dimensions, such inference becomes much
more difficult as the underlying residuals between model and reality will have a non-trivial multivariate structure (*i.e.* model-
observations residuals from proximate observations being more strongly correlated) that will generally not correspond to any
standard statistical distribution.

An additional challenge is that the high complexity and associated non-linearity of current models of geophysical systems
such as ice sheets and climate makes inferences with such models potentially much more sensitive to inaccurate specifica-
tion of structural discrepancy. **In summary, unless structural discrepancy is accurately accounted for, or at least not
under-estimated, even studies that claim to use Bayesian inference may have much less retrodictive value and statistical**
**confidence than indicated**.

---

[6]It should be noted that the blind linear regression in this example would be rejected by standard statistical diagnostics or by simple inspection of the
residuals shown in the figure.




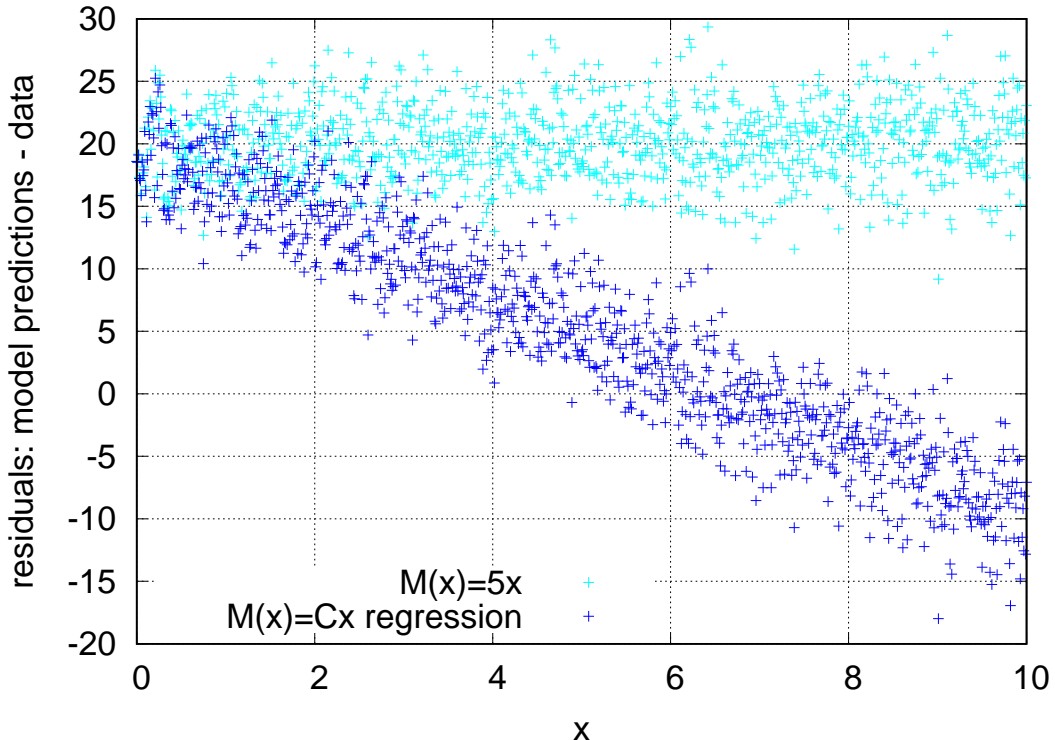

**Figure 3.** Data - model residuals for the toy model in Fig. 2.

## 2.6 Addressing structural discrepancy

In order to address structural discrepancy, it is useful to first break it down into two components. Internal discrepancy is the component that is assessed by experiments with the simulator and external discrepancy is the remainder. Depending on context, we also separately differentiate potential internal discrepancy as the component of external discrepancy that could be assessed
by tractable simulator experiments but to date have not been.

### 2.6.1 Internal discrepancy

Examples of internal discrepancy for paleo ice sheet models include the uncertainty in subglacial topography and deep geothermal heat flux for Antarctica and Greenland, uncertainty in initial conditions (such as the temperature field of the initial ice sheet, temperature field of the solid earth, initial isostatic disequilibrium, surface sediment distribution), and noise (randomness) in
the climate forcing. For paleoclimate models, examples include: all initial conditions, ice sheet and topographic boundary conditions, atmospheric concentrations of aerosols and dust, and some of the contributions from unresolved turbulence (as much of the impact of turbulence won't be assessable by tractable simulator experiments). Explicitly stochastic components of a model representing such processes would also be contributors to internal discrepancy.





As a concrete example of an internal discrepancy determination for a single discrepancy source, consider the uncertainty of
the subglacial topography under Antarctica and resultant impact on modelled glacial cycle evolution of Antarctica. After a pilot
exploration to find some ice sheet simulations that are not rejectable based on interim assessed uncertainties (as explained in
subsections 2.6.2, 3.1, and appendix A1), one could extract a small, high-variance, and low collinearity (*i.e.* parameter vector
directions well-separated) subset of approximately 10 parameter vectors[7]. By "high-variance", we mean the subset of param-
eter vectors that have the highest (or near highest) sum of vector component variances after component-wise normalization
across the whole set of simulations[8]. The associated simulations should also have high variance and low collinearity for values
of metrics not directly constrained by paleo data (such as ice volume or mean annual precipitation over North America for
some glacial time slice). The internal discrepancy determination would involve repeatedly re-running the simulator for each
of the parameter vectors in this subset with different bed topographies. These topographies could be created by adding ap-
propriately correlated noise to the available bed topography used in the original simulations. The correlated noise should be
fitted to observed topographic variations in geologically-corresponding regions that are free of ice cover after accounting for
the sampling length scale of the observations used to create the original subglacial bed topography.

Each parameter vector from the small subset would thus have an associated noise sub-ensemble of simulations with different
topographic realizations. From each such sub-ensemble, one could then extract appropriate statistics, such as the variance
and covariance, of relevant simulator output[9]. These statistics would provide estimates for the statistical parameters in the
probability density function for the internal discrepancy. For example, for each simulator parameter vector, one could extract
the variance of present-day simulated ice volume relative to the mean of the sub-ensemble. The maximum sub-ensemble
variance of each relevant simulator output across the set of sub-ensembles would provide an initial approximate estimate of
the internal discrepancy variance of the simulator arising from topographic uncertainty. The stability of the assessed internal
discrepancy could be assessed by examining how much present-day ice volume variance varied between the sub-ensembles.
For low stability, more basis parameter vectors should be added. Hebeler et al. (2008) and Gasson et al. (2015) offer concrete
examples of assessment of simulator sensitivity to topographic uncertainty for an ice sheet modelling context (though the
assessment was limited to a few basic ice sheet characteristics and without consideration of covariance between different
simulator outputs).

The set of variances and covariances from the set of noise sub-ensembles would be used to define a common variance/co-
variance representation for the given noise source (e.g., topographic uncertainty, basal drag uncertainty,...). This representation
could then provide a multivariate Gaussian representation of the internal discrepancy. For spatio-temporal fields (such as tem-
perature), the variance and covariance will need to include spatial and temporal dependencies. It may well be that the structure

---

[7]The choice of the number parameter vectors will be somewhat context dependent, but should at the very least be above 2 to extract at least a minimal
indication of how stable the results are across the parameter space.

[8]This example criterion is not meant to be prescriptive. It may, for instance, be better to choose the subset of parameter vectors that maximize the average
of the sum of component variances and the minimum of component variances.

[9]For those statistically literate, in the case of a multivariate Gaussian error model, one would extract the variance/covariance matrix for each sub-ensemble.
Matrix dimensions would include relevant simulator output for comparison against either observations (such as ice volume and basal temperature at ice core
sites) or paleo proxies (such as ice extent at given time and location).





of the internal discrepancy is non-Gaussian and/or has significant dependence on ensemble parameters. For both of these cases, a more generalized statistical model is required to represent it (*c.f.* appendix A6 on emulation).

Discrepancy from process uncertainties can also be partly assessed by similarly adding appropriate time-varying noise to, for instance, diffusion coefficients (to better represent turbulent mixing/transport) in climate models or to basal drag components in ice sheet models (to *e.g.,* account for the impact of changing subglacial water pressure). If the strength or structure of this noise addition is subject to parametric control, then the parameter can be added to the simulator ensemble vector and this source of process uncertainty can be partly subsumed into the simulator ensemble parameter calibration. However, model sensitivity

to the sampling of this noise (for a given parameter vector) will still need to be accounted for in the internal discrepancy assessment. This would entail repeated simulations with different initial seeds of the noise generator for the given process.

Internal discrepancy assessment is an important part of the simulator development process. Large sources of internal discrepancy point to processes that need to be explicitly incorporated into the simulator under ensemble parametric control. For our topographic glaciological example above, this might take the form of parametrizing the structure of the topographic noise,

perhaps via mean amplitude and wavelength and/or variance parameters, and then adding these parameters to the ensemble parameter vector for the simulator. Prior ranges for these parameters would need to be consistent with available observational constraints.

For inferential contexts, one would only need the internal discrepancy due to the combined effects of all sources at the same time. This would entail imposing all internal discrepancy noise sources simultaneously for each internal discrepancy

simulation. A concrete example of internal discrepancy assessment is provided in Goldstein et al. (2013).

To date, common practice implicitly assumes structural discrepancies are either minimal or, in the case of bias correction, constant across different simulator states. These assumptions underlie the common reasoning that a model that adequately fits past observations will have predictive value for the future. Internal discrepancy assessment can quantify the validity of such reasoning.

**2.6.2 External discrepancy**

The external structural discrepancy is the component of structural uncertainty not directly quantified by computer experiments. Some common sources of external discrepancy are limited grid resolution of the simulator, parametrized (or ignored) subgrid processes, and uncertainties in inputs for paleo contexts. Determining an appropriate external structural discrepancy is one of the most challenging aspects of applying Bayes rule to the inference of past earth system evolution.

The previous toy example makes clear that the models that best fit the data within observational uncertainty (dark blue line in Fig. 2) can not be directly used to extract the structural discrepancy by setting the latter to the model-data residual less observational uncertainty (*i.e.* difference between the blue line and red data-point error bars). Bayes rule is predicated on the independence of the choice of the error model used to create the likelihood function from the model-data residuals input into the likelihood function. Concretely this means that initial internal and external discrepancy assessment must be done without

consideration of model fits to observational data embodied in the likelihood. However, as discussed in the implementation section below, diagnostic testing can invalidate an error model and partially guide its subsequent re-specification.



The importance of structural discrepancy assessment can also be understood from the perspective of avoiding simulator over-fitting. The latter occurs when the predictive power of a model is sacrificed to increase model fits to data. Concretely, any set of N data points can be exactly fit by an $(N-1)$-th order polynomial. However, if the data contains any observational noise, then the regression is fitting reality plus noise, resulting in a loss of predictive power. This is especially clear when extrapolating beyond the data range given how fast high order polynomials can grow in magnitude as the independent input variable increases in magnitude. Bayesian model calibration would ensure model fitting to data is only within the context of a well-specified complete error model and thereby avoid over-fitting[10].

There are a variety of complementary approaches towards addressing external structural discrepancy. They all start with assessing scientific understanding about sources of external discrepancy and making judgements about the resulting error for both prediction/retrodiction and likelihood comparison to constraint data. The judgements may be as simple as "given ice sheet model intercomparisons, dynamical sensitivities, a typical 20 km model grid resolution, and my own extensive experience as an ice sheet modeller, the fit to the observed grounding line positions for a model of the Antarctic ice sheet is likely to have a root mean squared error of at least 30 km. Errors in grounding line position will in turn propagate into correlated errors in both past and present-day ice shelf areas." Each of these example considerations can then be backed up by appropriate referencing. The explicit specification of structural uncertainty enables reasoned evaluation in both the review process and more broadly in the general community.

A second approach posits a multi-model ensemble (MME) of different state-of-the-art models as a closer representation of reality. A partial estimate for simulator external discrepancy can then be extracted from the distribution of residuals between each MME member and the closest member of a perturbed parameter ensemble (PPE), of the simulator under consideration. This requires a chosen metric to compute the distance between model runs. This approach would be of value if the structural discrepancy of the simulator of interest is significantly larger than the structural discrepancy of the MME. Sexton et al. (2011) used this approach with a PPE for the HadSM3 climate model to generate probabilistic projections for the 21st century. They show (*e.g.,* their figure 9) a significant impact on projected annual global mean temperature change if their choice of structural discrepancy is ignored.

A challenge in the use of such MMEs for discrepancy assessment is that available MMEs are generally ensembles of opportunity as opposed to an ensemble of simulations that was designed for discrepancy assessment. As such, they incorporate models that are not independent (given the commonality of parameterizations, dynamical cores, grid resolution, and so forth) and to date are almost always only hand-tuned with no principled uncertainty assessment. Furthermore, ensemble parameter vectors have not been optimized for discrepancy assessment. All these factors limit the utility of current MMEs for structural discrepancy assessment.

One could also use a computationally expensive high-quality simulator to diagnose the structural uncertainty of a lower-quality, but computationally cheaper, simulator. Such an approach could be used to quantify a significant component of the external structural discrepancy contribution from an ice sheet model (ISM). It would require an ensemble of transient simulations of a high-quality ISM (such as a high resolution adaptive-mesh model with a higher order representation of ice flow,

---

[10]For those with a machine learning understanding, one effect of the error model and prior is to collectively regularize the fitting of the simulator to the data.



*e.g.,* Cornford et al., 2013). Application of the same climate forcings to both the expensive high-quality ISM and the faster lower-quality ISM would permit extraction of a lower bound estimate for the lower-quality ISM external structural discrepancy. The completeness of this estimate would depend on the quality of components for processes at the physical boundary of the ice sheet in the high-quality model.

It should be noted that if the high-quality simulator runs have yet to be done, there are much more efficient approaches for jointly using the information provided by a hierarchy of fast low-quality to expensive high-quality simulators (*e.g.,* Cumming and Goldstein 2009, as well the appendix concerning emulation).

     The fourth approach is to posit a parametrized form of the structural discrepancy and infer parametric coefficients. The posited form should be explicitly motivated by comparison with the results of other (hopefully structurally dis-similar) models

(thus overlapping with approach three above) and/or physical reasoning and/or expert judgement. Even better if at least some of these other models in the comparison were of higher quality (*i.e.* one that has higher resolution or invokes fewer approximations at the cost of increased computational expense). The form of the structural uncertainty can further be constrained by underlying symmetries in the system under consideration (cf Arthern, 2015, and references therein).

     The dividing line between internal and external discrepancy is in good part a choice by the modeller within the constraints of

available time and computational resources. Discrepancies due to uncertainties in climate forcing for a paleo ice sheet model can in part be assessed by the addition of appropriate noise to the climate forcing. Such assessment would then convert a non-trivial fraction of the external discrepancy due to climate forcing uncertainties to an assessed internal discrepancy. Expansion of internal discrepancy assessment and/or introduction of further simulator ensemble parameters and/or improvement to simulator components are all routes to reducing the challenge of external discrepancy assessment.

After careful consideration of structural discrepancy, the modeller may then judge it to be excessive for the given context. This would necessitate improvements to the simulator guided by consideration of the contributors to structural discrepancy. This involves some combination of : increasing simulator resolution, invoking fewer approximations in the simulator implementation of the relevant physics, and/or adding more parameterizations and associated calibration parameters.

     To date, most may posit (as is often implicitly or, less frequently, explicitly done) the option of ignoring structural discrep-

ancy and, for instance, just focus on finding the ice sheet chronology that most closely fits the observations within observational uncertainty only. One might then argue that the resultant chronology is that most consistent with data and physical relationships embodied in the employed simulator. If one's explicit purpose is to find such a chronology, in effect carry out simulator-based curve-fitting, this argument is unassailable. But what utility does this provide, especially for space-time locations where there are no high quality proximal observational constraints? If one is endeavouring to make an inference about the actual past and

not about one's model, than the above toy model example should make clear how incorrect such an inference could be when structural discrepancy is ignored. For the surface and groundwater hydrology contexts, clear real-world examples have been published over a decade ago (*e.g.,* Yang et al., 2007) showing the inferential errors that can ensue when structural discrepancy is ignored or under-specified.



### 2.7 The normalization term of Bayes rule, Markov Chain Monte Carlo (MCMC) sampling, and emulation

The normalization term, P(D), in Bayes rule need only be explicitly determined if absolute as opposed to relative probabilities are sought. To be a self-consistent probability, the sum of the posterior over all values of model parameters $\sum_{C_M} P(C_M \mid D) = 1$. From Bayes rule, this in turn

$$= \sum_{C_M} \frac{P(D \mid C_M) P(C_M)}{P(D)}. \tag{18}$$

Inverting this equation leads to (aka the law of total probability):

$$P(D) = \sum_{C_M} P(D \mid C_M) P(C_M) \tag{19}$$

If one were only concerned with determining which paleo history (as defined by $C_M$) was more likely for a given ensemble of histories, one could ignore $P(D)$ and simply compare the product of the likelihood and prior for each history. However, inferring past Earth/climate system evolution entails determination of absolute probabilities and complete sampling of the non-negligible posterior probability space of possible histories. This is where the computational challenge of Bayesian inference

becomes apparent. $P(D)$ alone entails a sum of model evaluations over all parameter vectors ($C_M$) and considered models (M) that have non-negligible prior probability and likelihood. For any model that can't be evaluated analytically, this sum must be approximated. For a single model system restricted to just 5 calibration parameters, even if we only considered a single parameter value from each decile of the prior, a simple factorial (grid) sampling would entail $10^5$ model runs. Especially when given an order of magnitude higher number of calibration parameters, this is clearly computationally unfeasible. Markov Chain

Monte Carlo (MCMC) approaches (*e.g.,* Andrieu et al., 2003; Richey, 2010, for accessible introductions) can significantly reduce this computational requirement via random walk searches through the parameter space, with the choice of walk steps weighted to the relative posterior probability of each trial parameter vector ($P(D)$ is not used).

The mechanics of MCMC sampling can be illustrated by a topographic example. Consider the problem of finding the lowest elevation point by in situ travel in a region of complex topography with sight-lines thereby restricted (*i.e.* given surrounding

topographic highs). Suppose the only tool is a set of small transponders that can relay local elevation and an individual with a super-human throwing arm. This individual will randomly (with weighting according to perhaps a prior distribution for direction and distance) throw one transponder beyond his/her sight-lines (and usually beyond a proximal topographic high) and receive the local elevation of the point where the transponder lands. If the elevation is below the present elevation, the individual would then proceed to this new location and repeat the sequence.

The crux of this illustration is the response to a resultant transponder elevation that is higher than the present position of the individual ( and therefore of lower analogous probability). If the individual were to consistently not go to regions of higher elevation, soon they would find themselves trapped in a local minimum (*i.e.* a valley deeper than adjacent valleys but likely not the deepest in the region).

To make this topographic search probabilistically self-consistent, or from a more limited optimization point of view, to

search for the global minimum, higher elevation regions also need to be sampled. One approach (approximately corresponding to the Metropolis-Hastings algorithm) is to make the decision on whether to go to a higher elevation transponder location via





a flip of a weighted coin. This weighting should be set to the relative probability of the transponder site being closer to the global minimum compared to the current location. Specification of this probability would properly need to take into account information such as the length scale of spatial correlation in elevation and surface slopes and the height of the transponder
relative to the current position. For a transponder site with 50% relative probability, this corresponds to flipping an unbiased coin, and only moving to that site if a heads results.

After convergence, the MCMC chains (sequence of sampled values such as measured elevations from the transponders in our analogy) will reflect the underlying posterior probability distribution[11]. This means that values from high posterior probability regions will occur proportionally more frequently than values from low probability regions of the parameter vector space.
This is a key difference from optimization algorithms that, for instance, will avoid repeated sampling of the same simulator parameter vector value. With MCMC, sampling from the full posterior may become more tractable. The actual number of sample points required will depend on the complexity of the model response to parameter variations or correspondingly the topographic complexity.

Continuing our topographic analogy, if one by chance starts in a high elevation plateau, it may take many transponder throws
before one one ends up sampling the lowest elevation regions. This corresponds to the initial pre-convergence "burn-in" part of a MCMC chain. Scaling up our topographic analogy from the given two spatial dimensions to the order 30 to 100 ensemble parameter dimensions of typical paleo simulators makes the burn-in phase potentially much longer.

The rougher and more complex the topography, the longer it can take to exit a sub-optimal region. Such conditions also increase the sensitivity of the MCMC chain (search path) to the exact specification of the likelihood and starting point.
When dealing with high-dimensional non-linear systems (such as earth and climate systems), just as one can never be confident that a global optimum has been found, so one can never be sure if an MCMC chain has converged. To address this challenge in part, any study using MCMC methods should use some combination of standard MCMC convergence metrics and multiple (preferably hundreds of) MCMC sampling chains, each started from a dispersed sample of ensemble vectors from the prior.
In the first author's own experience with ice sheet model calibration of approximately 40 ensemble parameters, at least order ten million point sampling is still required (as compared to the astronomical $10^{40}$ for a simple grid search over deciles). As this is still beyond computational tractability for ice sheet and climate models, one other component is required. This component, a set of emulators, consists of very fast approximate statistical models that predict statistical characteristics of simulator output of interest as a function of an input parameter vector.
A single emulator might just predict a probability distribution for simulator ice volume as a function of the parameter vector and time. Or it might incorporate latitude and longitude inputs to predict simulator deglaciation times at specified grid cells. Thus, a set of emulators could collectively predict model output required for likelihood evaluation. To maintain statistical integrity of the inference process, emulators, by definition, must embody a probabilistic distribution for simulator output and therefore also compute and output their predictive uncertainty.

---

[11]Technically this is true only when sequentially correlations of the MCMC samples are accounted for





A more technical discussion about emulation is provided in the appendix. Any study claiming to infer meaningful bounds on ice sheet or climate system evolution using relevant simulators will require emulators to at least address parameter uncertainty.

## 2.8   Uncertainty in making predictions/retrodictions: a missing term in most studies

Once a climate or ice sheet model has been calibrated, or an ensemble of simulations has been otherwise generated, the ultimate goal is to make prediction/retrodictions and/or improve process understanding. The determination of a collection of simulator

configurations consistent with constraint data after accounting for structural and observational uncertainties is only a first (though very large) step towards making a prediction. Consistent with the definition of structural uncertainty, predictions need to explicitly incorporate structural discrepancy. In our toy example, this would trivially entail adding a bias correction of 20 to the simulator prediction (light blue line in Fig. 2) to offset the structural error and recover the underlying signal (red line in Fig. 2).

Translating this toy reasoning into a Bayesian framework, one would seek the posterior probability for a future or past potential system state $S$ (or characteristic thereof, such as mean temperature) to equal value $s$ given constraint data $D$ and a simulator $M$ (or set thereof, by simply replacing $M$ with $\{M\}$ in what follows). Formally, this is denoted by:

$$P(S = s \,|\, D, M) \tag{20}$$

Invoking the standard statistical procedure of marginalization over all possible model configurations $(C_M)$, for the discrete case,

this equals

$$= \sum_{C_M} P(S = s, C_M \,|\, D, M) \tag{21}$$

Marginalization follows from the basic axioms of probability. It corresponds to the reasoning that $P(S = s)$, *i.e.* the probability of $S = s$ irrespective of the value of any other quantities (such as $C_M$), equals the sum of joint probabilities of $S = s$ and $C_M$, $P(S = s, C_M)$, over the complete set of mutually exclusive values of $C_M$. Application of the multiplication rule to eq. 21 then

provides a tractable expression in terms of the posterior distribution for simulator configurations $(P(C_M \,|\, D, M)$, *i.e.* from Bayes Rule, eq. 7) and the likelihood for $S$, $(P(S = s \,|\, C_M, D, M))$:

$$P(S = s \,|\, D, M) = \int P(S = s \,|\, C_M, D, M)\, P(C_M \,|\, D, M)\, dC_M \tag{22}$$

To make this more concrete, for a climate model, D might be the observed record of precipitation and temperature. $S$ would be a statistic for paleo or future temperatures or precipitation or any other climate system characteristic to be inferred from

simulator output. This statistic is any summary function of climate systems fields that are of interest, such as : mean monthly temperature, the exceedence or recurrence interval for some chosen extreme value threshold, yearly frequency of category 5 hurricanes,... For a paleo ice sheet model, $S$ would include ice sheet thickness and surface elevation over each geographic position and time, while D is the collective set of constraints used for the ice sheet model calibration.

    The likelihood $P(S = s \,|\, C_M, D, M)$ in the above equation makes explicit that any predictions must account for structural

discrepancy in the relevant simulator output (and not just in the simulator output data constraints used to determine the posterior





for $C_M$). As mentioned in the previous discussion on structural discrepancy assessment, the practice of bias correction is a simplified form of imposing the $P(S = s | C_M, D, M)$ term, by assuming that the total predictive error is a time-independent bias term (such as the difference between present-day reanalyzed temperature and simulated temperature from a climate model).

It should be noted, that the description to this point assumes a single best input choice for $c_M$. A more general approach,
especially when components are not physically well-defined, is presented in Sect. 3.1.

Aside from occasional bias correction, to date even most purportedly Bayesian studies with geophysical simulators ignore this crucial likelihood term and therefore underestimate predictive and retrodictive uncertainties. For those interested in implementation, Craig et al. (2001) offers a detailed example of Bayesian forecasting with a complex simulator.

## 2.9 Multi-model ensembles and dealing with structurally different simulators

For both paleo climate and paleo ice sheet (and sea level) contexts, there are a range of published simulations, many hand-tuned, some subject to ensemble-based scoring against limited sets of data constraints, and a few that attempt an approximate Bayesian inference. A natural question is how to make defensible joint inferences from such multi-model ensembles (MMEs) of simulations? This has received significant attention within the climate modelling community, in good part arguably driven by the challenge of drawing summary conclusions for International Panel on Climate Change (IPCC) contexts.

However, this topic has been clouded by a preponderance of ad-hoc approaches and reasoning. This is evident in contrasting conclusions drawn from existing literature: "Owing to different model performances against observations and the lack of independence among models, there is now evidence that giving equal weight to each available model projection is sub-optimal" (Eyring et al., 2019) versus "an emerging body of research suggests that an unweighted average of all simulator outputs often performs favourably by comparison with performance-based weighting schemes" (Chandler, 2013). The latter came after a
decade of literature considering different weighting schemes.

This confusion is most evident in approaches that take the ensemble mean and variance as the inferential best estimate and associated uncertainty. This choice presupposes that the ensemble members are random realizations of the physical system with independent, unbiased, additive errors. However for both climate and ice sheet contexts, simulator structural error will have dependence on the choice of simulator, invalidating this supposition. To make this clear, imagine if each MME member
were a trivial climate simulator that output a different time-independent random temperature field. The MME average would have no meaningful relationship to the actual climate, contrary to the logic underlying the use of MME means and variances.

As indicated above, many approaches to MMEs assign weights to different simulations according to some performance score against historical observations without accounting for structural discrepancy. As in our toy model example, mis-specified weights (and therefore mis-specified or ignored structural uncertainty) will result in potentially highly inaccurate predictions.
This has been borne out by explicit tests of performance-based weighting in climate simulation ensembles (*e.g.,* Deque and Somot, 2010) as well as via detailed analyses with idealized simulators (*e.g.,* Weigel et al., 2010).

The ice sheet modelling field has much less relevant literature on MMEs for paleo contexts. As one recent example, Batchelor et al. (2019) provide minimum and maximum extent time-slices for past Northern Hemispheric ice sheet evolution, in part via reliance on published numerical simulations. However, the set of chosen simulations do not include simulations de-



signed to bound uncertainties in past evolution. A similar problem occurs in the climate field, for which simulator contributions to the IPCC are generally "best-tuned" versions, even though the IPCC attempts to infer bounds on future climate change from this under-dispersive set of climate simulations.

The previous subsection on making predictions with models showed how a posterior inference for any observable quantity requires two structural discrepancy contributions: one ($P(D \mid C_M, M)$) to get the posterior for model parameters, and one for

the prediction ($P(S = s \mid C_M, D, M)$). Most published MME weighting schemes ignore structural uncertainty in the likelihood (and therefore for the constraint data). We know of no published MME analysis to date in ice sheet modelling (including the statistically advanced Edwards et al., 2021) that accounts for structural discrepancy in the final inferred quantity.

For those delving into MMEs, we recommend first the briefer but more accessible treatment (at least for most ice and climate modellers) of Williamson et al. (2013) followed by careful reading of Chandler (2013) and Rougier et al. (2013) which offer

practical and theoretically rigorous methods for drawing inferences from MMEs. For certain contexts, it may make sense to emulate the whole MME (treating it as a stochastic simulator as in Edwards et al., 2021).

### 2.10   Constraint data : availability and uncertainty specification

To date, based on the first author's experience with ice sheet modelling, assembling a relevant high-quality constraint data set with confident uncertainty specification has been a major challenge. Relatively recent (Briggs and Tarasov, 2013) and

ongoing efforts such as HolSea (https://www.holsea.org/) have improved the situation, but what is really needed is a one stop shop for complete quality controlled constraint databases that includes detailed uncertainty specification. To be effective and efficient, such a data site will require easy and flexible data submission, error checking, automated quality control, automatic age calibration using a choice of protocols, bulk retrieval, and automated generation of a reference list for subsequent citation. In the context of relative sea level (RSL), Dusterhus et al. (2015) provides an overview of the issues to consider for such

database design and implementation as well as a list of available RSL databases. Ghub (https://vhub.org/groups/ghub/paleo-datasets) is a hopeful new endeavour that aims to fulfill many of these criteria for past and present ice sheets. Efforts by relevant communities to spell out data collection and logging protocols (*e.g.,* Shennan et al., 2015) are also critical.

The ideal constraint data set will have narrow uncertainty and wide spatio-temporal coverage. It will also constrain diverse physical characteristics of the system. Consider the Greenland ice sheet. Present-day vertical velocities, relative sea level,

and cosmogenic dates constrain earth model parameters for glacial isostasy as well as deglacial ice extent and peripheral ice thickness. The present-day observed (or reanalyzed) melt rate fields provide a strong constraint on modelled melt. Observed deep ice core temperature profiles provide an integrated constraint on past temperature and precipitation for central regions of the ice sheet. The present-day observed ice thickness field also provides a strong dynamically integrated constraint for glacial cycle models.

As discussed above, the specification of the likelihood (or an alternative to be discussed below) requires specification of constraint data uncertainty. In terms of the paleo context, the sea level and C14 dating communities have arguably made the most progress in data uncertainty specification. Shennan et al. (2015) provides detailed guidance in assessing RSL proxy uncertainties, especially in consideration of the relationship between the proxy (eg a mollusc found at a certain elevation, embedded in





a specific stratigraphic framework, with a C14 date) and the "indicative meaning", *i.e.* the inferred probability distribution for the sea level associated with this datum. The dating communities are the most advanced with online applications that generate non-Gaussian probability distributions for calendar ages given a C14 or $^{10}$Be datum. The provision of a computationally concise and efficient representation of these distributions (*e.g.,* kernel density estimators) would facilitate their incorporation into paleo inference projects.

Error model specification for constraint data must account for the complete relationship between the model prediction and the datum. For the example of $^{10}$Be dating, in addition to the uncertainties provided by online calculators, one must also account for uncertainties arising from inheritance and in the case of samples from boulders, transport and subsequent possible disturbance. For the case of minimum limiting C14 dates from samples embedded in an end moraine, the inference of when an end moraine was formed should include a distribution for time of migration of relevant flora to the sample site. All data/proxy comparisons must also account for uncertainties arising from limited simulator grid resolution.

Accurate observational uncertainty assessment can be challenging for poor-quality data. In noisy data such as RSL paleo proxies, it is common for much of the data to have no constraint value beyond increasing the confidence in other data. For instance, most dated mollusc samples (as in Fig. 4) only indicate that sea level was above the relevant datum for a given sample (with allowance for downward uncertainty of order 1 to 10 meters for possible tidal changes and upward displacement of the mollusc by waves). As such, only the highest mollusc samples for a given time offer useful constraint. In the example of Fig. 4, the lower-bounding, pre-3 ka data-points below 30 masl have no constraint value and should be eliminated from the constraint data-set.

This example illustrates the value of database cleaning. Clear outliers and data that apriori provide no additional constraint should be removed from the calibration (but should still be retained for subsequent posterior checks). Outlier removal requires careful, transparent, and documented judgement with explicit consideration of the whole set of local data and relevant signal. It is best done by those most familiar with the data and outlier assessment should therefore be a part of database entry by the data-gatherer.

## 2.11  summary of what the Bayesian framework offers and challenges that ensue

The above provides a high-level framework for making meaningful inferences about the world around us, especially in the context of incomplete models used to describe a complex non-linear world. It is also a framework for how to improve these inferences. For instance, attention to structural discrepancy can guide efficient simulator development by making clear the largest sources of structural uncertainty for the given context.

However implementation of standard Bayesian inference for complex simulators is a challenging and potentially non-robust endeavour. Posterior distributions tend to be non-analytical for non-trivial simulators and therefore can only be sampled from. As described in our discussion of MCMC sampling, this sampling may take a long time to converge (perhaps beyond available computational resource limits). No matter how the sampling is carried out, the result can be highly sensitive to the exact specification of the error model and therefore the likelihood. This is especially the case for high dimensional constraint data and parameter spaces, such as is generally the case for complex geophysical simulators. Our toy example also illustrates how



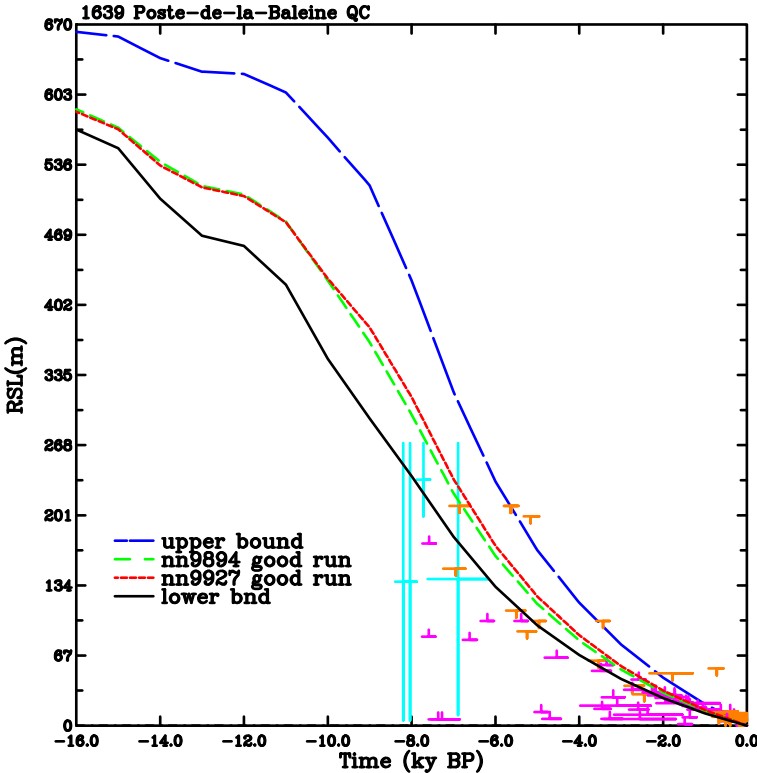

**Figure 4.** Comparison of modelled RSL histories and data for southwestern Hudson Bay Poste de la Baleine(Tarasov et al., 2012). To avoid clutter, 1-way error bars have been truncated. Orange 1-way (upper bound) error bars are generally for samples of terrestrial or mammalian origin, such as wood and bone. Purple 1-way (lower bound) error bars are generally non-intertidal species of molluscs that can live between shallows and depths of hundreds of meters.

inaccurate uncertainty accounting will lead to erroneous inference. For complex simulators, such error model specification will always be dependent on the uncertain specification of the external discrepancy. As such, inferences for say a most likely ice sheet history will have limited meaning contingent on a large set of assumptions. For those interested in a more complete and detailed accounting of the challenges involved, we refer the reader to the Frigg et al. (2015a) critique of the most descriptively detailed attempt at full Bayesian inference for future climate change to date (Sexton et al., 2011).

Unlike the norm to date, the Bayesian framework at least tends to force more of these assumptions to be explicitly stated and offers conceptual clarity. For many contexts, a more limited product than a rigorous posterior inference may have adequate utility and can be much more robust. Such a product could also act as a useful stepping-stone towards a complete posterior inference. We turn to this now.





## 3    Two stepping stones towards addressing uncertainty

Below we describe two stepping stones that meaningfully address uncertainty within a tractable framework. Instead of determining a likely non-robust posterior probability distribution, they collectively enable the identification of an initial set of
uncertainty bounds and a sample of simulations within those bounds. Such bounds help reframe the emphasis of the inferential process from finding the most likely system history to that of inferring credible bounds on said history. Both stepping stones also align with a major pre-occupation of many modellers: evaluating to what extent one's model has the capability to capture the system of interest to the requisite minimal accuracy, identifying the components most in need of improvement, and then using this evaluation to improve one's model.

### 3.1    The first stepping stone : History Matching

History matching is a Bayesian approach that sidesteps the challenging requirements of a well-defined likelihood function and complete sampling from the posterior. Instead, it assesses whether individual simulator runs are implausible as approximate representations of reality. It is much easier to identify clearly bad simulator runs than to assign a defensible likelihood to all simulator runs. Furthermore, history matching can reject a simulator by assessing that all possible simulator runs are implausi-
ble. A full Bayesian analysis is predicated on the existence of one true value of the simulator configuration and therefore can't directly reject a simulator (however, posterior diagnostic validation offers an indirect route to rejection, *c.f.* appendix A5).

History matching was originally successfully applied to the determination of parameter settings for computationally expensive geological oil reservoir models (Craig et al., 1995; Cumming and Goldstein, 2009). It has subsequently been applied to simulators in a variety scientific contexts, such as ice sheet models (McNeall et al., 2013, though more as a proof of concept
with limited metrics and only 5 ensemble parameters) and a general circulation climate model (Williamson et al., 2013). A complete application of history matching to a simulator for galaxy formation (Vernon et al., 2010) and to HIV transmission modelling (Andrianakis et al., 2017) in combination with an application of history matching to the NEMO ocean model that includes relevant code (Williamson et al., 2017) offers a clear presentation for those interested in implementation.

History matching involves a series of iterations ("waves") that, through an appropriate rejection criterion, incrementally
refines the subspace of not implausible parameter vectors. A vector is deemed implausible if the output of the simulator or emulator is sufficiently far away from the observed data given all relevant uncertainties. This requires the definition of a set of implausibility metrics which quantify data-model misfits (residuals) relative to specified approximate values for total uncertainty. The simplest implausibility measure is the ratio of the data-model residual to uncertainty specified as a standard deviation:

$$I = \frac{\text{model prediction} - \text{observed value}}{\sigma_{total}}. \tag{23}$$

Given that the square of total standard deviation for independent sources equals the sum of squared standard deviations for each source, it is more convenient to work with the square of the above equation. Furthermore, as adequate sampling of the parameter vectors will generally require emulators, "model prediction" in the above should be replaced by the mean emulator prediction for the simulator. Assuming respective emulator, structural, and observational standard deviations ($\sigma_{em}$, $\sigma_{struct}$, and



$\sigma_{obs}$) for datum $d_i$ at location and time $x$, the implausibility value for the emulator mean prediction (technically expectation, $E()$) of simulator output $E(M_i(x, cm))$ using parameter vector $cm$ then takes the form:

$$I_i^2(x, cm) = \frac{(E(M_i(x, cm)) - d_i(x))^2}{\sigma_{em}^2 + \sigma_{struct}^2 + \sigma_{obs}^2} \qquad (24)$$

A choice for an initial collective implausibility measure ($I_M$) that avoids dealing with correlations in the error model is the maximum implausibility over the set of chosen constraint data ($I_M^2 = \max_i(I_i^2)$). Simulator runs with $I_M$ above a rejection

threshold would be deemed implausible and ruled out. As $I_M$ would be very sensitive to inaccurate emulation of one single output for any simulation, it is generally advisable to set $I_M$ to the second or third largest implausibility $I_i^2(x, cm)$ for at least early waves.

A common $I_M$ rejection threshold is a value of 3 based on the 3 $\sigma$ rule (Pukelsheim, 1994) that at least 89% of a general probabilistic distribution should fall within 3 standard deviations of the mean. For any continuous unimodal distribution, this

rejection threshold is more palatable as 95% of the probability will then fall within 3 standard deviations of the mean according to the Vysochanskij-Petunin inequality (Vysochanskii D. F., 1980). If it can't be ruled out that the underlying distribution is multi-modal, then a threshold of 5 would cover 96% of the probability for any continuous probability distribution (of an integrable random variable) according to Chebyshev's inequality (Chebyshev, 1867). These theorems enable history matching to have a statistically rigorous and meaningful interpretation.

If the chosen threshold precludes an adequate number of not-implausible runs, then re-assessment of simulator configuration, inputs, choice of ensemble parameters, and/or simulator discrepancy distribution is required. This contrasts with a full Bayesian approach that requires the posterior probability distribution to integrate to 1 over all possible choices of the ensemble parameter vectors and therefore can't reject a chosen set of simulator, likelihood, and prior.

History matching is generally carried out with emulators, to ensure that the structure of the model response to the parameter

space is reasonably well resolved. In our topographic MCMC analogy, "well resolved" would mean that every valley has at least one sample, even though the minima are generally not identified. During each successive wave, additional points in the parameter space are ruled out as implausible, leading to an ongoing refinement of the remaining not-ruled-out-yet (NROY) subspace.

History matching is much simpler to credibly implement than a full application of Bayes rule. It features a much freer

selection of metrics and the lack of need for a multivariate error model and associated likelihood function, especially during initial waves. For instance, in calibrating a model for Greenland ice sheet deglaciation, one could initially separately apply RSL, present-day topographic, and ice temperature profile misfit metrics instead of a complete multivariate likelihood. A rigorous Bayesian inference would require the latter. As the NROY parameter space is narrowed through successive waves, internal discrepancy assessment can be carried out to refine the implausibility metric, including the addition of multivariate structure

(*i.e.* that accounts for error correlation between different constraints).

Another useful feature of history matching is that there is no need to include all metric components for the initial waves. Instead, it makes more sense to start only with metrics that use easier to emulate outputs of the simulator. Furthermore, the emulators only need to be accurate enough to significantly shrink the NROY parameter space during the given wave. As the





NROY parameter space is narrowed, accurate emulation becomes easier, especially for variables with spatial and/or temporal dependence. Further metric components can then be added (as demonstrated in *e.g.*, Vernon et al., 2010; Williamson et al., 2017). This improvement in emulator accuracy is due both to the narrowed range of simulator response and the increasing density of simulator runs in the NROY parameter space with each successive wave that is used to create the revised emulators.

History matching also offers an heuristic route to inferring an order of magnitude estimate of external structural discrepancy. Goldstein et al. (2013) provide an example of history matching using a fast rainfall runoff model with 17 ensemble parameters. In their case, out of 100,000 emulated model runs, they first select the best 8 that have the lowest implausibility when structural discrepancy is not counted (*i.e.* excluded from the denominator of equation 24). They then set the structural discrepancy for each metric component $i$ ($\sigma_{struct}$) so that the value of the full implausibility metric is just below the rejection criterion:

$$\max_{\text{over 8 models}} (I_i^2(x, cm, \sigma_{struct})) \leq 3. \tag{25}$$

Though crude, such an estimate for structural uncertainty can be used as a starting point to assess whether the current model configuration is of requisite accuracy for the given context.

History matching is a learning process about the relationship between the simulator and the system in question. One does not require a complete and accurate error model for the first wave, only an error model that doesn't underestimate structural and observational uncertainties to one's best judgement. To avoid such underestimation, one can go as far as specifying a large structural discrepancy such that any larger would lead one to judge the simulator as useless for the given context. As one progresses through the waves, the error model can be refined. This thereby enables both learning about the structural discrepancy and the impact thereof on simulator calibration and predictions.

A methodological inefficiency to avoid (provided you can find a reasonable upper bound) is to initially underestimate the structural uncertainty and subsequently require its expansion. This would require a complete repetition of all previous waves. Such an inefficiency is much more difficult to avoid in a full Bayesian analysis. Given this and the requirement of a single accurate multivariate emulator for all relevant outputs that a full Bayesian inference would entail, we strongly recommend initial history matching even for those seeking a complete posterior distribution.

Note that a low value of the implausibility does not imply a plausible or "good" parameter vector, only that the vector has not yet been ruled out. A low implausibility of a parameter vector can arise from large emulator variance and/or lack of data constraints in the current wave for which simulator fits would be poor. For those interested in working through an actual toy example of history matching, Vernon et al. (2018) provide a detailed, lucid presentation (along with a non-toy case study for a systems biology model) that includes the complete R script in the supplement.

## 3.2 The second stepping stone and structural discrepancy check: data-bracketing

A pragmatic criterion that classical history matching doesn't explicitly address is whether the simulator has enough degrees of freedom to fully bracket (*i.e.* from both above and below) critical constraint data within observational uncertainties given parametric degrees of freedom. The choice of what constitutes "critical data" will depend on context (including correspondence of constraint data to quantities of interest for prediction), simulator complexity, and simulator cost. An NROY ensemble



for which every member consistently overestimates present-day Greenland ice sheet ice thickness and ice area, for instance, would suggest some combination of too restrictive prior ranges for ensemble parameters and/or problems with inputs and/or some fundamental deficiency in the simulator. Such an ensemble would tend to have much less inferential value than one
whose members collectively bracket all constraint data, preferably with bracketing by just two bounding simulations for each qualitatively different subset of data (such as temperature and precipitation for a climate simulator).

For computationally expensive climate simulators, the presence of persistent biases would necessitate a more restricted and context-dependent choice of critical constraint data that the ensemble would need to bracket. It will also require careful selection of ensemble parameter vectors informed by understanding of simulator response to individual and joint parameter
variations. A persistent under-prediction of *e.g.,* monsoon intensity may be unavoidable, and acceptable for say the context of coupling with ice sheet models for which tropical features have no direct relevance (though this could be complicated by teleconnections). However, for a similar context, a climate simulator ensemble for which all models under-predict the observed mean seasonal cycle over Northern Europe (a critical control on ice sheet surface mass-balance) would tend to have much less inferential value than one that fully bracketed the observed cycle.

From a statistical perspective, such bracketing would not be directly considered. Instead, any persistent biases would properly be subsumed into the (combined internal and external) structural discrepancy error model used to defined the likelihood. Such an allowance for simulator bias could therefore also be added to the implausibility metric in history matching. However, in non-linear, spatially-temporally-coupled systems such as ice sheets and climate, large persistent biases are structural discrepancies that potentially make the resultant inferences of minimal value due to error amplification from positive feedbacks. This is
especially the case for inferences related to simulator predictions not closely associated with available constraint data which is generally the case for paleo contexts. If the lack of data-bracketing were due to large external discrepancy (as opposed to internal discrepancy), this would induce a further inferential challenge of adequately specifying a large external discrepancy.

It is important to note that data-bracketing should not be a simulator tuning or calibration target as it doesn't account for structural discrepancy. It is also unlikely to hold for a final NROY set (given the lack of accounting for structural discrepancy).
Instead, data-bracketing is a pragmatic early-stage modelling check on observational error models, parametric choices, and prior ranges for parameters. It may also force a reconsideration of whether the simulator structural discrepancy is acceptable for the given context.

### 3.3   An interim inferential reframing: bounding reality:

Given the previously described challenges and fragility of full Bayesian inference, we urge a shift from the common focus on
a best-guess chronology for the system under consideration to that of "bounding reality", *i.e.* determining credible upper and lower bounds on, for example, the glacial cycle evolution of an ice sheet. Even for very expensive simulators, such as general circulation climate models (GCMs), this would entail multiple simulator configurations designed to credibly offer some bound on key characteristics of the climate system that are of interest.

A full Bayesian calibration with a well-specified error model provides a posterior probability distribution for simulator
predictions and therefore such bounds. However, for at least the near future, most modelling projects are unlikely to invest the





resources for the required careful specification of the error model. Bounding reality offers a more accessible target. And history matching offers a tractable approach to reach this target.

An important part of bounding reality is ruling out that bounds are due to inadequate sampling of ensemble parameters, incomplete use of available data constraints, or inadequate structural discrepancy assessment. Ruling out of inadequate sampling requires use of validated emulators and an appropriate sampling scheme. The creation of relevant online community databases should help address data availability and assessment of the extent to which relevant paleo constraint data is used. Internal structural discrepancy assessment can be judged on the extent to which noise was introduced to all relevant processes and the appropriateness of the amplitude and structure of the introduced noise. The adequacy of external discrepancy assessment is more difficult to judge, though diagnostic checks described in subsequent sections can help.

A further check on the inferred bounds is from analysis of the contribution from data constraints and this ties in closely to data-bracketing. For instance, a number of studies using glaciological simulators for the last glacial cycle have inferred varying contributions to last glacial maximum sea level from Antarctica (*e.g.,* Whitehouse et al., 2012; Golledge et al., 2014). However, to date, only Briggs et al. (2014) and Albrecht et al. (2020) have explicitly demonstrated that their glaciological simulators are able to produce significantly larger Antarctic contributions that were ruled out (tentatively in the first case) by data-constraints. These are also the two such studies that have inferred the largest upper bound estimates for Antarctica contribution to last glacial maximum sea level.

## 4 Moving forward

We offer below some key steps that relevant communities can take so that inferences about past ice sheet and/or climate states have meaningful value.

### 4.1 Ensuring uncertainty is addressed in model-based studies

Over the last two decades there has been an increasing rate of publication based on computer-based simulations of past earth and climate system evolution. Yet very few offer any clear uncertainty assessment. As such, the relationship between simulator output and system state is unclear. This becomes especially challenging for the non-modeller faced with often opposing inferences from simulation-based studies.

One core trade-off is between ensemble size and model resolution. How does one weigh the conflicting inferences of say a 5 km grid resolution hand-tuned model of Antarctica equilibrated for last glacial maximum (*e.g.,* Golledge et al., 2012) versus a large (4000 member) ensemble of 40 km grid resolution last glacial cycle simulations with 31 ensemble parameters (Briggs et al., 2014)? More constructively, how do we intelligently synthesize the information from a handful of higher resolution simulations that lack uncertainty assessment with that provided by a much larger ensemble of lower resolution simulations that have been subject to some limited form of history matching and uncertainty assessment?

The above quandaries would strongly diminish if all model-based studies making inferences about states and state changes in past earth system evolution were required to have clear uncertainty assessments. Studies that focus on understanding the role





of physical processes (*e.g.,* through model sensitivity and/or feedback analyses) may not require data comparisons but should still explicitly consider to what extent their analyses are about the physical system versus just the simulator representation thereof.

A key step to towards encouraging rigorous uncertainty assessment about earth system evolution would be for modellers, editors, and reviewers to ensure the following questions are answered:

- What assumptions and approximations are made?

- What model parameters are adjusted and to what extent is the criteria used for their selection explicit and appropriate (*c.f.* appendix A3)?

- To what extent does the chosen set of constraint data capture what is available and relevant?

- To what extent have the associated uncertainties been explicitly accounted for? In detail:

  - What are the error models for the data and how have they been justified (*c.f.* Sect. 2.10)?

  - How is structural discrepancy assessed for the numerical simulator (*c.f.* Sect. 2.6)?

  - How has parametric uncertainty been addressed (*c.f.* Sect. 2.7 and Sect. 3.1)?

  - Are predictive/retrodictive uncertainties appropriately assessed (*c.f.* Sect. 2.8)?

  - How sensitive are the inferences to assumptions and approximations in the error model (*c.f.* appendix A4)?

  - Have relevant uncertainties been clearly communicated to the intended audiences (which usually will include those not familiar with the details of the model used)?

- Is an appropriate representative subset of the results of the higher likelihood or not-ruled-out-yet set of simulations provided from an open-access server?

As we've outlined in this survey, addressing all of the above are required if one is to make a principled approach to uncertainty assessment. Sufficiently addressing all points above is a major endeavour for any one or even a small group of researchers. Therefore a judgement has to be made and justified as to what are the most important aspects for the present context. At the very least, the answers to these questions should be explicitly provided in submissions that claim to make inferences about Earth system evolution to ensure that such inferences are scientifically interpretable.

## 4.2 Addressing uncertainty in data-based studies

As many data-oriented studies invoke models to relate measured quantities to inferred system characteristics, many of the considerations from the previous subsection are also relevant for them. Additional relevant questions for data-oriented studies include:

- Clear statement of assumptions in data interpretation.





- – Clearer specification of errors than just ±. Data providers need to consider a broad audience and detail uncertainties in the relation between proxy and inference. This should include an appropriate observational/indicative meaning error model for the data that is specified or cited (*c.f.* Sect. 2.10).

- – Emphasis on data quality. Though there is some trade-off, high quality data (*i.e.* with tight error bars and strong signal) is generally of much more value than a large quantity of low quality data (*c.f.* appendix A4). Data gatherers can also use the uncertainty maps from calibrated modelling studies to prioritize data collection and ensure that their efforts will offer the greatest constraint value for modelling (*c.f.* Sect. 4.5). Tier rating of data quality in databases would also facilitate intelligent data usage.

- – Deposition of data in centralized online databases. To ensure data-utilization, easy-to-use online data-servers are critical. Servers such as the world data centres (http://www.ncdc.noaa.gov/paleo/icgate.html) are a good example, but need to much more strongly encourage detailed uncertainty specifications, ease of conversion of data formats, and fully integrated age calibration tools (*c.f.* Sect. 2.10).

Those who gather, process, and make direct inferences from paleo proxies are most likely to have the clearest understanding
of associated uncertainties. Data gatherers generally have their own conceptual uncertainty models that are only starting to be fully elucidated in peer-reviewed literature. An important contribution would be for each proxy community to develop a consensus error model for their specific proxies. By "consensus", we mean an error model that incorporates an extended distribution of all scientifically credible assessments of relevant uncertainties.

Data assessment and use requires easy data access in public online databases. Some journals and funding agencies are already
implementing requirements to ensure such access. Perhaps the community can work towards entrenching this throughout the discipline. This requirement should not just apply upon publication, but also to unpublished data that has languished for say more than a PhD interval of 5 years since being gathered.

### 4.3 Improving computational accessibility : component-based history matching

A key component of the history matching paradigm is to first history match what is easy and/or efficient to do so, such as the
early wave choice of simulator outputs that are easy to emulate. This concept can be extended to history matching simulator components before history matching the whole simulator (Couvreux et al., 2021; Hourdin et al., 2021). When the component is chosen on a process basis, such process-based history matching can help ensure that the physical foundations of the simulator are maintained (Couvreux et al., 2021).

Component-based history matching is an obvious continuation of the dominant climate simulator tuning paradigm: sequen-
tially tune individual parameterizations and then larger components against relevant data where possible and then retune a few key parameters in the full simulator as needed to meet chosen constraints. But the effectiveness of the traditional tuning approach tends to be broken by non-linearities in the coupled system and over-tuning of individual components. More simply, the optimal parameter vector for a component on its own can easily give far from acceptable results once embedded in a complex





non-linear simulator. A concrete example of such tuning failure is arguably the apparently excessive equilibrium sensitivity in
the recent CMIP6 climate model intercomparison results (Zhu et al., 2020; Wang et al., 2021).

Component-based history-matching on the other hand, if done well, should eliminate the risk of over-tuning. As shown in
Hourdin et al. (2021), it could also make good simulator history matching much more accessible for computationally expensive
and complex geophysical simulators. However, the component-based approach still needs whole simulator history-matching
to address nonlinearities arising from inter-component feedbacks. But this will be strongly facilitated by the strong parameter
space reduction from the computationally much cheaper component-based history matching.

Component-based history matching raises a number of interesting questions for the context of this overview. How can
the emulators of the history-matched components be efficiently used to facilitate history matching of the coupled simulator
(or larger component of the simulator such as the ocean component of an earth systems simulator)? How can the structural
discrepancy assessment of each component be synthesized into the structural discrepancy assessment of the coupled simulator?
To what extent are internal discrepancies amplified due to feedbacks between the components? Or are such amplifications more
than offset by the parameter space restriction required for stability of the coupled simulator?

### 4.4    A potentially new level of constraint : fully coupled ice and climate modelling of glacial cycle intervals

As already evidenced in past climate model development, the coupling of previously separately forced dynamical earth sys-
tem components, such as atmospheric and ocean general circulation models, can make the coupled system more unstable in
response to mis-calibration and/or structural errors. Coupled systems, as such, can impose more self-constraint on simula-
tor parameters and structure. The paleo ice and climate modelling communities are moving to a similar juncture with efforts
towards fully coupled ice and climate modelling of extended glacial intervals including the complete last glacial cycle (*c.f.*
https://www.palmod.de/).

To date, paleo ice sheet modellers have made extensive use of the large uncertainties in past climate evolution to provide easy
tuning knobs for their glacial simulations. Fully coupled ice and climate models will have more limited tuning ranges as well
as a wide collective set of constraints from both ice sheet relevant proxies and climate proxies. It may well be a major challenge
for a fully coupled ice and climate model to just replicate the rapid last glacial inception inferred sea level lowering and the
even more rapid sea level rise of the last deglaciation (Grant et al., 2012). Such replication may provide a strong constraint
on model responses to radiative changes such as we are currently imposing on our planet. A tightening up of proxy-based
inferences for sea level changes and rates of change during the last glacial cycle would therefore be very beneficial for such
model constraint.

### 4.5    Treasure maps: using inferential uncertainty to prioritize data collection

The widely read book Sapiens (Harari, 2015) argues that the key reason for the success of the scientific revolution over the last
few hundred years was an underlying focus on searching for ignorance in the western scientific community. The quantification
of uncertainty in Bayesian inference enables clear delineation of what is unknown and what is in need of further constraint,
arguably consistent with a search for ignorance.



An example of quantifying this search for ignorance are maps of say 90% uncertainty ranges (or ranges from a NROY set in the context of history matching) perhaps in combination with space-time correlation maps. These individual and/or combined maps represent a "treasure map" of where new data is most needed to better constrain the system. A simple example is a map
of ensemble variance for Last Glacial Maximum ice thickness (*e.g.,* fig 14d in Briggs et al., 2014).

This treasure map concept can be further expanded by extracting sensitivity kernels that display the dependence of some potentially measurable or proxy-inferable quantity on some system characteristic of interest. Meltwater pulse fingerprints (*e.g.,* Mitrovica et al., 2011) and emergent constraints (*e.g.,* Hall et al., 2019) are existing examples of this.

## 4.6    A needed community test of past inference: a blind, noisy inference intercomparison

As we move towards more rigorous inference and uncertainty assessment, there is a growing need for an independent test of methodologies. Noisy twin tests are a standard test for data-model inference algorithms. They involve adding noise to the output of a (high-complexity) "truth" simulator which then acts as input into the inferential algorithm. However, when done in-house, it is difficult for the researcher to not carry over relevant prior information from the truth model. Furthermore, if the same simulator is used for both (which has generally been the case), then even with added noise, the test is unlikely to
adequately address structural discrepancy.

A complete test for the paleo ice sheet context would therefore involve the following. Firstly, the selection of an advanced, higher-order ice sheet model, coupled climate representation, and coupled 3D glacio-isostatic adjustment, with all components at a resolution and complexity above that currently employed for ensemble-based inference of past glacial cycle ice sheet evolution. The model would be run over the last glacial cycle, preferably in a configuration that is approximately consistent
with model paleo proxies given relevant uncertainties. It would be important to have the configuration approximately replicate inferred ice and climate system variance. Secondly, participants would only have access to the model output with subtracted structural bias and added observational noise in a form that reflects available proxy data. Thirdly, the selection of a representative ice sheet that minimizes the computational burden. The Eurasian and Greenland ice sheets are the two smallest paleo ice sheets with extensive marine and terrestrial components and therefore either one could be appropriate. The Greenland ice
sheet would add the significant challenge of testing appropriate ice sheet initialization and spin-up. Organizers would then compare inferential submissions against the actual chronology of the truth simulator less its structural discrepancy. Ideally, submissions would include inferences from multiple approaches and multiple error models using the same simulator to isolate the evaluation of the inferential approach and error model assessment.

The machine learning community has long relied on community-wide machine learning competitions and this is arguably
one reason why machine learning has achieved such large progress over the last two decades. This blind test would entail a major community effort, especially given what is involved in complete model calibration. A major benefit is that this could thereby be used to test and refine inferential strategies, algorithms and structural discrepancy models. The results would also inform the general community of the maximum extent to which inferences to date of past ice sheet evolution might correspond to their actual evolution.





## 4.7 a community methodology research and development agenda

This review is in part a challenge to the community and in part a statement of vision. Much is still needed to get to a point where inferential probability distributions or at least bounds are robust and make efficient use of available simulators and proxy data for paleo contexts. We've identified a number of key issues in the discussion above. They largely break down as follows:

– **Emulator development:** What relevant simulator outputs and internal discrepancies can be appropriately emulated, especially given computational costs of earth system simulators? To what extent can this apply to joint emulators for slow/fast versions of a simulator as well as structurally different simulators? For the latter case, how can emulators for different simulators be usefully combined? What are the pros and cons of different types of emulators (and/or combinations thereof) for the ice and climate modelling contexts? Can near turn-key toolkits be developed for easy application to different simulators? To what extent can emulators for climate models be used to drive paleo ice-sheet models (*e.g.,* building on the emulation example of Tran et al., 2019)?

– **Specification of observational uncertainties:** Examples of observational uncertainties that are not yet well specified include those due to: inheritance in cosmogenic dating samples, time required for plant migration into post-glacial terrain for $^{14}$C ka ages, changing tidal ranges in the context of relative sea level proxies, identification of non-analogue assemblages of foraminifera, and all the uncertainties that go into relating oxygen and deuterium isotope records to past climate. As none of the above uncertainties are likely to be well represented by a Gaussian distribution, the efficient representation of the uncertainties for likelihood application also needs consideration.

– **Specification of external discrepancy:** Well-formulated expert elicitation (*e.g.,* Sexton et al., 2019) of external discrepancy is needed to both inform its specification and help guide simulator development to most efficiently reduce structural uncertainty. Intelligent experimental design of future model intercomparison projects would also aid external discrepancy assessment.

– **Specification of internal discrepancy:** This includes the identification and extraction of the largest sources of internal discrepancy for individual and classes of simulators as well as the examination of the extent to which internal discrepancy varies between structurally-different simulators. There is a need to determine plausible noise structures to use for internal discrepancy experiments. Simulators also need to be configured to facilitate such noise injection.

– **Parameter space sampling:** The core challenge is sampling the apriori unknown and likely very small high posterior probability (or larger NROY) subspace of a large dimensional parameter space, the so-called "curse of dimensionality". This includes comparative testing of available sampling schemes for both history matching and full Bayesian inferential contexts. Such testing may inform the development of new schemes that are more efficient and/or reliable especially for exploiting parameter space reduction via restriction to active variables (*c.f.* A6).





– **Forecast/retrocast:** What are the pros/cons of using ensemble predictions, emulator predictions, and full Bayes? How can the design for the final history matching wave improve retrocast accuracy? How might forecast context affect stopping criteria for history matching waves?

– **Useful and tractable steps after history matching:** Given the challenges of a completely rigorous and robust Bayesian inference, the development and testing of suitable approximations might permit meaningful steps beyond history match-
ing.

– **Synthesis of results from different simulators and/or different simulator analyses:** Suggestions towards a systematic approach for this are presented in Goldstein and Rougier (2009) and Rougier et al. (2013).

Each of the above should be judged as to how they contribute towards the efficient use of the available set of simulators and computational resources to make meaningful inferences about past earth system evolution. Past experience with community
simulators and model intercomparison projects indicate the progress that can be achieved from community collaboration. The challenge of working towards meaningful inference about past earth system evolution is large enough to also strongly benefit from, if not require, similar or larger scales of collaboration.

## 5    Conclusions

This review started from the premise that a defining feature of any aspect of science which is concerned about making state-
ments about the real world is the rigorous quantification of uncertainties. Within the context of computational models, this claim can be easily supported if one recognizes that uncertainty assessment is simply the principled assessment of the relation of model results to the physical system. Without robust uncertainty estimation or, at the very least, a more limited mix of quantitative and qualitative assessment by the modeller, the reader has no basis to interpret the relevance of modelling results to the actual physical system. As our toy model illustration demonstrated above (*c.f.* Sect. 2.5), ignorance of structural uncertainties
will generally result in model predictions that do not intersect the physical system within computed prediction limits, even if Bayes Rule is used for the inference.

The challenges in robust uncertainty assessment for complex systems, such as the earth and climate system, imply that the paleo community needs to move away from relying on a single or a few "best fit" simulations. Instead, the development and analysis of a distribution of reconstructions that bound reality within scientifically meaningful uncertainties would offer a
considerable step forward. Though still requiring detailed uncertainty assessment, the focus on bounding permits one to safely err on the side of excessive uncertainty. The history matching framework described herein offers a tractable route to such inference.

This review is intended to encourage uncertainty assessment that is scientifically defensible within the paleo community and to sketch a picture of what it can look like. As should be clear, this is a highly non-trivial process. Many if not most modellers are
more interested in improving model process representation, understanding model sensitivities, and analyzing modelling results than working through the numerous statistical issues and algorithms required for such meaningful uncertainty assessment.





Collaboration with uncertainty quantification practitioners provides an efficient way forward. To be fruitful, such collaboration has to be from the ground up, and not just seen as an add-on near the end of a modelling project.

As discussed above, a key part of learning is the clear delineation of where theories, hypotheses, and models break down. Clear uncertainty assessment enables meaningful interpretation of inferences and the clear delineation of misfits between models and paleo data. It is these misfits that can guide both model improvement and the prioritization of future data collection. Arguably, there should therefore be as much published attention to simulator/data misfit as there is to simulator/data fits and predictions.

To conclude, for all modellers (in the general sense of modelling) making real world inferences about Earth system evolution, we reiterate some of Rougier (2007) guiding questions: What do your probabilities or inferred bounds represent? Why should a scientist believe your inferences about real world properties from simulation results or proxies more than that of someone else's? Relatedly, what are your observational and structural error models? How have they been justified? If these are not addressed, then a core question underlying many model-based studies has no answer: What does your modelling work actually contribute to our collective understanding about the physical world?

## Appendix A: Implementation and shortcuts

The first parts of this perspective have outlined the general framework and challenges in carrying out a complete Bayesian inference for past ice and climate evolution. Rigorously carrying out such an inference can be a large-scale multi-year project beyond the ambitions of most modellers.

However, as we've argued above, uncertainty assessment is a critical part of the scientific endeavour and needs to be a core part of the whole modelling process. Just as in model building and experimental design, the researcher will need to make defensible judgements about trade-offs between inferential rigour and computational ease. Below, we sketch out a simpler framework for inferring past earth system evolution based on history matching. This framework provides a credible uncertainty assessment as well as a stepping-stone to a more rigorous approach. We then selectively discuss implementation considerations for various components of the inferential process of relevance to both approximate and completely rigorous approaches.

All of this appendix, except for the next sub-appendix A1, can be skipped by those not interested in the actual implementation of simulator-based inferences of past ice/climate/earth system evolution.

### A1    An example minimal framework for inferring past ice and climate system evolution with uncertainty quantification

The following simplified framework offers an example approximate approach towards meaningfully quantifying past system evolution. Furthermore, it would also form the stepping stone towards a complete Bayesian inversion. The example numbers of model runs given below are for paleo ice sheet modelling contexts with approximately 30 to 50 ensemble parameters. For much more expensive paleoclimate modelling contexts, run numbers can likely be reduced, in most steps, by a factor of 4 to 10, in part with more attention put on efficient emulator development and experimental design. These numbers are based on the





authors' experience and judgements; and are therefore tentative. For the paleo ice sheet context, the framework is tractable and

largely implemented in on-going work by the first author and his research group. For the paleoclimate modelling context with computationally expensive near current generation general circulation climate models, the framework will need refinement for computational accessibility, but all steps will still generally be required to varying degrees.

1. **Assemble a team** who can provide relevant expertise on: a) interpretation of paleo data and associated uncertainties, b) emulator development and uncertainty quantification, and c) proficiency and understanding of the simulator, including

assessment of systemic simulator uncertainties and prior ranges for ensemble simulator parameters.

2. **Select and/or develop a simulator** relevant for the given context with both fast and computationally expensive higher quality configurations. The slow, higher-quality version of the simulator would include some combination of finer grid resolution and/or higher-order representation of relevant processes. The fast configuration should enable order one to ten thousand simulations. The exact number of runs will depend on the number of simulator parameters, choice of emulators,

and complexity of simulator response. If the fast simulator has adequate accuracy for the given context, a slow version is not needed and the implementation framework can be appropriately simplified.

Relevant criteria for selection and development of both fast and slow simulators include the following. a) Process proximity to paleo constraint data (*e.g.,* the use of relative sea level constraints requires a confident representation of glacio isostatic adjustment in the model system). b) Quality of the modelling, including : level of numerical validation, inclu-

sion of processes judged to play a significant role, and defensible choice of approximations to the dynamical equations. c) Ease/breadth of internal discrepancy assessment possible. For example, a model with multiple easy-to-implement options for basal drag (for an ice sheet model) or cloud representation (climate model) would enable a reduced estimate for contributions from harder-to-assess external discrepancy. d) Relatedness between fast and slow simulators to an extent that permits the fast simulator to be appropriately informative of the slow simulator. e) An adequate number of degrees

of freedom in the forcing components (such as climate forcing for a paleo ice sheet model or volcanic forcing for a paleo climate model) to capture uncertainty in past forcing evolution on a scale relevant for the context. All of the above criteria/choices likely involve trade-offs, with, for example, a poorer quality simulator presumably being faster and therefore enabling more simulations, some of which may be required to address more sources of internal discrepancy.

For at least the paleo ice sheet modelling context, there are existing simulators available that can largely meet the

above criteria with criteria (e) necessarily involving some trade-offs. This model selection step may require subsequent iterations to determine a fast simulator configuration (*e.g.,* choice of resolution, and process approximations) that can be sufficiently informative of the expensive slow simulator response in the steps below.

3. **Assemble paleo data**. Selection should emphasize constraint value and spatio-temporal coverage (*c.f.* Sect. 2.10). Identify a provisional hierarchy of the data (*c.f.* sub-appendix A2) for history matching waves. Top-level data should provide

wide spatio-temporal constraint. Consider what transformation of the data may be appropriate to facilitate implausibility metric specification ((*c.f.* sub-appendices A2 and A4). The choice of data hierarchy and transformation will need subse-



quent reconsideration according to which relevant simulator outputs can be accurately emulated during a given history matching wave.

Set aside a small hold-out subset of the data for assessment of the final NROY predictive/retrodictive confidence.

4. **Identify potential ensemble parameters and specify their prior ranges**. Start with listed simulator ensemble or tuning parameters. Identify the main sources of model uncertainty (such as basal drag and climate forcing in an ice sheet model). Introduce further ensemble parameters, as needed, to minimize the model uncertainties not covered by existing ensemble parameters (*c.f.* sub-appendix A3 below).

5. **Run an initial order 200 member ensemble of the fast model with a dispersed random sample** of parameter vectors from the prior ranges. Latin hypercube sampling (Urban and Fricker, 2010) is recommended as it ensures that the final set of vectors has values for every individual ensemble parameter well-spaced across its entire prior range. The ensemble should have variations in all simulator parameters that are not physically well constrained and that do not result in effectively duplicate simulator response.

6. **Examine whether the model can bracket critical data** (cf previous Sect. 3.2 for selection of "critical data"). Assess whether all top level data are collectively bounded within observational uncertainties by the ensemble. If not, reconsider: priors for the model parameters, deficiencies in inputs (especially degrees of freedom in the climate forcing for paleo ice sheet models), missing processes that could be explicitly incorporated, existing processes that could be better represented, the possibility of data processing and/or code errors, and the observational uncertainty error model (which specifies the relation of the datum to the model output, *c.f.* Sect. 2.10). If none of these solve the deficiency, consider if the slow simulator provides adequate bounding for the problematic critical data.

Ideally this bounding is surface-wise so that only a few ensemble members are required to collectively bound key data. However, it may be the case that a significant fraction of the key data can only be bounded point-wise, *i.e.* each datum requiring distinct ensemble members.

From past experience, this is one of the most time-consuming steps for model development. It also is an informative step in developing understanding of requisite minimal processes and inputs for the given modelling context as well as serving as a partial diagnostic check on specified parameter ranges, model configuration, and observational uncertainties.

7. **Increase initial simulator ensemble size** as needed for emulator development. For last glacial cycle ice sheet modelling with order 40 ensemble parameters, the first author has found that about one thousand fast simulator runs are more than adequate for configuration of Bayesian artificial neural network emulators of requisite accuracy for wave 1 history matching. Those using Regression Stochastic Process emulators (RSPE, *c.f.* sub-appendix A6) may find a few hundred simulations adequate and would start fast/slow simulator emulator development at this stage, for which case slow simulator runs will also be required. Such development would use the global structure of the fast simulator RSPE to inform the structure of the RSPE for the slow simulator. More details on emulator development are in sub-appendix A6.




8. **Provisionally assess external discrepancies** for the slow model (*c.f.* Sect. 2.6.2) and add these to the error model used in the implausibility metric. This will require informed judgement by one or more experts appropriately familiar with the model.

9. **Carry out initial internal discrepancy assessment**. Select two to three runs that best fit data constraints within provisional external discrepancies and observational uncertainties while being substantially distinct (sufficiently different parameters and features in the output). Generate a 50 member internal discrepancy noise sub-ensemble with the fast simulator for each parameter vector from the above runs. The noise should incorporate as many potential sources of internal discrepancy as feasible. Extract the resultant estimates for bias and variance components of internal discrepancy from the sub-ensembles (*c.f.* sub-appendix 2.6.1).

   This assessment should also consider how internal discrepancy from fast simulator experiment can be used to estimate internal discrepancy for the slow simulator. At least a few internal discrepancy simulations with the slow simulator will therefore be required. Inflate the assessed discrepancy by an ad hoc amount, *e.g.,* 10%, to account for the limited number of runs and reliance on the fast simulator.

10. **Refine simulator configuration if required**. Evaluate if the internal discrepancy is unacceptably large. If so, one will need to identify the specific problematic sources of internal discrepancy. To determine the strength of internal discrepancy sources, generate noise ensembles with a single noise source each (such as noise added to the climate forcing for paleo ice sheet modelling). Start with those sources judged likely to be the largest contributors to internal discrepancy. Use the two basis parameter vectors from the previous internal discrepancy step that provided the largest internal discrepancy estimates (*i.e.* variance of relevant simulator outputs across their respective noise sub-ensembles).

    Consider reducing the largest contributions by the addition of new ensemble parameters (*i.e.* not those already removed in the ensemble parameter selection step above). This can take both deterministic or stochastic forms, such as a coefficient controlling the addition of physically motivated precipitation anomalies in the climate forcing of a paleo ice sheet model in a correspondingly deterministic or stochastic fashion. However, ensemble parameter addition comes at the cost of repeating a number of the steps above.

11. **Construct a test ensemble for emulators**. Randomly sample 100 or more parameter vectors from the prior range and run these with the fast simulator. If using RSPEs, also create a small test set of slow model simulations.

12. **Construct a set of emulators from the initial ensemble and select those that validate on the test ensemble for history matching.** The validation should verify that the distribution of emulator-simulator residuals is consistent both with the distribution predicted by the emulators and with any relevant assumptions that were made in the structure of the emulators. Emulators that fail this test can be rebuilt/retested during later waves with a refined NROY space.

13. **Carry out an initial history matching wave with the emulators**. The error model for history matching should include observational, internal and external discrepancy, and emulator uncertainty components. Uncertainty components should err on the side of over-estimation of possible uncertainty.





14. **Refine NROY space with successive history matching waves**. Preferentially choose parameter vectors for simulator runs from the NROY space with high emulator uncertainty. However, a few parameter vectors with low emulator uncertainty can provide a diagnostic check on emulator uncertainty self-estimation.

15. **More carefully assess internal discrepancy**. From the set of model runs, select 8 more runs with low correlation between key metrics and between associated parameter vectors that otherwise have the closest fits to the constraint data within observational and structural uncertainties. Use the cumulative 10 parameter vectors (including the two from the original assessment) to reassess internal discrepancy.

    In addition to internal discrepancy ensemble variance, consider biases and correlations. If there are significant correlations, then implement a multivariate implausibility. Consider which aspects of internal discrepancy can be emulated or transferred into a stochastic component of the simulator.

16. **Construct an emulator for the slow simulator if not already done**. Carry out 50-100 runs with the slow simulator using a dispersed NROY sample of the same parameters vectors already used for the fast simulator ensembles. Use the joint set of matched parameter slow and fast simulator runs to construct an emulator for the slow model. This emulator may be built on top of the existing fast emulator with the addition of emulation to predict the difference between slow and fast simulator runs for the same parameter vector. Both practitioner experience and some experimentation will facilitate effective emulator development for the slow simulator.

17. **Carry out further history matching waves** with expanding constraint data sets and/or refined emulators built on the cumulative set of simulator runs. History matching will be complete when the whole chosen constraint data set is imposed via the implausibility metric and emulator uncertainty has stopped decreasing. The mix of fast/slow simulator runs will depend on the choice of emulator, computational costs, parameter vector dimension, and accuracy of the slow emulators.

18. **Reduce structural error as needed** (and as is possible within the timeline of the project): Assess retrodictive/predictive confidence on the hold-out test data set (*c.f.* sub-appendix A5). Refine the simulator and add ensemble parameters to address persistent unacceptable misfits. Such refinements may necessitate reconsideration and revision of the structural error model.

19. **The final NROY set of slow model runs, associated emulators, and associated structural uncertainties using the complete set of paleo data provides one's inferential estimate of past system evolution**. This is an inference consistent with the simulator and constraint data, subject to the complete error model. This error model itself represents an important inferential product for making predictions/retrodictions (*c.f.* Sect. 2.8). Though a rigorous posterior probability distribution can not be defined without significantly more effort, the distribution of implausibility metric values has interpretative value as long the distinction between low implausibility and high probability is clearly conveyed.

    Unless the NROY parameter space is very small, the set of simulations with this set will likely be an incomplete sample and therefore only provide a partial inference. As such, key predictions/retrodictions from the NROY set of simulations



should be compared against predictions/retrodictions by emulators run over the whole NROY parameter vector set. Predictions for which emulator uncertainties are large can be further constrained by careful selection of further simulations to reduce emulator uncertainties.

20. **Clearly state remaining misfits**. This is of value to data gatherers to re-evaluate relevant constraint data and/or prioritize future data. The modelling community will also benefit from associated prioritization of required simulator improvements.

After a set of history matching waves that has reduced the NROY space adequately to allow appropriately accurate emulation of relevant simulator outputs, this minimal framework can be expanded in various ways. This could entail the specification of a full multivariate covariance structure for use in a Bayes Linear forecasting/retrocasting approach (Craig et al., 2001) or in sampling from the posterior (Andrianakis et al., 2015). The latter could also be carried out with an approximate, simplified likelihood as discussed below. Whatever approach is chosen, a posterior diagnostic checking of the error model (*c.f.* sub-appendix A5), and verification of MCMC convergence (if employed) should also be done.

## A2    Selection of a hierarchy of constraint data

The initial phase of history matching will use a limited high-quality subset of the data of wide spatio-temporal constraint value for which model response is easy to emulate. For instance, a relative sea level curve from Hudson Bay near the centre of the Laurentide ice sheet will offer much more constraint and easier emulation than a deglaciation age from the northern periphery of the ice sheet. The latter would be quite sensitive to local ice load history and would only offer a spatially local constraint on past ice sheet evolution.

This hierarchical selection of constraint data can be guided by physical reasoning in conjunction with judgment of what are priority characteristics for the given context. Consider a climate model coupled to an ice sheet model for paleo contexts. First-order climate characteristics most relevant to ice sheet evolution would be mid- to high-latitude regional summer day-time temperatures (controlling surface melt) and yearly net snow accumulation. The strength of the seasonal cycle (summer minus winter temperatures) reflects sensitivity to insolation forcing and therefore would also be an important primary target for this context (assuming this sensitivity on seasonal scales has some relation to that on orbital timescales).

There are also rigorous options for hierarchical selection of data for simulator constraint. For example, Cumming and Wooff (2007) and Cumming and Goldstein (2009) describe a relatively straightforward approach (principal variables) based on analysis of ensemble model output of variables that correspond to available data. They step-wise select variables that are most correlated with the remaining not-yet-selected variables after eliminating the effects of correlations already accounted for by the expanding set of selected variables. This is akin to step-wise variable selection in linear regression.

## A3    Selection of ensemble parameters

The choice of ensemble parameters should attempt to take into consideration all the possible parameters in the model. Geophysical models will have a set of specified explicit parameters, but there are a host of effectively implicit parameters for processes




not explicitly parametrized or otherwise lacking appropriate parametric degrees of freedom. If one considers only the uncertainty in past climate over a glacial cycle, parametrized climate forcings for glacial cycle ice sheet models can easily add dozens of ensemble parameters. However most paleo ice sheet models use a handful or less of ensemble or tuning parameters for their climate forcing. For comparison, GCM earth system models generally have more than one hundred poorly constrained explicit parameters. Even for simpler earth system models of intermediate complexity such as LoveClim (Goosse et al., 2010), there are at least a few dozen explicitly listed tunable parameters (Shi et al., 2019).

Ideally, the modeller will start with a large set of possible parameters, and then rigorously select a minimal subset that adequately probes the possible response of the model. This selection should encompass all parametric uncertainties, while minimizing duplicate response. For paleo ice sheet models, in addition to those from parametrized processes in the model (surface mass balance, sub-shelf melt, calving, basal drag, glacio-isostatic adjustment, and especially climate), consideration should be given to parametrizing uncertainties in ice sheet initialization and boundary conditions (such as the deep geothermal heat flux for all ice sheets as well as subglacial sediment cover and bed topography for Greenland and Antarctica).

The parameter selection process partly falls under the model sensitivity analysis rubric which quantifies model response to parametric variations. There is a large literature on this topic for approaches that at least partially account for interactions between parameters (*e.g.,* Saltelli et al., 2008; Santner et al., 2003).

However an arguably more direct, informative, and efficient approach for parameter selection (at least when then emulator relies on a low order regression component to capture the global response of the simulator) is a step-wise parameter selection for linear regression with linear, quadratic, and lowest order interaction terms[12] (*e.g.,* as in Cumming and Goldstein, 2009; Andrianakis et al., 2017). The regression should be repeated for all critical simulator outputs (or statistics thereof) on an initial ensemble with parameter vectors from an orthogonal Latin Hypercube (*e.g.,* using the lhs R package) covering their full prior range.

## A4   Simplifying the error model: data transformation (dimensional reduction and aggregation) and data weighting

Simulators of ice sheet evolution or paleoclimate will generally have correlated data-model residuals over possible ice sheet and/or climate trajectories. For instance, a simulator with chronically delayed deglaciation over northern Greenland will tend to have increased regional RSL and deglaciation-age data-model residuals. These correlations may arise between residuals of the same constraint data type (*e.g.,* RSL or deglaciation age) but at different locations and/or times and between residuals of different data types. If this deglaciation delay was structural, then all these correlations would propagate into the structural discrepancy. As our above toy example has shown, accurate inference requires accurate accounting of structural uncertainty. Therefore, in general, structural discrepancy models need to account for this multivariate structure when sampling from a posterior.

The importance of accounting for such correlations for posterior inference can also be understood by considering the relative constraint value of different data. Consider, for instance, the relative sea level database for North America (Fig. A1). The largest spatial density of RSL data is over the Canadian High Arctic, a region covering only a few percent of the total LGM ice sheet

---

[12]The interaction terms should be intelligently selected if simulator computational expense precludes an adequate number of runs.





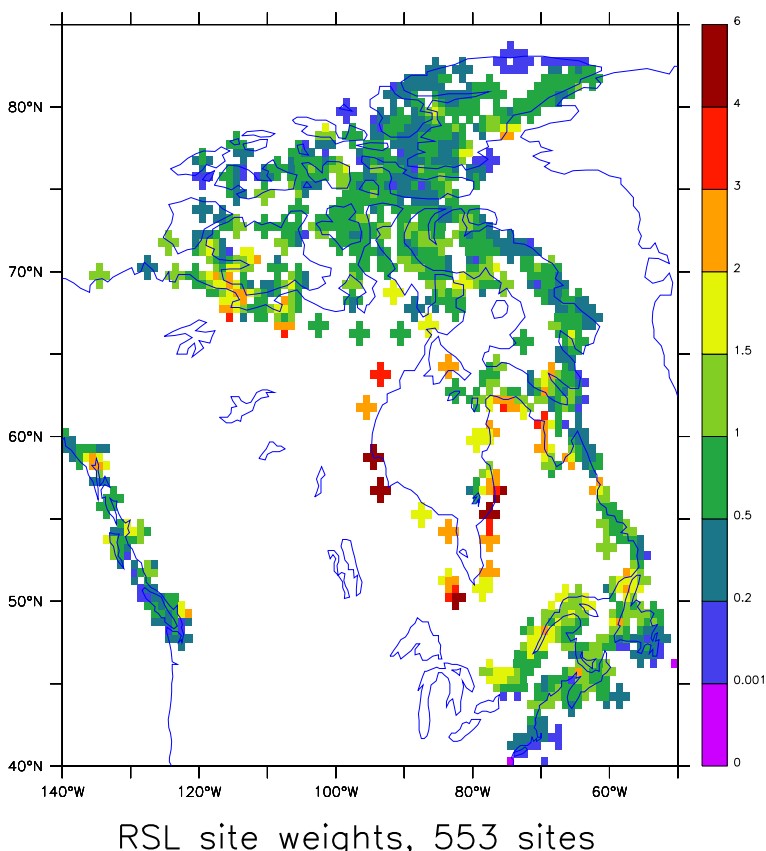

**Figure A1.** Specified site weights for a relative sea level database for North America().

extent. If each data-point were given equal weight and correlations were ignored, the much higher density of RSL data over the
Canadian High Arctic would skew any sampling from a posterior toward models with better fits to this relatively small region
at the likely cost of worse fits for the rest of the ice sheet. From a statistical point of view, equal weighting would imply that
one is treating all the data-points as statistically independent, and thereby ignoring the high correlation between data-model
residuals for RSL in this region arising from spatial proximity.

A multivariate likelihood function (such as a multivariate Gaussian) would account for all correlations between all available
data-model residuals. Such a function would thereby, for example, not let the high RSL data-point density in the High Arctic
skew a calibration. However, the robust assessment of multivariate structural discrepancy can be highly non-trivial and can
become computationally expensive for large constraint datasets.

For at least all but final waves, history matching can avoid much of these challenges by not requiring a fully specified
likelihood. This is because history matching only uses the error model in implausibility metrics for sample rejection (*c.f.*
sub-appendix 3.1) and not for posterior inference. The metrics need only be based on those features of the joint structural



discrepancy model that are easy to specify. Furthermore, especially for early waves, constraint data for history matching need only be a small highly informative (low uncertainty, high variation, and spatio-temporally dispersed) subset of what is available that is easy to emulate.

Even after database cleaning, for most geophysical contexts, the quantity of available constraint data can challenge likelihood
specification and inferential tractability. As such, dimensional reduction of the constraint data space will often be required.

The previously discussed use of principal variables (cf sub-appendix A2) is one option to consider for dimensional reduction. Principal components are another option that extract high-variance, dimensionally-reduced representations of the data. They are useful when there is extensive spatially-dependent data. They have been used to efficiently represent present-day grounding line position for a limited calibration of a glacial cycle model of the West Antarctic ice sheet (Chang et al., 2016). Sexton et al.
(2011) similarly used principal component reduction in the context of GCM climate model calibration. However, principal components can be noise sensitive due to a non-uniqueness problem for subsets of components corresponding to very similar eigenvalues. They can also complicate functional dependencies, resulting in larger uncertainties for the emulator predictions and increased difficulty in specifying external discrepancy. Salter et al. (2019) have shown that principal component reduction can lead to a mis-characterization of the NROY set (effectively history matching failure) when applied to the Canadian Climate
Model. They also provide a principal components rotation algorithm that can avoid this problem.

The community would benefit from an explicit comparison of options for dealing with large space-time dependent datasets, including: principal components, principal variables, and the addition of spatial dependence into emulators for different contexts. For instance, principal components are likely a good choice for global scale characteristics, especially when the components (or rotations thereof) are physically meaningful. Principal variables would tend to be better suited for characteristics with
regional dependencies and scope.

The effective data dimension can also be reduced by spatial and temporal aggregation. Consider the collection of data-points aggregated to site Poste de La Baleine in Fig. 4. Instead of having the likelihood directly a function of each data-point residual, the whole set of residuals could be replaced by a single RSL site score.

Unless all constraint data can be sensibly dimensionally reduced to a tractable size, both the computational costs of internal
discrepancy assessment and the challenge of emulator development for a high dimensional multivariate likelihood may lead modellers to adhoc shortcuts that weight the data by some measure of effective constraint value (*e.g.*, Briggs and Tarasov, 2013). Such weighting is a clear break from posterior assessment as the latter effectively measures simulation consistency with data given uncertainties and prior. This is logically distinct from measuring constraint value.

However a consistent synthesis is perhaps possible. The choice of constraint data for posterior inference or history matching
requires informed judgement. One criterion is the constraint value of the data and data-weighting can be use to created dimensionally reduced representations of constraint data. For the RSL example above, this could take the extreme form of weighting RSL scores for all sites on an ice sheet to create a single RSL score. For which cases this is more appropriate than previously discussed approaches such as principal components and principal variables is a research question yet to be addressed. A significant challenge will be the specification of the likelihood or implausibility as the observational error model will be convoluted
into the aggregate data score.



To date, data-weighting is a heuristic approach and should therefore be explicitly justified. This could involve demonstrating that the inferential results have limited sensitivity to an appropriate range of plausible assumptions in the weighting scheme. Much better would be explicitly comparing results against a multivariate error model for a regional subset of the output space derived from internal discrepancy experiments.

## A5    How to test structural discrepancy specification?

Structural discrepancy specification requires scientifically informed and defensible judgement. Although internal discrepancy can be much more rigorously assessed (via simulator experiments), it is still subject to, for instance, the choice of noise forcing used in the assessment. As such, error models are always provisional. Furthermore, at least some of the assumptions that go into error model specification are likely to be broken for most geophysical modelling contexts.

If one is aiming to infer a posterior probability distribution, it is therefore important to assess whether inconsistencies between resultant high probability (or low implausibility) data-simulator residuals and error model assumptions significantly impact inferences for the given context. History matching is much less sensitive to the detailed structure of specified structural discrepancy and especially in early waves, one only need ensure that structural discrepancy has not been under-specified. A empty NROY space (the whole parameter space ruled out) would be strong evidence of such under-specification. A very small NROY space may also be indicative of such under-specification, but it could also reflect inadequate sampling or very informative constraint data.

For the final history matching wave, a simple diagnostic check on error model specification is the comparison of NROY data-simulator residual distributions against the total error model. This comparison should be done for one simulation at a time over a small sub-ensemble that has the least implausibility score. If then proceeding to a full Bayesian determination of the posterior, one would examine residuals for a sample from the final high posterior probability subset. This check is potentially stronger if applied to a hold-out set of data not used in the history matching or posterior inference. For multivariate Gaussian error models, Bastos and OHagan (2009) offers a detailed example of such diagnostic checking.

The consistency between residuals and the error model is not a sufficient condition for error model validation. For instance, if the discrepancy between the observations + observational uncertainties and the blue linear (least squares) regression line in our toy model was chosen as the structural discrepancy, residuals would be consistent but the model would have much less predictive value than the optimal choice line after accounting for the biased structural uncertainty (light blue line in Fig. 2). This example reinforces the importance of making the selection of the structural error model (and therefore a component of the likelihood) independent of simulator fit to data. More fundamentally, the parametric form of the structural error and the prior for all simulator and error parameters needs to be set apriori, while parametric values of the structural error representation can be selected as part of the Bayesian inference.

Unlike history matching, full Bayesian inference assumes that there is a simulator configuration that is consistent with the likelihood, prior, and constraint data. It is therefore important to check for such consistency. Two relevant diagnostics for such checking are the prior (*e.g.,* Rougier, 2007) and posterior predictive checks (*e.g.,* Kruschke, 2013).



Few detailed examples of diagnostic error model assessments for geophysical contexts have been published. In the context
of a Bayesian calibration of a GCM, Sexton et al. (2011) and their followup Sexton and Murphy (2011) offer an example of
some relevant tests that can be carried out for checking the order of magnitude of the structural error estimate as well as
examining posterior sensitivity to assumptions in the error model. Schoups and Vrugt (2010) provide an example of how to
assess a posteriori validity of the error model for a hydrological context which has the further complication of auto-correlation.

## A6    Emulation

Emulators are approximate models of simulators that explicitly estimate the uncertainties in their predictions of simulator
output as a function of simulator inputs. For our purposes, they can be thought of as generalized regression models.

Emulators are commonly regression models with Gaussian process residuals (*e.g.*, OHagan, 2006; Rougier et al., 2009), or
their second order equivalents which we'll denote as regression stochastic process emulators (relaxing the Gaussian process to
a general second order stochastic process, *e.g.*, Goldstein and Huntley, 2017). Bayesian artificial neural networks (BANNs)
(Neal, 1996) can also be used (*e.g.*, Hauser et al., 2011; Tarasov et al., 2012). Given their greater familiarity and lower im-
plementation costs, pure linear regression models (such as the *lm*() function in R) can be efficiently used for initial waves
(or depending on the desired reduction in the non-implausible space, can be used for a full history matching exercise, *c.f.*
Ferreira et al., 2020).

The use of standard linear regression emulators has the major advantage of much wider familiarity within and outside of the
statistical community. For initial waves in history matching, Ferreira et al. (2020) have demonstrated for a high dimensional
(2136 outputs) synthetic case study of a geological water reservoir model that standard linear regression for second-order
polynomial emulators can be quite effective. They achieved a 99.5% reduction in the parameter space after two waves. At this
point, the reduction in the parameter space enabled adequate emulation of a multivariate objective function for the third wave
(with only an increase of emulator complexity to third order polynomials, still with standard linear regression).

Regression stochastic process emulators (RSPEs) typically contain three terms. Firstly, a "global" (*i.e.* across the parameter
space) term from linear regression with basis functions (such as polynomials) chosen with consideration of the expected
response surface of the simulator[13]. The second term is a stochastic process to characterize the residuals from the global term.
This is a noise generating function with explicit spatial dependence, as well as dependence on the simulator parameter vector.
This contrasts with linear regression that assumes residuals are independent and therefore will be especially more limited in
emulating the simulator response to small variations of parameter vectors.

The last "nugget" term in an RSPE represents the remaining sources of predictive error that do not show up in the residuals
as a local white noise. This relates to a key feature of emulators with a linear regression global component for history match-
ing contexts. Especially during initial waves, emulators of adequate accuracy for a given simulator target will tend to only
require a small subset of the whole set of simulator ensemble parameters as inputs. RSPE's account for the impact of dropped
parameters via the nugget. The active subset of inputs will vary between emulators for different targets. The potentially much
smaller dimension of the active parameter vector simplifies both emulator construction and parameter space sampling. For

---

[13]Our choice of "RSPE" is partly to emphasize the regression component



those interested in learning more about RSPEs, Vernon et al. (2018) includes a supplementary R script that reproduces RSPE construction and history matching for a simplified toy example. A complete history matching and emulation package using RSPEs has also recently freely been made available R (https://cran.rstudio.com/web/packages/hmer/index.html).

A key advantage of emulators with regression and stochastic process components is that emulators developed for fast simulators can strongly inform the development of emulators for expensive slow simulators. This thereby minimizes the number of expensive simulator runs required for developing the emulator for the slow simulator. Rougier et al. (2009) and Williamson et al. (2012) offer two relevant detailed examples of such fast/slow emulator construction in the context of climate modelling. Cumming and Goldstein (2009) provide a more detailed derivation for such joint construction.

Emulation intersects with machine learning. Machine learning tools generally build computational models that generalize a training set of input/output relationships, such as in speech recognition. They could therefore also apply to advanced ice sheet and climate models. Machine learning (useful reviews include Ghahramani, 2015; LeCun et al., 2015) has undergone an explosive advance in capability over the last few decades as evidenced by the quality of speech and image recognition and successful application to highly complicated games such as Go. However, aside from the development of BANNs (*e.g.,*

Neal, 1996), the machine learning community has to date put much less emphasis on uncertainty estimation. Therefore many machine learning tools cannot be used as emulators unless an added structure is imposed. A second challenge is that machine learning generally relies on large training data sets, which might not be computationally feasible with complex simulators. The non-linear formulation of most machine learning algorithms also calls into question their ability to extrapolate as effectively required for the high dimensional parameter spaces of complex geophysical models.

As for any regression tool, emulators will tend to more accurately predict quantities that have a smoother and more systematic response to input variables. This can result in trade-offs between direct emulation of fields of interest (eg the ice thickness field over time for an ice sheet simulator) versus emulation of smoother quantities that can be more easily compared to certain observations such as relative sea level.

     The development of emulators of requisite accuracy can be challenging and time-consuming, especially given a wide prior
range of ensemble parameters values. An important shortcut within the history matching framework is to identify a subset of data constraints for which the relevant simulator predictions are easier to emulate (eg scalar values with smooth and more linear dependence on inputs such as present-day ice volume or global mean temperature). Initial history matching waves should be carried out with this subset. The resultant reduced NROY parameter subspace (and thereby reduced dynamical range of the simulator) can facilitate the sequential development of emulators for more difficult-to-emulate simulator outputs in subsequent
waves.

     An informative calibration requires validation that the emulator-predicted uncertainties are consistent with actual simulator-emulator discrepancies. Given sufficient simulator runs, this is most easily done with a set of parameter vectors not used to create the emulators. For a single output emulator with Gaussian uncertainty, this could be done by comparing the standardized residuals (simulator - emulator output expectation all divided by the predicted standard deviation of the emulator) to a standard
Gaussian distribution. For expensive simulators, extra validation runs can be avoided with leave-one-out cross-validation.



Bastos and OHagan (2009) provides a more detailed examination of emulator diagnostics that account for correlation between residuals.

Emulators are not just used to represent deterministic simulators. Emulators can also emulate models that have stochastic components. When internal discrepancy has spatial, temporal, and/or ensemble parametric dependence, an emulator can there-
fore offer an efficient representation. Such emulation also may enable the introduction of ensemble parameters (*i.e.* thereby subject to history matching and/or posterior inference) to set the structure or amplitude of the stochastic noise used to assess the internal discrepancy.

*Author contributions.* LT did the writing but both authors contributed equally to intellectual content.

*Competing interests.* The authors have no competing interests.

*Acknowledgements.* We gratefully acknowledge the Durham Institute for Advanced Study that provided a fellowship to LT in 2012 and thereby enabled the start of a long discussion leading to this survey. The comprehensibility of this survey strongly benefitted from comments and suggestions from Fabrice Lambert and LT's research group, including: Matt Drew, Marilena Geng, Kevin Hank, Benoit Lecavalier, Ryan Love, and especially Heather Andres and April Dalton.





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
