# Peer review of "Assessing uncertainty in past ice and climate evolution : overview, stepping-stones, and challenges"

_EGUsphere, 2022_

## Referee Comment (RC1)

This paper attempts to introduce members of the paleo-modelling community to uncertainty quantification, in particular using Bayesian statistics. It is in part pedagogy, in part a review, and in part a series of suggestions and recommendations to modellers. I am a Bayesian statistician with extensive experience in the climate sciences and so will be reviewing this paper primarily from a statistical point of view. I agree with the authors on the need for a positional/translational piece of writing to serve as a guide for modellers – statistical analysis isn't easy, especially for those who do not necessarily have formal mathematical training. Despite the want for this paper, I found it long, dense and difficult to follow, largely containing opinions rather than evidence-based guidance, and ignorant of the existing statistical literature. This paper has been previously submitted for publication (a public process in Climate of the Past), and previously reviewed. I have reservations that many of the previous reviewers' comments have not been sufficiently addressed, and find myself agreeing with the previous reviewers on many points. For this review, I have detailed a number of major concerns (in no particular order) and defer any minor editorial concerns for a later submission.

1. **Length.** I struggled to get through the paper (and, in fact, was not able to without multiple sittings). 22 pages are spent introducing facets of the Bayesian framework after which there were a short(er) 4 pages on some useful techniques and then 7 pages of, from what I can tell, opinions with no examples. There are many points which are belaboured (such as the super-strong transponder throwing arm – which does not work as an analogy for me) and unfortunately sections are lost that are valuable to the intended audience (e.g. paragraph starting line 284). I would recommend drastically shortening the paper, particularly section 2, to only retain concepts key to the narrative, and cite the many existing introductory texts for the interested practitioner. For example, I don't think it is necessary to the main text of the paper to include 1.5 pages that build Bayes rule from conditional probability laws. Further, I don't see where MCMC is needed after it is introduced. MCMC is one of the most difficult early conceptual hurdles for the Bayesian statistics practitioner; introducing MCMC and letting it burden the mind when it is not needed is unnecessary. In fact, I struggle to see where any of the mathematics are used in Sections 3 and 4 of the paper. There are statements made in Section 4 such as *"…challenges and fragility of full Bayesian inference…"* where you go on to then recommend history matching, in effect, doing away with pages of statistics that you have tried to introduce. If we're just going to history match why do we need MCMC and posterior predictive distributions? In fact, why do we need Bayes rule at all? All we need is subjective probability and a mild understanding of uncertainty.

2. **Writing and grammar.** As a whole I find the paper to be quite sloppy. To name just some of my concerns, title cases are off, table formatting is inconsistent, there are erroneous parenthesis, the figure fonts are massive, and Tony O'Hagan needs an apostrophe. Some grammatical and editorial errors are inevitable, but I would expect a more thorough edit to be conducted before the submission of a journal article (especially before the second submission). I would also check your usage of colons throughout, the clause preceding the colon should be a complete sentence: see, for example, line 395. Also, the authors' colon spacing (with a space preceding the colon, and only sometimes) is foreign to me and I can't see a similar example in either US or UK style guides. Finally, the writing reads like a series of dot-points that

ramble on rather than with any real narrative or structure. I keep finding myself lost and have to remind myself of what is happening and where am I going. I find this concerning given that I am already familiar with most of the content and applications.

3. **Literature.** There is a lot of literature in statistics on calibration of computer models, prior selection and elicitation, emulation, modelling simulator discrepancies, as well as accessible introductory texts on Bayesian analysis. Not much of this is cited. Some of the more statistical texts are potentially inaccessible to the uninitiated; however, to ignore them does the paper a disservice (I am often surprised by what a motivated student can learn). I also feel the paper would benefit by acknowledging, in text, what applied work is being conducted, what is it doing well, and what is it missing (with appropriate citations). A dense table of citations is hard for the human brain to process, and despite my best efforts I still don't have an appreciation for what the authors are trying to say in the tables.

4. **Lack of examples.** This paper is quite dogmatic about the need to do a seemingly arbitrary sub-set of modelling stages but does not provide an example of it actually being done. Surely the authors have a more relevant toy simulator (that is not a linear model) that can be used to discuss the statistical concepts and also provide a helping hand to the struggling reader. In reference to point 2, adding an example that traverses the paper will add narrative and continuity.

5. **Feasibility.** I have a fundamental problem with the feasibility of some of the methods recommended. In my experience climate models are not computationally cheap to run, although I admit that the authors have more experience than me here and so could perhaps provide run times of certain simulations that are valuable to the community. In the internal discrepancy section the authors recommend that the parameter space is effectively explored and for each of these points the boundary conditions are sufficiently explored to accurately calculate potentially quite large variance covariance matrices. The authors admit that MCMC samples can take 10s of millions of runs, and then further we are required to predict from the simulator to obtain the posterior predictive distribution. The process of (1) exploring the parameter space in the order of millions of times, (2) exploring the spatio-temporal boundary conditions at each of these millions of locations, and (3) predicting and calibrating from these models seems computationally demanding in the extreme. Further, the boundary conditions are generated from *"adding appropriately correlated noise"* – in my personal experience this process is not simple and to cast it as such is wrong. Accurately representing, modelling and quantifying uncertainty for many spatio-temporal processes is exacting on even the most seasoned statistician and I think at least a nod to this should be included. Later in the paper, emulators are thrown into the mix with no explanation or introduction. Do your methodologies need emulators, and if so where and why?

6. **Point of the paper.** I feel the paper tries to do too many things, and so falls short on each of them. It attempts to provide textbook level mathematics on fundamental statistical principles, ground said mathematics in the application, and then provide recommendations and guidance. I know that the authors have tried to write a document that is accessible to a non-mathematical audience, but by taking the middle ground and then trying to teach mathematics it distracts from the rest of the paper. I would recommend removing as many equations as possible and instead

focusing on what the point of the mathematics is. The equations and the rigor can instead be deferred to a supplement for people to read once they've decided that it is worth the time and burden to learn and implement probability theory, MCMC, history matching, and the other number of techniques mentioned.

To reiterate, I agree with the need and usefulness for a document to teach fundamental uncertainty quantification techniques to a modelling audience. In my experience there is much appetite for the adoption of such methodologies in the modelling community. Both authors are senior and respected members of their respective scientific communities with invaluable experience in this space and so are in a good position to write such a document. Unfortunately, I do not think that the current version of this manuscript does a good enough job. I recommend a significant re-write that prioritises the point of the mathematics, and not equations for equations' sake; that has narrative and verve rather than 40+ pages of dense writing; and that leads by example with some demonstration of feasibility so that we are motivated to follow.

---

## Referee Comment (RC2)

**Bayesian analysis and paleo ice sheet modelling: a commentary on the proposal by Tarasov and Goldstein**

Evan J. Gowan

`evangowan@gmail.com`

**1 Overview**

Tarasov and Goldstein propose that paleo (specifically ice sheet) modellers and data practitioners should be making use of Bayesian analysis in order to ensure that everything has an error uncertainty attached to it. This paper is primarily a proposal, since all they are saying is that they want to see a united framework for uncertainty assessment. They do not present any new models or code, or provide a detailed literature review to back up what they want. They suggest that most ice sheet modelling exercises do not provide what they think is an adequate assessment of uncertainty, and that the only way forward is to run hundreds or thousands (or even millions – line 586) of model simulations to create an uncertainty range. In the first part of the paper, they go over what Bayesian inference is (using a strange and poorly explained example – line 141), and how it can be applied to create an assessment of uncertainty in models (and specifically paleo ice sheet modelling). The second part of the paper goes over the way that they would like to see a Bayesian framework applied to paleoclimate and ice sheet modelling. The third part of the paper lists off what the authors want to see in any paleoclimate data and modelling study.

Since 2005, I have been involved in the world of paleoclimate science. I have been involved in the collection of data for determining sea level proxies, compiling paleoclimate and sea level data into databases, ice sheet model development, ice sheet modelling, ice sheet reconstruction using glacial isostatic adjustment techniques, and paleoclimate modelling. I would say that I have experience in the entire gamut of paleoclimate sciences and therefore the target audience of the proposal at hand. Because I work on parallel topics to Dr. Tarasov, it should not be surprising that we have had a number of encounters. Dr. Tarasov has made it very clear to me (and probably many other paleoclimate scientists) that any study that does not involve a full Bayesian uncertainty assessment is not worth doing. This paper is an extension of his opinion on this matter. However, I doubt that this dense, jargon and equation filled paper that gives few workable examples will convert anyone. It is thoroughly unapproachable to anyone who is unfamiliar with advanced statistics (and maybe even to those with such training, judging by the other reviewer's comments).

I want to use this opportunity to provide an alternative view to what Tarasov and Goldstein are proposing, because I disagree with the idea that Bayesian uncertainty assessment is needed in every paleoclimate problem. In fact, as I will demonstrate, using Bayesian uncertainty assessment is probably a wasteful exercise in most paleoclimate and ice sheet modelling experiments. Before I start this, I want to make it clear that I have no problem with Tarasov and Goldstein applying Bayesian methods to their modeling experiments. The joy of science is the freedom of experimentation and exploring curiosity. It is for this very reason that I disagree with the entire premise of the proposal of this paper, as I believe it amounts to stifling creativity.

**2 Bayesian Analysis is already widely used in paleoclimate sciences**

The first point I will raise is that if you read this commentary, Tarasov and Goldstein make it sound like they are the only people who are using Bayesian analysis in paleoclimate applications. However, this is not true – tools that make use of Bayesian analysis are widely used in the paleoclimate community. Anyone who is calibrating radiocarbon dates or making age models for sediment cores use tools like OxCal (Bronk Ramsey, 2009) or MatCal (Lougheed and Obrochta, 2016). The reason these tools are widely used are because they are easily accessible, and can be run quickly on any computer. You rarely see any papers that do not make uses of tools like these for making age models. Therefore, the general premise of that Tarasov and Goldstein make, that paleoclimate scientists are not sufficiently aware of Bayesian analysis, is not true.

If Tarasov and Goldstein want the paleoclimate community to start using their proposed Bayesian uncertainty analysis technique, they could start by making their code and programs available. The Bayesian analysis tools used for radiocarbon dating and age modelling would not have caught on if they were restricted to the laboratories of the primary authors of those tools. I can speak from my own experience. I made my ice sheet reconstruction program, ICESHEET (Gowan et al., 2016a), available on Github, and several groups have used it with minimal to no involvement on my end. One group has even created a Bayesian analysis framework for it (Pollard et al., 2023)! The code for the Glacial Systems Model that Dr. Tarasov uses for his ice sheet modelling and Bayesian analysis exercises have never been made publicly available, despite the fact that the primary paper describing the application of it was published over a decade ago (Tarasov et al., 2012). In my opinion, it is arrogant for Tarasov and Goldstein to say with regards to paleo ice sheet modelling efforts "to date none have adequately addressed relevant uncertainties (Line 71)", when they have not contributed their own programs that, according to them, are the only way to evaluate model uncertainties.

**3 The usefulness of modelling**

Lewis (2017) presented an interesting treatise on the philosophy of climate modelling (the conclusions that also apply to ice sheet modelling). She asked the question "Is climate modelling really science?" A climate model represents an approximation of the real world with various simplifications and approximations to make something that is solvable on a computer. As such, can a test truly be made to demonstrate whether a model is right or wrong? In order to falsify an ice sheet model, it would be necessary to wait centuries to see if the predictions made by them are correct. Because of this, Lewis (2017) states that for practical reasons, a climate (or ice sheet) model fails Popper's test of falsifiability, and therefore is a form of "psudo-science".

Instead of restricting the definition of science in the restrictive bounds of falsifiability, Lewis (2017) instead suggests that we should define the science of modelling in terms of its usefulness, rather than whether or not it is strictly right or wrong (something that may not be possible to assess):

Climate models are useful because of the change they *themselves* introduce into the world. Within the climate science discipline, climate models push us introspectively to the limits of our understanding of the physical system. Extrospectively, climate models are powerful, flexible tools that are useful to society because they provide us with the capacity to pose problems and, ultimately, to act.

In the introduction, Tarasov and Goldstein argue that with the usage of Bayesian analysis that it is possible to "... go confidently well beyond storytelling but, as detailed below, only when all uncertainties are rigorously addressed and assessed" (line 37-38). But as highlighted above, this is impossible, because an ice sheet model is an approximation of the real world and will always have some level of uncertainty that will be impossible to statistically model. The usefulness of ice sheet modelling therefore comes not from being able to predict some geologically constrained reality, but rather from their utility in storytelling (that is, to pose problems) to help explain our world.

I can understand the appeal of the Bayesian ensemble modelling approach, as it is comforting to think that this will make up for the unknowns that are inherent with models. I will illustrate a couple of examples of why this is not an advisable approach for modelling paleo ice sheets.

**4    Past climate - the black box**

At the most fundamental level, the evolution of an ice sheet is dependent on the surface mass balance. If the amount of snow and ice that melts in the summer is less than the amount that accumulates in the winter, the ice sheet grows. Therefore, the climate forcing that is included in an ice sheet modelling experiment is the most important thing to get right.

In 2019, the ice sheet modelling group I was involved with published a study comparing the Paleoclimate Intercomparison Project 3 (PMIP3) Last Glacial Maximum (LGM) experiments (Niu et al., 2019). A number of climate modelling groups were involved with PMIP, and the generated precipitation and temperature fields for the LGM and the preindustrial period. Niu et al. (2019) interpolated these fields using an climate index from a Greenland temperature proxy record, in order to create a psudo-climate time series for the last glacial cycle (*i.e.* the last 120,000 years). This climate forcing was run in an ice sheet model covering the Northern Hemisphere. The resulting LGM ice sheets were then compared. The range of ice sheet geometries for the Northern Hemisphere ice sheets was vast. Some climate models produced massive ice sheets that far exceeded the LGM limit. Other models produced ice sheets that were very small compared to the geological evidence.

The topography and ice sheet boundary conditions of the LGM PMIP3 experiments were all the same. Yet the results were very divergent. Why? This exercise showed that the simplifications, parameterizations and resolution used in climate models are important factors in the outcome of these experiments. In the areas where the Northern Hemisphere ice sheets covered, there are no proxy records that can tell us which of the PMIP3 experiments is actually the closest to the truth.

The consequence of this problem in terms of a Bayesian calibration procedure that Tarasov and Goldstein

are proposing, is that the resulting error bars are going to be strongly biased to the climate model that they are using. This problem is likely so severe that I question the usefulness of this exercise.

**4.1   The glacial inception experiment**

In order to demonstrate this problem, I will highlight the recent paper from Dr. Tarasov's group, a Northern Hemisphere glacial inception experiment (Bahadory et al., 2021). The glacial inception is geologically constrained to have happened sometime between 120 and 110 ka. In this paper, they apply their Bayesian analysis scheme on a model that couples their Glacial Systems Model with a climate model of intermediate complexity. The results of their modelling ensemble shows a broad, continuous ice sheet linking Iceland, Greenland, the Canadian Arctic Archipelago, the Cordillera and Alaska, and variable amounts of ice in Europe.

There is a big problem with this though – many of the areas that their ensemble consistently shows to be ice covered are very well constrained to have not been ice covered during the glacial inception period. First, a large part of Alaska has never been glaciated (Kaufman and Manley, 2004). Second, the Laurentide and Cordillera Ice Sheet did not merge during the glacial inception period, as it is well constrained by geological observations that this only happened once – at the LGM (Duk-Rodkin et al., 1996; Trommelen and Levson, 2008; Jackson et al., 2011). Banks Island, located in the western Canadian Arctic Archipelago, likewise was only glaciated once, during the LGM (England et al., 2009).

The severity of this mismatch means that the true ice sheet volume during this period will fall far outside of the range of the uncertainty produced by the ensemble. The amount of ice produced in these simulations, even if those areas are not included, is likely to be excessive, as far-field sea level probably remained above present until 116 ka (Clark et al., 2020). This points to problems in the climate model – it is too cold in the summer and/or there is too much snow in the winter in high latitudes. At the end of the paper, they state:

> We especially hope that the field data community will use this archive to test, refute, and/or validate which, if any, of the model-derived LGI [last glacial inception] trajectories (and characteristics thereof) are consistent with the paleo record.

However, considering the major problems with the climate forcing, I can't see how these model results will be useful for the geologists investigating the glacial inception. In their current state, ice sheet models do not have the predictive power to precisely reconstruct ice sheet history. They cannot be used as a "treasure map" (line 999). The storyline of this paper should have been that it demonstrated that it is possible to grow an ice sheet rapidly after inception started, and that the Eurasian and North American ice sheets have different sensitivities to external forcing. That, to me, would be a more useful application of this modelling exercise, as it tells us something interesting about the climate system without overinterpreting the results in terms of how the ice sheets incepted.

**5 The Latin Hypercube Shotgun**

The Bayesian analysis technique generally uses Latin hypercube sampling in order to select the values of the parameters that are varied in the modelling experiments (lines 1164-1168). Latin hypercube sampling selects random values for all of the given parameters in the modelling study, such that the parameter space is well sampled given the number of ensemble members that are in the experiment. In essence, it is like shooting a shotgun at a target from a distance – some of the values will undoubtedly hit the bullseye, but others will miss the target completely.

A few years ago, I reviewed a paper by Gandy et al. (2021), which explored the deglaciation of the North Sea sector of the Eurasian Ice Sheet using a dynamic ice sheet model. The story they told in their paper was compelling, and I felt it was easy to suggest publication. In their study, they ran an ensemble of 70 simulations using seven variables. The values of these variables were selected via Latin hypercube sampling. The main criticism I had was that in varying the parameters using Latin hypercube sampling, it was not possible to say what the relative influence of each parameter on the outcome of the simulation. As a result, they stated:

> There is no clear distinction between the parameter values of the NROY [not yet ruled out] simulations and those of the rest of the ensemble, indeed the range of values for each of the seven parameters is almost the same in the full ensemble as within the subset of NROY simulations. This is partly because all parameters are varied in tandem and the parameter effects can compensate each other.

This presents what I consider the biggest weakness of large parameter studies that are used for Bayesian analysis. In a large ensemble with many parameters varied in tandem, it is not possible to clearly tell why one simulation fails while others succeed. This makes storytelling difficult, and reduces the utility of the model to tell us something about the behavior of the ice sheet. It is much easier to tell a story with an ice sheet model by holding most variables as constant, and varying a small number of variables in a controlled way. In that way, we know exactly how sensitive the outcome of the experiment is to a parameter.

**6 In modelling, there is always subjectivity**

The main appeal of applying the Bayesian analysis approach for ice sheet modelling is that it gives the illusion of objectivity by assigning a probability value to every decision (section 2.1). Since the experiment allows the parameters for each ensemble member to be selected at random from the experimenter's probability range, it removes responsibility of the outcomes from the experimenter. For practical reasons, there is always going to be a certain limits to how wide of a range of values that can be used in a Bayesian analysis modelling study. I will use the North American deglacial study by Tarasov et al. (2012) as an example.

In Tarasov et al. (2012), they varied 39 different components of the ice sheet model, with parameters

related to climate forcing, bed conditions and floating ice calving. After calibration, they ran 50,000 simulations, which were assessed based on how well they were able to fit observations of past sea level change, proglacial lake strandline tilts, and present day uplift rates. The modelled versions of these parameters come from a glacial isostatic adjustment (GIA) model. This model approximates the Earth as a series of spherical shells of uniform rheology. The typical Earth model used in this kind of analysis is one with an outer shell with elastic rheology (i.e. the lithosphere), and two or more shells representing the mantle, which behaves as a viscoelastic material at the time scales that ice sheets operate on. The response of the Earth to the time varying ice load is calculated by transforming the load history into the spectral domain, as the solution to the response of a viscoelastic material to loading is much easier than in the time domain. The weakness of this model is that the Earth is not a series of uniform spherical shells, and the thickness of the lithosphere and mantle viscosity are spatially variable. A 3D Earth rheology model takes at least a couple of orders of magnitude more time to solve, meaning that this is not a practical way to run a large ensemble of ice sheet model simulations. The viscosity of the lower mantle, which is an important parameter for the uplift rate in the center of the Laurentide Ice Sheet, is also not well constrained.

Tarasov et al. (2012) settled on a single Earth model in all of their simulations. This is not necessarily a bad thing. As mentioned above, it can allow for a better understanding of how the variables that are actually varied impact the ice sheet trajectories. The end product of this paper was GLAC-1D, which is an average of 10,000 of the best performing ensemble members, and an uncertainty on ice thickness. However, since the scoring of how "good" the simulation did is strongly dependent on the choice of Earth model, the average and uncertainty range is biased to their subjective choice of a single Earth model. In the paper, there are hints that the Earth model they chose might not be the best choice:

> Especially disconcerting is the weak fit to the data-rich southeast Hudson Bay sites. ... We have been unable to create a model that can dynamically (*i.e.* without ad-hoc brute force reduction of ice) produce thin enough ice over the Hudson Bay region to fit the local RSL record

So in the end, the result of the study produces a mean and uncertainty range for the deglacial of the North American ice sheets, but there is no way to know how precise this mean is to reality because it strongly dependent on their subjective choice of Earth model. Tarasov et al. (2012) acknowledge this problem, but still insist that the uncertainty ranges they create are meaningful.

**7    Using the right tool for the job**

One of the best GIA constraints for the center of the Laurentide Ice Sheet are the strandlines of proglacial Lake Agassiz. As the Laurentide ice sheet retreated, it opened up a vast depressed basin that meltwater could collect in, forming the lake as the ice blocked the northward drainage routes. The preserved strandlines and beaches formed by Lake Agassiz stretch for hundreds of kilometers. By measuring their present day elevation, it is possible to determine the relative amount of GIA induced uplift along the length of the strandline since its formation, and use it to constrain the ice load history.

The importance of this constraint means that it is commonly used for evaluating ice sheet reconstructions, and has been used by both myself (Gowan et al., 2016b) and Dr. Tarasov (Tarasov et al., 2012). Our approach to ice sheet reconstruction is very different, though. A dynamic ice sheet model, though in theory should be more "realistic", can never hope to precisely model the geometry of Lake Agassiz, even with 50,000 model simulations (Tarasov et al., 2012):

> Given the partially lobate structure of the geologically inferred ice margin, as well as the high sensitivity of ice margin location to what will invariably be a poorly constrained climate forcing, it is unlikely that any glacial systems model will ever freely approach inferred margin chronologies to the degree required for accurate modeling of proglacial lakes (required for strandline predictions) and surface drainage.

In my methodology, I use an ice sheet model with perfectly plastic ice and assumes the ice sheet is equilibrium (Gowan et al., 2016a). This model requires just three inputs – the ice margin at a specified time slice, a model of the basal shear stress, and GIA deformed topography. Though this is a very rudimentary model of an ice sheet (for instance, the ice sheet was likely never in equilibrium), it has the advantage in that the ice margin can be precisely defined so that the geometry of the lake can be reproduced to match the geological observations. When my reconstruction was tested with a lake filling algorithm, it successfully captured the geometry Lake Agassiz during the periods that had strandline data (Hinck et al., 2020). Only a couple of dozens of iterations were needed to tune the ice sheet reconstruction to fit the strandline data (in combination with many other GIA constraints).

When designing a modelling study, it is important to use the right tool to realize the goal of your modelling exercise. If you are interested in finding the geometry of Lake Agassiz to make estimates of its volume to say, figure out how much water could have potentially drained out to disrupt Atlantic circulation at the start of the Younger Drays (Broecker et al., 1989), a dynamic ice sheet model would not be an appropriate choice, no matter how many ensemble members are used. The climate forcing and/or calving in the dynamic ice sheet model would have to be manually manipulated to do it, defeating the purpose of the Bayesian framework that Tarasov and Goldstein are proposing. It is better to use a model where the margin location is strictly defined.

**8 The data are never perfect**

Paleoclimate and ice sheet reconstruction generally requires some kind of geological data for validation. For paleo ice sheet reconstruction, this usually comes in the form of records such as past sea level change, dated morainal features or ice flow direction indicators.

GIA based ice sheet reconstructions generally are judged based on paleo sea level data. Most available paleo sea level data from the areas covered by Late Pleistocene ice sheets are what I would call legacy data. These data were collected over 30 years ago, mostly through reconnaissance style surveys that prioritized collecting data from a wide area for constraining the ice sheet retreat history rather than detailed stratigraphic studies for constraining sea level position. The constraints on age come from

liquid scintillation measured (conventional) radiocarbon dates, which generally were a composite age of dozens of mollusc shells. This means the potential for contamination is high. In most legacy studies, the elevation is stated with no explanation on how it was measured or determined, and without any reported uncertainty. For places not close to the coast, the uncertainties on elevation could exceed 10 m. In most cases, the indicative meaning concept (Shennan, 1982) cannot be applied, so the data represent minimum limiting constraints on sea level. Ross et al. (2012) and Woodroffe et al. (2014) showed that the legacy data should be treated with caution. It is unlikely that governments are going to allocate millions of dollars to resurvey using AMS radiocarbon dating and differential GPS elevation measurements, so we have to accept this.

For these reasons, I tend to judge the fit of a the data using a simple "consistent or not" metric, because developing an framework for inclusion in a more complicated statistical model is not clear. The uncertainties of the age of the legacy dates are likely larger (potentially a lot larger) than the reported laboratory error, and the uncertainty range for a past sea level position cannot really be confidently determined when details are not provided. So, instead, I think it is fine to assess the models in a less rigourous (and definitely less computationally intensive) way rather than attempting to create some probability distribution that is going to be hand-wavey. It also removes the need for database cleaning that might remove data points that are more accurate (line 705-715). The "consistent or not" assessment has served me well in evaluating my models. A more complex statistical model is not needed.

This problem becomes worse if you look at periods older than the LGM. One could easily make the argument that any radiocarbon date that has an age over 30,000 years should be considered suspect, and other dating methods tend to have uncertainties of thousands or even tens of thousands of years. For ice sheet reconstruction, you also have to start using undatable features like flow direction indicators to reconstruct the ice sheet, and infer the history though logic alone in a conceptual forward model. This is what I did in my most recent reconstruction (Gowan et al., 2021). I fully acknowledge that such a reconstruction might be wildly off in terms of the timing, but based on what I know about ice sheet dynamics and the patchy framework of pre-LGM geological markers of the ice sheet, I am confident that the general history of the ice sheet is correct. Since the climate forcing used in ice sheet modelling exercises tend to cause the ice sheet to grow by accumulation over broad regions rather than through ice flow from an accumulation center (a concept that fits geological observations better), these kind of observations likely cannot be used for assessment in the framework that Tarasov and Goldstein propose at present time.

**9 Are error bars of paleo ice sheet simulation ensembles useful?**

Tarasov and Goldstein state (lines 35-38):

> The exponential growth of accessible computer power over the last few decades has permitted the ongoing development of a synthesis of the above two approaches: inference based on rigorously combining computational modelling with paleo observations. This potentially offers detailed pictures of ice sheet and climate system evolution that can go confidently well beyond storytelling but, as detailed below, only when all uncertainties are rigorously addressed

and assessed.

This sounds great, but the question I have is, are the error bars generated from these assessments useful for paleo scientists? Do people pay attention to them?

The main product of Dr. Tarasov's modelling that people use is GLAC-1D. This is an amalgamation of his lab's modelling exercises for several of the ice sheets, where they run thousands of ice sheet model simulations by varying a bunch of parameters after reducing the range through a calibration step. GLAC-1D is an average of a subset of those simulations that performed well against the chosen evaluation metrics, along with an ice thickness uncertainty range.

The primary usage of GLAC-1D is as a boundary condition for climate modelling intercomparison experiments (Kageyama et al., 2017). Although it is stated in this paper that GLAC-1D is an ice sheet reconstruction, that is not strictly true. The ensemble average will not fit any particular observation that the modelling exercise used to evaluate the individual simulations, and it is not glaciologically consistent. There is no guarantee that the ice sheets looked anything like the ensemble average. Still, it has been demonstrated that GLAC-1D is useful, because the climate modelling community does not see this as a significant problem. The key point is that the error ranges presented in GLAC-1D are not being used. No one is performing climate modelling experiments using the minimum and maximum ranges. This is largely because running climate modelling experiments are expensive (a single experiment may take several months to run), and they cannot do large ensembles. So, if the error ranges are not being used, is it really necessary to go through thousands of simulations to create one? I'd argue that it would have been better to use this exercise to pick a few simulations that performed well, and make those available. That would give the climate modelling community glaciologically plausible ice sheet reconstructions. It would make it very explicit that there is a range of possibilities for ice sheet configuration and give them an idea of the strengths and weaknesses of each reconstruction.

Tarasov and Goldstein complain about this (lines 62-69), but I think it would be more productive to be attentive to the realities and needs of other modelling groups rather than lecturing them about the need for ensemble studies to produce an uncertainty range.

**10    Telling a story**

Tarasov and Goldstein are of the opinion that storytelling alone is not an adequate way to use ice sheet models (lines 38-48). I would like to highlight one recent study that shows that this is not the case. Lofverstrom et al. (2022) performed two ice sheet/climate modelling experiments to simulate the conditions at the glacial inception period. In one experiment, they closed off the straits of the Canadian Arctic Archipelago, and in the other they forced them to remain open. The experiments showed that closing the straits was a sufficient condition to initiate glaciation in far away Scandinavia. I was one of the reviewers of this paper, and my main comment was that the current (though sparse) geological evidence points towards these straits becoming blocked at a later time, and not at the glacial inception. The story of this study, however, was very compelling and interesting, and I was happy to see it published as a great example of a well designed modelling experiment. Their apples to apples comparison tells

us something about the behavior of the climate system, even if it does not necessarily match a specific geological constrained reality. It was not necessary to do more than two simulations to tell that story.

Tarasov and Goldstein state (line 1089-1091):

> Many if not most modellers are more interested in improving model process representation, understanding model sensitivities, and analyzing modelling results than working through the numerous statistical issues and algorithms required for such meaningful uncertainty assessment.

There is a good reason for that, and it is because it is much easier to understand the results of a model and tell an interesting story if the model is used in a controlled way. Most modellers fully understand the limits of what their models can predict. There are limits to how much an advanced statistical framework can produce new insights when there are still so many processes that we do not fully understand.

**11    Ethical modelling**

**11.1    Carbon Footprint**

In Gowan et al. (2023), I ran three glacial cycle ice sheet model simulations of the Laurentide Ice Sheet and a number of idealized simulations to test a basal conditions module that I programmed. Due to the discovery of bugs, I had to run these simulations a few times. Each glacial cycle simulation took roughly one week to complete on 144 processors, while the idealized simulations too between one and two days. These simulations likely represented a sizable fraction of my annual carbon footprint, and I hesitated to consider doing more simulations, especially since the simulations I performed demonstrated what I wanted to show.

As the cost of computers has decreased over time, there has been a tendency to create more complex models that require more processors and longer computational times. Complex models are desirable because they are more likely to capture small scale processes that may govern the trajectory of a simulation. However, complex models that run for a long time on hundreds or thousands of processors also use a lot of energy. Considering that one of the motivations of a climate and ice sheet modeller is to warn of the dangers of anthropogenic climate change, we should consider the impact of our simulations on the environment. We should be thoughtful about the impact of our research and not to run more simulations than what are needed to tell our story. Imagine running thousands of model experiments and finding a bug that invalidates the results! This is not as severe of a problem if the experiments are chosen wisely.

**11.2    The realities of modern science funding**

At line 1127, Tarasov and Goldstein propose the first step of realizing their goal is to "assemble the team". In their ideal world, a modelling group would include people who are knowledgeable about the

data that is used to assess model results, people who develop the statistical models, and someone who runs and understands the ice sheet model.

The reality is that modelling research groups are typically led by one or two permanently employed principle investigators who handle the logistics for the group, and a team of tenuously employed students and postdoctoral researchers who do model the developement, run the experiments and interpret the results. The funding cycles for projects typically last between one and five years (three years is pretty common). Is it practical to develop a sophisticated statistical and modelling framework under these conditions? Do they think that a student or postdoctoral researcher can afford to wait months or years to get enough results to publish something? This is just not feasible under the current funding regime of science. We can only design modelling exercises that can be accomplished under short timeframes. It would be unfair to students and postdoctoral researchers to be forced into a narrow pathway to accomplish their research with little chance to explore.

Scientific careers are unfortunately tied to timely publication of results. Developing a model and running hundreds or thousands of ice sheet simulations for Bayesian analysis are not possible given the time frames for most tenuously employed researchers.

**12    Final remarks**

A couple of years ago (Gowan et al., 2021), I published an ice sheet reconstruction study where my goal was to investigate the "missing ice problem", the problem that the global sea level drop at the LGM did not match the evidence of ice sheet configuration (*i.e.* volume). Using a single, hand tuned reconstruction, I found that the far-field sea level observations could be matched with a smaller volume ice sheet configuration than previously assumed. I remarked:

> Our reconstruction also demonstrates that there is no consensus on Late Pleistocene ice volume, and we anticipate future refinements, for instance with different Earth rheology assumptions and ice margin histories, will produce different configurations.

In this statement, I made it very clear that I do not consider the reconstruction I published to be the one true reality, but rather that the uncertainties on ice volume are likely larger than was previously assumed (*i.e.* the uncertainty on ice volume at the LGM may be as large as 20 m of sea level equivalent). The single reconstruction, however, was sufficient to tell the story I wanted to tell. Now, if you read what Tarasov and Goldstein have written, I should not have published this because I have not done the job of quantifying the uncertainty ranges explicitly. Would I have loved to perform the ice sheet reconstruction exercise with a greater range of Earth models and ice margin histories to produce an uncertainty range? Of course! The reality is that I (along with many other modellers) do not always have the luxury of time, computer resources and funding to do this. If this makes me a "mongrel" (line 88), so be it.

From my perspective, there is a still long way to go before dynamic models can reliably be used to precisely reconstruct past climate and ice sheets. I believe that the uncertainties in ice sheet and climate

models are more likely to be narrowed through laboratory experiments, field data measurements and idealized modelling experiments where a small number of parameters are varied in a controlled way.

In the end, I am not going to suggest that the methodology that Tarasov and Goldstein want everyone to use is necessarily wrong, even if I do not intend to follow it for the reasons I have stated above. I am not a person who will stifle the creativity and imagination of other scientists. Instead, I challenge Tarasov and Goldstein to create a model development study demonstrating their Bayesian ensemble analysis concept, and publish the programs and scripts that make it possible to perform. At an even more basic level, they need to show that what they are proposing is even possible (from my reading of section 4.7, it doesn't sound like they are sure). If people agree with their methodology, they would then have the option to incorporate it into their own modelling framework.

Best Regards,
Evan J. Gowan

**References**

Bahadory, T., Tarasov, L., Andres, H., 2021. Last glacial inception trajectories for the Northern Hemisphere from coupled ice and climate modelling. Climate of the Past 17, 397–418. doi:`10.5194/cp-17-397-2021`.

Broecker, W.S., Kennett, J., Flower, B., Teller, J., Trumbore, S., Bonani, G., Wolfli, W., 1989. Routing of meltwater from the Laurentide Ice Sheet during the Younger Dryas cold episode. Nature 341, 318–321. doi:`10.1038/341318a0`.

Bronk Ramsey, C., 2009. Bayesian analysis of radiocarbon dates. Radiocarbon 51, 337–360. doi:`10.1017/S0033822200033865`.

Clark, P.U., He, F., Golledge, N.R., Mitrovica, J.X., Dutton, A., Hoffman, J.S., Dendy, S., 2020. Oceanic forcing of penultimate deglacial and last interglacial sea-level rise. Nature 577, 660–664. doi:`10.1038/s41586-020-1931-7`.

Duk-Rodkin, A., Barendregt, R.W., Tarnocai, C., Phillips, F.M., 1996. Late Tertiary to late Quaternary record in the Mackenzie Mountains, Northwest Territories, Canada: stratigraphy, paleosols, paleomagnetism, and chlorine-36. Canadian Journal of Earth Sciences 33, 875–895. doi:`10.1139/e96-066`.

England, J.H., Furze, M.F., Doupé, J.P., 2009. Revision of the NW Laurentide Ice Sheet: implications for paleoclimate, the northeast extremity of Beringia, and Arctic Ocean sedimentation. Quaternary Science Reviews 28, 1573–1596. doi:`10.1016/j.quascirev.2009.04.006`.

Gandy, N., Gregoire, L.J., Ely, J.C., Cornford, S.L., Clark, C.D., Hodgson, D.M., 2021. Collapse of the Last Eurasian Ice Sheet in the North Sea Modulated by Combined Processes of Ice Flow, Surface Melt, and Marine Ice Sheet Instabilities. Journal of Geophysical Research: Earth Surface 126, e2020JF005755. doi:`10.1029/2020JF005755`.

Gowan, E.J., Hinck, S., Niu, L., Clason, C., Lohmann, G., 2023. The impact of spatially varying ice sheet basal conditions on sliding at glacial time scales. Journal of Glaciology , 1–15doi:10.1017/jog.2022.125.

Gowan, E.J., Tregoning, P., Purcell, A., Lea, J., Fransner, O.J., Noormets, R., Dowdeswell, J.A., 2016a. ICESHEET 1.0: a program to produce paleo-ice sheet reconstructions with minimal assumptions. Geoscience Model Development 9, 1673–1682. doi:10.5194/gmd-9-1673-2016. 2016.

Gowan, E.J., Tregoning, P., Purcell, A., Montillet, J.P., McClusky, S., 2016b. A model of the western Laurentide Ice Sheet, using observations of glacial isostatic adjustment. Quaternary Science Reviews 139, 1–16. doi:10.1016/j.quascirev.2016.03.003.

Gowan, E.J., Zhang, X., Khosravi, S., Rovere, A., Stocchi, P., Hughes, A.L.C., Gyllencreutz, R., Mangerud, J., Svendsen, J., Lohmann, G., 2021. A new global ice sheet reconstruction for the past 80 000 years. Nature Communications 12, 1199. doi:10.1038/s41467-021-21469-w.

Hinck, S., Gowan, E.J., Zhang, X., Lohmann, G., 2020. PISM-LakeCC: Implementing an adaptive proglacial lake boundary into an ice sheet model. The Cryosphere Discussions 2020, 1–36. doi:10.5194/tc-2020-353.

Jackson, L.E., Andriashek, L.D., Phillips, F.M., 2011. Limits of successive middle and late Pleistocene continental ice sheets, interior plains of southern and central Alberta and adjacent areas, in: Ehlers, J., Gibbard, P.L., Hughes, P.D. (Eds.), Quaternary Glaciations - Extent and Chronology A Closer Look. Elsevier. volume 15 of *Developments in Quaternary Sciences.* chapter 45, pp. 575–589. doi:10.1016/B978-0-444-53447-7.00045-3.

Kageyama, M., Albani, S., Braconnot, P., Harrison, S.P., Hopcroft, P.O., Ivanovic, R.F., Lambert, F., Marti, O., Peltier, W.R., Peterschmitt, J.Y., Roche, D.M., Tarasov, L., Zhang, X., Brady, E.C., Haywood, A.M., LeGrande, A.N., Lunt, D.J., Mahowald, N.M., Mikolajewicz, U., Nisancioglu, K.H., Otto-Bliesner, B.L., Renssen, H., Tomas, R.A., Zhang, Q., Abe-Ouchi, A., Bartlein, P.J., Cao, J., Li, Q., Lohmann, G., Ohgaito, R., Shi, X., Volodin, E., Yoshida, K., Zhang, X., Zheng, W., 2017. The pmip4 contribution to cmip6 – part 4: Scientific objectives and experimental design of the pmip4-cmip6 last glacial maximum experiments and pmip4 sensitivity experiments. Geoscientific Model Development 10, 4035–4055. doi:10.5194/gmd-10-4035-2017.

Kaufman, D.S., Manley, W.F., 2004. Pleistocene maximum and Late Wisconsinan glacier extents across Alaska, USA. Quaternary Glaciations-Extent and Chronology: Part II: North America 2, 9–27. doi:10.1016/S1571-0866(04)80182-9.

Lewis, S.C., 2017. A Changing Climate for Science. Palgrave Macmillan Cham. doi:10.1007/978-3-319-54265-2.

Lofverstrom, M., Thompson, D.M., Otto-Bliesner, B.L., Brady, E.C., 2022. The importance of Canadian Arctic Archipelago gateways for glacial expansion in Scandinavia. Nature Geoscience 15, 482–488. doi:10.1038/s41561-022-00956-9.

Lougheed, B., Obrochta, S., 2016. MatCal: Open source Bayesian 14 C age calibration in MatLab. Journal of Open Research Software 4.

Niu, L., Lohmann, G., Hinck, S., Gowan, E.J., Krebs-Kanzow, U., 2019. The sensitivity of Northern Hemisphere ice sheets to atmospheric forcing during the last glacial cycle using PMIP3 models. Journal of Glaciology 65, 645–661. doi:`10.1017/jog.2019.42`.

Pollard, O.G., Barlow, N.L.M., Gregoire, L., Gomez, N., Cartelle, V., Ely, J.C., Astfalck, L.C., 2023. Quantifying the uncertainty in the eurasian ice-sheet geometry at the penultimate glacial maximum (marine isotope stage 6). The Cryosphere Discussions 2023, 1–31. doi:`10.5194/tc-2023-5`.

Ross, M., Utting, D.J., Lajeunesse, P., Kosar, K.G.A., 2012. Early Holocene deglaciation of northern Hudson Bay and Foxe Channel constrained by new radiocarbon ages and marine reservoir correction. Quaternary Research 78, 82–94. doi:`10.1016/j.yqres.2012.03.001`.

Shennan, I., 1982. Interpretation of Flandrian sea-level data from the Fenland, England. Proceedings of the Geologists' Association 93, 53–63. doi:`10.1016/S0016-7878(82)80032-1`.

Tarasov, L., Dyke, A.S., Neal, R.M., Peltier, W., 2012. A data-calibrated distribution of deglacial chronologies for the North American ice complex from glaciological modeling. Earth and Planetary Science Letters 315–316, 30–40. doi:`10.1016/j.epsl.2011.09.010`.

Trommelen, M., Levson, V., 2008. Quaternary stratigraphy of the Prophet River, northeastern British Columbia. Canadian Journal of Earth Sciences 45, 565–575.

Woodroffe, S.A., Long, A.J., Lecavalier, B.S., Milne, G.A., Bryant, C.L., 2014. Using relative sea-level data to constrain the deglacial and Holocene history of southern Greenland. Quaternary Science Reviews 92, 345–356. doi:`10.1016/j.quascirev.2013.09.008`. aPEX II: Arctic Palaeoclimate and its Extremes.

---

## Referee Comment (RC3)

**1 General Overview**

This paper should be viewed as a review and tutorial for the Bayesian analysis of computer models aimed at paleo climate and ice-sheet modellers, although the general framework transcends many scientific disciplines. The authors have a clear aim which is to encourage the paleo modelling community to employ a full Bayesian uncertainty quantification framework in the analysis of their (coupled systems of) computer models, and most importantly making inferences for the real world physical system by jointly incorporating all sources of uncertainty, including structural model discrepancy. Moreover, the paper sets out the importance of ensuring that all aspects of a modeller's statistical framework are transparent, defensible and explicitly outlined within papers; as is good practice for peer-reviewed scientific publication and for obtaining reproducible results. Other positives from this manuscript include: the discussion of the limitations of Multi-Model Ensembles (MMEs) which are prevalent across paleo modelling, as well as in present-day and future modelling; and the call for a high-quality paleo constraint database, including the associated uncertainties, thus greatly decreasing the time spent devising suitable constraints whilst enabling consistency between studies in terms of the data with which model output is compared.

The framework presented within sections 2 and 3 does not constitutes novel methodology, instead forming an amalgamation of the Bayesian analysis of computer models literature, as well as some applications to other scientific disciplines. However, the authors attempt to present this as a complete Bayesian framework for paleo modellers, including numerous practical suggestions for its implementation. The content is therefore of greatest use to those less familiar with such methods, although could also be used as a reference by those more experienced in statistical uncertainty quantification.

**2 Major Concerns**

Below I outline my major concerns regarding this manuscript.

**2.1 Manuscript Length**

Firstly the length of the manuscript is excessive and would require a committed and motivated reader to finish the paper. In addition, the abstract, introduction and conclusion do not necessarily provide an adequate summary of the main methods and the consequences of (none) implementation. The authors are ambitious in the breadth of

topics from the Bayesian analysis of computer experiments literature that they include resulting in this manuscript having the feel of the notes from a lecture course. For any reader with even a moderate understanding or familiarity of Bayesian statistics, this treatment seems unnecessary. For example, sections 2.1 and 2.2 could be much briefer with the reader referred to an introduction to probability theory undergraduate textbook or online resource where necessary, rather than repeating the material within this manuscript. The discussion of MCMC in section 2.7 is not the main purpose of this work, hence it could also be condensed without detriment to this manuscript by referring the reader to other papers, textbooks and resources where desired.

**2.2 Structure and Examples**

Throughout the manuscript the authors incorporate numerous examples, both toy models, and from across paleo climate and ice-sheet modelling, but also in relation to present and future modelling efforts. Whilst these provide clarity to the meaning of the statistical framework, in particular, relating this to the analysis of paleo computer models, it also seems like an attempt to conform to the scope of Climate of the Past. In many instances these examples seem unnecessarily long and are often vague in their description without any actual results, for example, lines 399-429, in the discussion of internal model discrepancy. Consequently, these examples greatly add to the length of the manuscript without adding much to the contents. Another concern is the examples interrupt the flow in the presentation of the Bayesian framework that the authors are keen for the reader to adopt. Greater clarity of exposition could be achieved by taking inspiration from Vernon et al. (2018) and restructuring the manuscript, only using short examples where absolutely necessary. Firstly, I would suggest combining sections 2 and 3 to describe the framework including: prior and model specification; emulation (see comments in section 2.3); uncertainty quantification; history matching and the need for simulations to bound reality; and uncertainty quantification in making predictions/retrodictions. This should be followed by a separate section demonstrating how (the majority of) this framework is then applied to a paleo climate and/or ice-sheet model. The paper should finish by briefly highlighting what further steps are required by the paleo modelling community, rather than the extensive section 4 which seems to repeat many of the points found in sections 2 and 3.

**2.3 Statistical Emulation**

Statistical emulators form a vital tool in the analysis of computationally expensive simulators such as paleo climate and ice-sheet models. The authors refer to emulators at multiple points throughout the main body of the paper, however they are only first discussed in any detail at the end of section 2.7, lines 590-596. Given their importance within the presented Bayesian framework as a fast statistical approximation to the simulator's output(s) for as yet unevaluated paper settings along with a corresponding statement of the uncertainty, I believe that emulators warrant a more detailed exposition within the main body of paper, rather than leaving it to Appendix A6. This should

be introduced early on within the manuscript, as discussed in section 2.2.

It is not guaranteed that the reader will know what is an emulator, or more specifically, a Regression Stochastic Process Emulator (RSPE) as it is termed in this manuscript. It is therefore necessary to expand on the discussion currently provided in Appendix A6 with further mathematical details of their formulation, as in Vernon et al. (2010), highlighting the range of possible choices to encapsulate the possible model output behaviour, for example, see Rasmussen et al. (2006) for an in-depth discussion of Gaussian Processes (GPs).

Within this manuscript there is a focus on emulating outputs one-by-one and treating them independently which for many applications is a suitable and justifiable approach. I would recommend that the authors also reference that there exists numerous multivariate emulation techniques such as: separable GP emulation (Conti et al. (2010)); the outer-product emulator which extends the separability assumption onto the regression components (Rougier (2008) and Rougier et al. (2009)); non-separable emulators (Fricker et al. (2013)); parallel partial GPs (Gu et al. (2016)); and through the use of basis representations of multivariate outputs (Higdon et al. (2008) and Salter et al. (2019, 2022)). It is unnecessary to provide further methodological details in this manuscript. A further minor point regarding the approach to emulation described in lines 500-502; it would help to provide the name for this technique as multilevel, multiscale or multi-fidelity emulation, in order to aid the reader in identifying other literature or code implementations.

Practical implementation of the described framework is important and the authors therefore signpost the reader to the recently released `hmer` R package. It would also be useful to mention the accompanying website, `https://hmer-package.github.io/website/index.html`, which provides detailed tutorials, as well as links to published research using the package. For GP emulation, see the `RobustGaSP` R package, `https://cran.r-project.org/web/packages/RobustGaSP/index.html`, and the associated papers (Gu et al. (2016, 2019)). In addition, many within the paleo modelling community use Python as their main programming language, hence it would be of use to suggest similar Python modules such as `GPy` for GP emulation, `https://sheffieldml.github.io/GPy/`, which also includes tutorials.

**2.4   Uncertainty Quantification**

In section 2.4, lines 270-278, the authors introduce what is commonly referred to as an additive error structure. It would aid the reader to state this. Moreover, this seems like the default choice with only a brief mention given to a multiplicative error structure in lines 293-295, whilst it should also be noted that it is not strictly necessary to log-transform the output (both the simulation and observation/reconstruction data) to simply return to the more common additive error structure. It would also add to the content of section 2.4 to comment on biases in observational errors and structural model discrepancy and how these should be accounted for within the error model. A comment should also be included regarding any sources of uncertainty exhibiting a parameter dependency, with this being particularly relevant for (internal) structural model

discrepancy. In equation 14, the authors should comment on the implications of whether the model $M$ is deterministic or stochastic. This is not immediately clear when going from the first to the second line of this equation.

The authors include the following overly strong statement about structural model discrepancy: "Within most (if not all) scientific disciplines, the tendency to date has been to effectively ignore this source of uncertainty" (lines 340-341). Whilst it is true that model discrepancy is often overlooked (or given minimal treatment) within the (paleo) climate and ice-sheet modelling literature, there exist examples across a range of applications where model discrepancy is explicitly assessed, for example: in cosmology (Vernon et al. (2010)); epidemiology (Andrianakis et al. (2015, 2017)); and in climate modelling (Edwards et al. (2019)).

**2.5 A Bayesian Framework for the Analysis of Paleo Computer Models**

Appendix A1 serves as a useful recipe for performing a Bayesian analysis of paleo computer models. Given that the main aim of this manuscript is to promote this framework to the reader for their analyses, would it be best to include a more compact version of this recipe within the main body, for example, at the start of section 2, with cross-referencing to the relevant subsection for each numbered step? A more discursive example could then be left to the appendices.

Throughout appendix A1, the authors provide guidance on the number of simulations required for each step of the framework. However, the exact number of simulations really depends on numerous factors including: the computational cost of running the model; whether multiple models of differing complexities within a hierarchy are being used; what are the available computing resources; whether an iterative analysis such as history matching being performed, or a single stage analysis such as a full Bayesian calibration; the smoothness in the behaviour of the simulator outputs of interest with respect to changes in the parameter settings; prior beliefs about the simulator(s) combined with information gained from previous studies; and the type or form of the statistical model to be fitted. Without explicitly referring to such factors in the authors' previous analysis or analyses, the quoted number of simulations holds limited value. The authors should therefore provide more details of the configuration of previous study or studies to which they are referring, noting that this also links to my above concerns about the lack of a concrete and numerical example raised in section 2.2. In addition, the manuscript would be further strengthened by referencing guidance on the practical choice of the number of simulations detailed in Loeppky et al. (2009).

**3 Technical Corrections**

The following is a list of the technical corrections I have spotted during my review of the manuscript. This is by no means complete and should serve as examples of the types of errors found. I would advise that the authors take greater care to check their manuscript before submission as the many errors or vague choice of language made the

paper difficult to follow.

- Line 48 – Remove repeated fullstop.
- Tables 1 & 2 – Improve the presentation of these tables to aid the reader by adding horizontal and vertical lines to better delineate rows and columns.
- Table 3 – In the row for the "Mathematically limited paleo researcher", there is a "" in the second column. Evidently something is missing here.
- Section 2 title – Capitalise "t" in "the Bayesian framework".
- Line 162 – Model configuration is denoted by the parameter vector $C_M$. To adhere to general conventions regarding scalars and vectors, this would be better denoted in bold font as $\mathbf{C}_M$.
- Line 173 – There is a rogue closing bracket at the end of the line after "multiplication rule".
- Lines 171-175 – Combine equations 7 and 8 as the repetition is unnecessary.
- Footnote 2 – This point seems more important to the understanding of the reader and as such should be elevated to the main body of text.
- Equation 12 (page 9) – The denominator, $P(D)$, is computed using the law of total probability which is not introduced until equation 19 (page 22). For those readers less familiar with probability, it would be useful to provide this formula before (or within) equation 12.
- Lines 195-202 – The translation of the hypothesis based science to Bayes rule would be clearer as a numbered list.
- Line 302 – Unnecessary opening bracket before $R - M(C_M)$.
- Line 328 – Remove the second "this".
- Equations 15-17 – There is a change in notation from capital "$M$" to lowercase "$m$" when describing the model. Is this deliberate? If so, please explain why. Otherwise, change to $M$ for consistency with the rest of the paper.
- Figure 2 – Add light blue confidence interval for the structural error corrected model to demonstrate that this overlaps the data.
- Line 501 – Correct the formatting for the citation for "Cumming and Goldstein, 2009" which runs into the margin.
- Footnote 11 – Add a fullstop to sentence.
- Section 3.1 – The notation $cm$ appears to be used to denote a parameter vector. I assume this is supposed to be $C_M$ for consistency with section 2.
- Line 906 – Remove the word "to" before "towards".
- Section 4.7 title – Capitalise the first word, "a".

**4   Conclusion**

I am supportive of the Bayesian uncertainty analysis framework presented in this manuscript and strongly agree with the need for such reproducible and defensible analyses, whilst admiring the authors' ambition to unify the paleo modelling community in their approach to uncertainties. At this stage I recommend the authors consider rewriting this manuscript due to the numerous points mentioned in this review. In particular, it is

necessary to greatly condense the description of the framework to the core elements, with additional information provided for those interested through supplementary material and via references to other resources that already provide an appropriate coverage of the facet. The exposition would be greatly helped by including a substantive example applying this framework to paleo models, along with appropriate code, which will also motivate and demonstrate to the reader how such an analysis can be practically implemented.

**References**

Andrianakis, Ioannis, Ian R. Vernon, Nicky McCreesh, Trevelyan J. McKinley, Jeremy E. Oakley, Rebecca N. Nsubuga, Michael Goldstein, and Richard G. White (2015). "Bayesian History Matching of Complex Infectious Disease Models Using Emulation: A Tutorial and a Case Study on HIV in Uganda". In: *PLOS Computational Biology* 11.1. ISSN: 1553-7358. DOI: 10.1371/journal.pcbi.1003968.

— (2017). "History matching of a complex epidemiological model of human immunodeficiency virus transmission by using variance emulation". In: *Journal of the Royal Statistical Society: Series C (Applied Statistics)* 66.4, pp. 717–740. ISSN: 0035-9254. DOI: 10.1111/rssc.12198.

Conti, Stefano and Anthony O'Hagan (2010). "Bayesian emulation of complex multi-output and dynamic computer models". In: *Journal of Statistical Planning and Inference* 140 (3), pp. 640–651. DOI: 0.1016/j.jspi.2009.08.006.

Edwards, Tamsin L., Mark A. Brandon, Gael Durand, Neil R. Edwards, Nicholas R. Golledge, Philip B. Holden, Osabel J. Nias, Antony J. Payne, Catherine Ritz, and Andreas Wernecke (2019). "Revisiting Antarctic ice loss due to marine ice-cliff instability". In: *Nature* 566, pp. 58–64. DOI: 10.1038/s41586-019-0901-4.

Fricker, Thomas E., Jeremy E. Oakley, and Nathan M. Urban (2013). "Multivariate Gaussian Process Emulators With Nonseparable Covariance Structures". In: *Technometrics* 55.1, pp. 47–56. DOI: 0.1080/00401706.2012.715835.

Gu, Mengyang and James O. Berger (2016). "Parallel Partial Gaussian Process Emulation for Computer Models with Massive Output". In: *The Annals of Applied Statistics* 10.3, pp. 1317–1347. ISSN: 1932-6157. DOI: 10.1214/16-AOAS934.

Gu, Mengyang, Jesus Palomo, and James O. Berger (2019). "RobustGaSP: Robust Gaussian Stochastic Process Emulation in R". In: *The R Journal* 11.1, pp. 112–136. DOI: 10.32614/RJ-2019-011.

Higdon, Dave, James Gattiker, Biran Williams, and Maria Rightley (2008). "Computer Model Calibration Using High-Dimensional Output". In: *Journal of the American Statistical Association* 103.482, pp. 570–583. DOI: 10.1198/016214507000000888.

Loeppky, Jason L., Jerome Sacks, and William J. Welch (2009). "Choosing the Sample Size of a Computer Experiment: A Practical Guide". In: *Technometrics* 51.4, pp. 366–376. DOI: 10.1198/TECH.2009.08040.

Rasmussen, Carl Edward and Christopher K. I. Williams (2006). *Gaussian Processes for Machine Learning*. The MIT Press. ISBN: 0-262-18253-X. URL: https://gaussianprocess.org/gpml/chapters/RW.pdf.

Rougier, Jonathan (2008). "Efficient Emulators for Multivariate Deterministic Functions". In: *Journal of Computational and Graphical Statistics* 17.4, pp. 827–843. DOI: 10.1198/106186008X384032.

Rougier, Jonathan, Serge Guillas, Astrid Maute, and Arthur D. Richmond (2009). "Expert Knowledge and Multivariate Emulation: The Thermosphere–Ionosphere Electrodynamics General Circulation Model (TIE-GCM)". In: *Technometrics* 51.4, pp. 414–424. DOI: 10.1198/TECH.2009.07123.

Salter, James M., Daniel B. Williamson, John Scinocca, and Viatcheslav Kharin (2019). "Uncertainty Quantification for Computer Models With Spatial Output Using Calibration-Optimal Bases". In: *Journal of the American Statistical Association* 114.528, pp. 1800–1814. DOI: 10.1080/01621459.2018.1514306.

Salter, James M. and Daniel B. Williamson (2022). "Efficient Calibration for High-Dimensional Computer Model Output Using Basis Methods". In: *International Journal for Uncertainty Quantification* 12.6, pp. 47–69. ISSN: 2152-5099. DOI: 10.1615/Int.J.UncertaintyQuantification.2022039747.

Vernon, Ian, Michael Goldstein, and Richard G. Bower (2010). "Galaxy Formation: a Bayesian Uncertainty Analysis". In: *Bayesian Analysis* 5.4, pp. 619–670.

Vernon, Ian, Junli Liu, Michael Goldstein, James Rowe, Jen Topping, and Keith Lindsey (2018). "Bayesian uncertainty analysis for complex systems biology models: emulation, global parameter searches and evaluation of gene functions". In: *BMC systems biology* 12 (1). ISSN: 1752-0509. DOI: 10.1186/s12918-017-0484-3.

---

## Author Comment (AC1)

**Note to editor**

It is unfortunate that reviewers 1 and 3 did not adequately take into account that they are not part of the target audience (nor fully take into account the reader road-map table 3) when writing their reviews. The only reviewer with a paleo background (Evan Gowan, reviewer 2), placed more attention in their review on their own work and previous papers by LT's group and has largely missed the key points of this submission. Reviewer 3 is the most favourable though appears to want this for a broader audience given their statement that the choice of examples "seems like an attempt to conform to the scope of Climate of the Past".

   The core issue we address is what is needed to meaningfully relate model results to the actual real world and thereby make inferences about past ice and climate evolution that the reader can have confidence in. This is fundamental to a good part of the paleo modelling enterprise. It is non-trivial and given the breadth of relevant technical understanding within the paleo community, this is not something that can be accessibly and adequately addressed in ten or twenty pages. The more than 400 downloads of our original submission to Climates of the Past attests to interest in the community in what we present.

   The length issue could be addressed by making our submission into a monograph, but the importance of the above issue warrants a more accessible forum. LT thought Climates of the Past would be more appropriate.

**General response to the reviewers**

What struck the authors is that none of the reviewers addressed a core and logically self-evident message of our submission: As a modeller interested in making inference about actual past ice sheet and/or climate evolution, you need to specify what the relationship is between your model results and the real world and why the reader should have confidence in your assessment of this relationship.

   Reviewers 1 and 3 raise the main issue of length, but present somewhat opposing solutions. Neither reviewer seems to consider that this paper is not meant for either of them (except for subsection 4.7 : a community methodology research and development agenda). The abstract clearly states: "This overview is intended for all interested in making and/or evaluating inferences about the past evolution of the Earth system (or any of its components), with a nominal focus on past ice sheet and climate evolution during the Quaternary". LT has run drafts of this paper by a number of grad students (both in and outside of LT's research group), as well three colleagues in the paleo field. The length is in part due to addressing suggestions and difficulties in comprehension raised by those students and by a reviewer of a previous version.

   The reviewers also fail to take into account that for many in the target audience, the length going by the reader road-map (table 3) is only a fraction of the whole paper.

   The reviewers and editor need to consider, not whether this is a good read for statisticians, but whether this will provide a useful reference for at least some of those within the stated target audience. LT has in good part pursued this project as a document he would have liked to have had 20 or even 15 years ago. Reviewer 1 talks about the need for evidence-based presentation. Then as evidence to the utility of and interest in this document, consider the more than 400 recorded downloads of the original submission to Climates of the Past. This download count is surpassed by only one of the five 2023 highlighted (ie not just published) papers on the Climates of the Past

home web-page.

Yes, we could boil down the underlying message of the paper to 1 paragraph: "For the contexts of making inferences about a physical system, one needs to appropriately specify the relationship between one's model(s) and the system and between one's observation or experimental data and the physical system. Without this specification, the inference, whether it is a hypothesis test, posterior (*i.e.* Bayesian) inference, or whatnot, is by definition uninterpretable. These relationships are respectively the model and data uncertainties." However for the majority of our target audience, the above will have little meaning. Furthermore, "Bayesian" is starting to become a buzzword in paleo-modelling, but many in the field do not have the conceptual basics to critically evaluate what the results of some self-professed Bayesian modelling study means or if it has any interpretable meaning for the stated intent.

In support of the above concerns/motivation, we point the editor and reviewers to a current The Cryosphere submission: "Quantifying the Uncertainty in the Eurasian Ice-Sheet Geometry at the Penultimate Glacial Maximum, Pollard et al." (https://tc.copernicus.org/preprints/tc-2023-5/) that does a far from complete uncertainty assessment contrary to the claim of the title. This is especially evident in the contrast between their adhoc model uncertainty estimate (simply a chosen fraction of ensemble variance) and their plotted misfit of a test "history matching" to an old glaciological model (one of the GLAC1-D chronologies) for last glacial maximum. Furthermore, the abstracts states "We perform Bayesian uncertainty quantification", even though Bayes rule is never invoked.

Instead of reviewing the submission, reviewer 2 (Evan Gowan) focused on presenting their own work as well as arguing against a number of points that are in fact opposite to what we make in our submission. Their review also make a number of erroneous claims concerning two papers from LT's group (one of which is over a decade old). Nevertheless some of their arguments offer a useful foil for expanding our key points. Gowan's whole review also provides a clear example of why we feel our submission is important for the paleo community.

In the following we address reviewer points.
* * *
**Reviewer 1**

**Reviewer Point P 1.1** — Despite the want for this paper, I found it long, dense and difficult to follow, largely containing opinions rather than evidence-based guidance, and ignorant of the existing statistical literature.

**Reply**: After 4 months of not looking at the submission, LT has gone through a complete read and edit. He has cleaned up a few spots where reading stumbled, but did not find it "difficult to follow". The reviewer seems to expect this paper to be a review of relevant statistical litterature, it is not. It would also help if the reviewer provided some example evidence to back their claim of "largely containing opinions". The appendix offers a suggestion (not prescription) for how to address a number of the issues raised based on the authors' experience from working on these issues for decades. Perhaps one is free to call experience "opinion". We do not see anything in the main text that can be called an "opinion".

**Reviewer Point P 1.2** — This paper has been previously submitted for publication (a public process in Climate of the Past), and previously reviewed. I have reservations that many of the

previous reviewers' comments have not been sufficiently addressed, and find myself agreeing with the previous reviewers on many points.

**Reply**: The previous version of this paper that was submitted to Climate of the Past was heavily revised in the second stage review, but was rejected by the editor (given the paper length) without going back to the reviewers. Given our extensive response (24 pages mostly in response to the 10 page review by Danny Williams) to the original reviewers (https://cp.copernicus.org/preprints/cp-2021-145/cp-2021-145-AC2-supplement.pdf) along with the edits, it would help to have specific examples of which of the reviewer comments were not adequately addressed.

**Reviewer Point P 1.3** — 22 pages are spent introducing facets of the Bayesian framework after which there were a short(er) 4 pages on some useful techniques and then 7 pages of, from what I can tell, opinions with no examples.

**Reply**: We are curious about what the reviewer see's as "opinions". Section 4.1 states "A key step to towards encouraging rigorous uncertainty assessment about earth system evolution would be for modellers, editors, and reviewers to ensure the following questions are answered: What model parameters are adjusted and to what extent is the criteria used for their selection explicit and appropriate? To what extent does the chosen set of constraint data capture what is available and relevant? To what extent have the associated uncertainties been explicitly accounted for?..."

Are the above all just opinions that any modeller can defensibly choose to ignore when making claims about inferring past ice sheet or climate evolution? Or taking the more clear "opinion" subsection 4.7 a community methodology research and development agenda. Yes there is judgement involved in the selection of agenda items. Is the reviewer equating opinion with judgement?

**Reviewer Point P 1.4** — If we're just going to history match why do we need MCMC and posterior predictive distributions? In fact, why do we need Bayes rule at all?

**Reply**: Even if we are history matching, we still need a conceptual framework for understanding what history matching is and isn't and what by contrast a meaningful Bayesian inference entails. "Bayesian" is starting to become a buzzword in the paleo modelling community, but unfortunately usually as a stated claim not reflected in the actual analysis given the limited uncertainty assessment and often limited sampling.

History matching also requires emulators which currently are generally either Bayesian or Bayes Linear.

**Reviewer Point P 1.5** — Writing and grammar. As a whole I find the paper to be quite sloppy. To name just some of my concerns, title cases are off, table formatting is inconsistent, there are erroneous parenthesis, the figure fonts are massive, and Tony O'Hagan needs an apostrophe. Some grammatical and editorial errors are inevitable, but I would expect a more thorough edit to be conducted before the submission of a journal article (especially before the second submission). I would also check your usage of colons throughout, the clause preceding the colon should be a complete sentence: see, for example, line 395. Also, the authors' colon spacing (with a space preceding the colon, and only sometimes) is foreign to me and I can't see a similar example in either US or UK style guides. Finally, the writing reads like a series of dot-points that ramble on rather than with any real narrative or structure. I keep finding myself lost and have to remind myself of what is happening and where am I going.

**Reply**: Figure fonts are "massive" on purpose. Contrary to the attention placed on colour-blind accessible colour maps, the usage of difficult to read small fonts seems to get no attention. For a description of writing and grammar being "quite sloppy", the listed issues seem relatively minor and are easy to fix (and have now been fixed).

**Reviewer Point P 1.6** — Literature. There is a lot of literature in statistics on calibration of computer models, prior selection and elicitation, emulation, modelling simulator discrepancies, as well as accessible introductory texts on Bayesian analysis. Not much of this is cited. Some of the more statistical texts are potentially inaccessible to the uninitiated;

**Reply**: This submission is not meant to be a comprehensive review on the topic. We have carefully chosen a limited set of references to provide the interested reader with no Bayesian stats background a starting point into the litterature. This is an obvious difference in judgement (or "opinions") between the reviewer and the authors. LT, for instance, when trying to grasp a new methodology, would rather start with a targeted list of refs instead of a complete and overwhelming list of the current litterature. We would welcome suggestions of more appropriate choices of citations with the above in mind.

**Reviewer Point P 1.7** — I also feel the paper would benefit by acknowledging, in text, what applied work is being conducted, what is it doing well, and what is it missing (with appropriate citations). A dense table of citations is hard for the human brain to process, and despite my best efforts I still don't have an appreciation for what the authors are trying to say in the tables.

**Reply**: For eg table 1, we feel this is clearly spelled out in the text: "To provide some motivation for this overview, and the chosen focus on a paleo context, it is worth considering glaciological model reconstructions of past ice sheet evolution (Table 1). To date, none have adequately addressed relevant uncertainties. Furthermore, the number of ensemble model parameters varies widely (from 2 to 39), as does the range and quantity of data constraints. No published studies after 2014 have used more than 5 ensemble model parameters, raising concerns about methodological progress in assessing uncertainties within the paleo ice sheet community".

**Reviewer Point P 1.8** — 4. Lack of examples. This paper is quite dogmatic about the need to do a seemingly arbitrary sub-set of modelling stages but does not provide an example of it actually being done. Surely the authors have a more relevant toy simulator (that is not a linear model) that can be used to discuss the statistical concepts and also provide a helping hand to the struggling reader. In reference to point 2, adding an example that traverses the paper will add narrative and continuity.

**Reply**: The reviewer's assessment contrasts with that of reviewer 3 who states "throughout the manuscript the authors incorporate numerous examples". The reviewer is forgetting the technical breadth of the audience. Reviewer 1 for our original submission, though an internationally accomplished glacial geologist, never had a University level math course. Going beyond a linear model would limit accessibility to a part of our target audience, and for what gain given the conceptual intent?

As to the suggestion of an example that traverses the paper, going beyond the conceptually very accessible linear regression example would entail a major step up in details, the simplest contextually relevant model would probably be a shallow ice approximation flow-line model. But that will already involve a number of details (mass-balance forcing, basal drag, basal topography, temperature forcing)

that will result in a longer paper, with potentially reduced accessibility for part of the audience. For the part of the audience interested in detailed implementation, such a paper is currently being written up by LT for the context of approximate ice sheet history matching.

**Reviewer Point P 1.9** — Feasibility. I have a fundamental problem with the feasibility of some of the methods recommended. In my experience climate models are not computationally cheap to run, although I admit that the authors have more experience than me here and so could perhaps provide run times of certain simulations that are valuable to the community. In the internal discrepancy section the authors recommend that the parameter space is effectively explored and for each of these points the boundary conditions are sufficiently explored to accurately calculate potentially quite large variance covariance matrices.

**Reply**: The comment reflects a limited understanding of the range of climate and earth system models as well as the computational cost of ice sheet models. LT's 3D glacial systems model (GSM) can run a full glacial cycle for Eurasia at roughly 50 km grid resolution on a single commodity compute core in about 5 hours (depending on ensemble parameter values using a cluster that is over 7 years old). LT's group is also running the fully coupled LCice (LoveClim EMIC and GSM) that can run about 1500 years in 24 hours on a single core. As already cited in the submission, LT's group explored efficient approximate Bayesian calibration of reduced complexity general circulation climate models back 2011. The core issue is something that is already raised in the paper. There are design and resource choices being made to focus all efforts and resources on the latest most complex and highest resolution climate models that can be run over the intervals of interest. There is much less effort put towards using/building somewhat simpler climate models with low enough resource requirements to enable more than trivial uncertainty assessment.

**Reviewer Point P 1.10** — The process of (1) exploring the parameter space in the order of millions of times, (2) exploring the spatio-temporal boundary conditions at each of these millions of locations, and (3) predicting and calibrating from these models seems computationally demanding in the extreme.

**Reply**: Nowhere do we state a step (2) above for each location in the parameter vector sampling. Either the parameter vector includes components to specify the boundary conditions and/or the rest of the impact is addressed via internal discrepancy assessment. And yes, this is computationally demanding, thus our suggestion that history matching would be a first major step (and overall a more appropriate tool for this context) for which the uncertainty assessment is much simpler.

**Reviewer Point P 1.11** — Further, the boundary conditions are generated from "adding appropriately correlated noise" – in my personal experience this process is not simple and to cast it as such is wrong. Accurately representing, modelling and quantifying uncertainty for many spatio-temporal processes is exacting on even the most seasoned statistician and I think at least a nod to this should be included

**Reply**: The reviewer is mis-representing the sentence by not quoting the critical prior "could be created by" in the sentence, making clear this is not prescriptive (though something like this would be needed for full Bayesian posterior inference). We will add some simpler options if developing more limited internal discrepancy error models for history matching.

**Reviewer Point P 1.12** — Later in the paper, emulators are thrown into the mix with no explanation or introduction. Do your methodologies need emulators, and if so where and why?

**Reply**: "Later" is a bit disingenuous, as emulators are first introduced right after the scale of the sampling problem is indicated: "In the first author's own experience with ice sheet model calibration of approximately 40 ensemble parameters, at least order ten million point sampling is still required (as compared to the astronomical $10^{40}$ for a simple grid search over deciles). As this is still beyond computational tractability for ice sheet and climate models, one other component is required. This component, a set of emulators, consists of very fast approximate statistical models that predict statistical characteristics of simulator output of interest as a function of an input parameter vector."

**Reviewer Point P 1.13** — Point of the paper. I feel the paper tries to do too many things, and so falls short on each of them. It attempts to provide textbook level mathematics on fundamental statistical principles, ground said mathematics in the application, and then provide recommendations and guidance. I know that the authors have tried to write a document that is accessible to a non-mathematical audience, but by taking the middle ground and then trying to teach mathematics it distracts from the rest of the paper. I would recommend removing as many equations as possible and instead focusing on what the point of the mathematics is. The equations and the rigour can instead be deferred to a supplement for people to read once they've decided that it is worth the time and burden to learn and implement probability theory, MCMC, history matching, and the other number of techniques mentioned.

**Reply**: We have already kept the math to the minimum and unlike as implicitly suggested by the reviewer, no equations are provided on MCMC. One intent was for the mathematical competent reader to understand how a Likelihood can be specified and we do not see how this could be done without the minimal number of equations. Again the reviewer is pointed to the Table 3 road-map. The choice of target audience and submission goals should be that of the authors, not the reviewer.

**Reviewer Point P 1.14** — Unfortunately, I do not think that the current version of this manuscript does a good enough job. I recommend a significant re-write that prioritises the point of the mathematics, and not equations for equations' sake.

**Reply**: Instead of broad swipes, it would be more helpful if the reviewer gave concrete examples of where he/she thinks we are using "equations for equations' sake".

**Reviewer Point P 1.15** — I recommend ... that has narrative and verve rather than 40+ pages of dense writing; and that leads by example with some demonstration of feasibility so that we are motivated to follow.

**Reply**: We may strongly differ on what narrative and verve mean and where it's appropriate.
* * *
**Reviewer 2**

As detailed below, reviewer 2 (Evan Gowan) is using their "review" to espouse their own methodology, make numerous erroneous claims, and has fundamentally missed the point of the submission. Our submission does not advocate full Bayesian inference for the stated context in at least the near

term. The submission presents Bayesian inference as a conceptual framework for making inferences that we argue is the probabilistic underpinning of the classical scientific method. This presentation is in part so that the reader can understand that Bayesian inference without robust uncertainty assessment has limited to uninterpretable meaning.

**Reviewer Point P 2.1** — Tarasov and Goldstein propose that paleo (specifically ice sheet) modellers and data practitioners should be making use of Bayesian analysis in order to ensure that everything has an error uncertainty attached to it.

**Reply**: The reviewer has seemed to miss a key point of the paper. We are not advocating determination of Bayesian posteriors anytime soon. We do state, and it follows by the definition we provide, that a scientific inference with no meaningful uncertainty assessment has no usefully interpretable meaning about the actual physical system. That assessment might be trivial depending on the context but this is generally not the case for paleoclimate and paleo ice sheet modelling contexts.

**Reviewer Point P 2.2** — They suggest that most ice sheet modelling exercises do not provide what they think is an adequate assessment of uncertainty, and that the only way forward is to run hundreds or thousands (or even millions, line 586) of model simulations to create an uncertainty range.

**Reply**: The reviewer ignores the subsequent relevant lines starting at 590: "As this is still beyond computational tractability for ice sheet and climate models, one other component is required. This component, a set of emulators, consists of very fast approximate statistical models that predict statistical characteristics of simulator output of interest as a function of an input parameter vector."

**Reviewer Point P 2.3** — In the first part of the paper, they go over what Bayesian inference is (using a strange and poorly explained example (line 141),

**Reply**: We are not clear what is "strange" about the conditional probability example (which is not a Bayesian inference example as the reviewer erroneously states). Nor do see what needs to be explained, given that this is just an example of the precisely stated definition of conditional probability in the previous sentence, sic: "The expression $P(A = a|B = b)$ denotes the conditional probability of the variable A having some specific value "a" if the statement that the variable has some value "b" were true".

**Reviewer Point P 2.4** — Dr. Tarasov has made it very clear to me (and probably many other paleoclimate scientists) that any study that does not involve a full Bayesian uncertainty assessment is not worth doing.

**Reply**: Dr. Gowan is misrepresenting what LT has communicated. Furthermore, bringing in allegations and/or confused past perceptions instead of the concrete task at hand of addressing what is in the submission, is very unprofessional.

**Reviewer Point P 2.5** — It is thoroughly unapproachable to anyone who is unfamiliar with advanced statistics (and maybe even to those with such training, judging by the other reviewer's comments).

**Reply**: That claim doesn't match the response LT has received from a number of grad students and a couple of colleagues in the paleoclimate and paleo sealevel fields who've gone through earlier drafts.

**Reviewer Point P 2.6** — I want to use this opportunity to provide an alternative view to what Tarasov and Goldstein are propos- ing, because I disagree with the idea that Bayesian uncertainty assessment is needed in every paleoclimate problem.

**Reply**: We are not clear what the reviewer means by Bayesian uncertainty assessment. If they mean Bayesian posterior inference, than as already been made clear in the general remarks to reviewers, this is clearly not what we are proposing.

If the reviewer means full accounting for uncertainties, then we would like to know where the reviewer disagrees with the following chain of logic presented in the first paragraph of the conclusions: "This review started from the premise that a defining feature of any aspect of science which is concerned about making statements about the real world is the rigorous quantification of uncertainties. Within the context of computational models, this claim can be easily supported if one recognizes that uncertainty assessment is simply the principled assessment of the relation of model results to the physical system. Without robust uncertainty estimation or, at the very least, a more limited mix of quantitative and qualitative assessment by the modeller, the reader has no basis to interpret the relevance of modelling results to the actual physical system. As our toy model illustration demonstrated above (c.f. Sect. 2.5), ignorance of structural uncertainties will generally result in model predictions that do not intersect the physical system within computed prediction limits, even if Bayes Rule is used for the inference."

**Reviewer Point P 2.7** — The First point I will raise is that if you read this commentary, Tarasov and Goldstein make it sound like they are the only people who are using Bayesian analysis in paleoclimate applications.

**Reply**: The reviewer makes a derogatory and misguided allegation without any substantive evidence (eg example text) to back up the statement. First, we are not advocating Bayesian inference for at least near term paleo modelling work. Furthermore, if the reviewer were to look at Table, 1, only 1 of the two cited papers from LT's group are indicated as having any structural uncertainty assessment with the added qualification of "limited".

**Reviewer Point P 2.8** — The code for the Glacial Systems Model that Dr. Tarasov uses for his ice sheet modelling and Bayesian analysis exercises have never been made publicly available, despite the fact that the primary paper describing the application of it was published over a decade ago (Tarasov et al., 2012). In my opinion, it is arrogant for Tarasov and Goldstein to say with regards to paleo ice sheet modelling efforts to date none have adequately addressed relevant uncertainties (Line 71)", when they have not contributed their own programs that, according to them, are the only way to evaluate model uncertainties.

**Reply**: It is also unprofessional for the reviewer to make claims that cannot be backed up by relevant text, in this case "their own programs that, according to them, are the only way to evaluate model uncertainties". Where do we make such claims in the text?

As for the Glacial Systems Model code, the model has heavily evolved over the last decade (changelog since 2015 has currently 21273 lines). Such ongoing changes would have made it a pain for anyone outside of LT's group to use. LT has been working on making the model easier to use and port, as

well as writing an associated paper that will document the model. A bare-bones version of the GSM for idealized configurations is already available in a publicly accessible archive (cf assets for Hank et al. https://egusphere.copernicus.org/preprints/2023/egusphere-2023-81/ ). The paper with a full code archive for paleo ice sheet modelling is nearing completion for submission.

As for the statistical tools, we already cite a new freely available history matching code suite in the text.

**Reviewer Point P 2.9** — As such, can a test truly be made to demonstrate whether a model is right or wrong?

**Reply**: The Popperian framework doesn't include testing to demonstrate a model (in the general sense) is right, it only includes falsifiability. And yes, models in the form of theories (which generally assume 0 uncertainties for a stated context), such as Newtonian Gravitation, can and have been falsified for specific contexts. If the reviewer is restricting their concept of "model" to computational models ("simulators" within the uncertainty quantification community), this does not change as both theories and computer models are often used as approximate representations of the physical world around us. The whole history matching approach (described in section 3 of the submission) is based on falsification ("ruling out") of trial models within a chosen statistical threshold (and this can be done in milliseconds not "centuries").

**Reviewer Point P 2.10** — Tarasov and Goldstein argue that with the usage of Bayesian analysis that it is possible to :. go confidently well beyond storytelling but, as detailed below, only when all uncertainties are rigorously addressed and assessed" (line 37-38). But as highlighted above, this is impossible, because an ice sheet model is an approximation of the real world and will always have some level of uncertainty that will be impossible to statistically model.

**Reply**: Any model is an approximation of the real world. So, as we spell out in the paper, physical world = model + uncertainty. If you are advocating we can ignore uncertainty (or some component of it), then you are either saying uncertainty is negligible or that you are not concerned about relating the model to the physical world.

To simplify this discussion, let's focus on history matching (which we are advocating as to where efforts should be focused the next while). All history matching needs is uncertainty quantification that brackets the relationship between model and reality within a chosen confidence range (being loose on the term "confidence"). And that is doable.

**Reviewer Point P 2.11** — The usefulness of ice sheet modelling therefore comes not from being able to predict some geologically constrained reality, but rather from their utility in storytelling (that is, to pose problems) to help explain our world.

**Reply**: We do not denigrate hypothesis creation (aka storytelling). We are challenging claims about actual past ice or climate system evolution based on models for which the uncertainty relationship between model and physical system is not meaningfully (within a scientific context) specified.

**Reviewer Point P 2.12** — The consequence of this problem in terms of a Bayesian calibration procedure that Tarasov and Goldstein are proposing, is that the resulting error bars are going to be strongly biased to the climate model that they are using. This problem is likely so severe that I question the usefulness of this exercise.

**Reply**: Anyone looking at the divergence of climate sensitivity or the precipitation errors of current PMIP 4 models will already have a sense of the large uncertainties even in current state-of-the-art earth system models. So again, what value are the models in relation to inferences about past physical system evolution without some kind of uncertainty assessment specifying the relationship between the model and actual physical system. As such, the reviewer's point repeats some of the arguments we present for the need for robust uncertainty assessment.

As an aside, instead of blind attacks, the reviewer would benefit from actually reading some of LT's modelling papers in which it is made quite clear he does not use a single climate model and that the majority of order 40 ensemble parameters for each paleo ice sheet that his group has worked on are there to try to address the uncertainties in climate forcing. LT is curious if the reviewer has complained about the large majority of published paleo ice sheet modelling papers that only use a handful of ensemble parameters?

**Reviewer Point P 2.13** — In this paper, they apply their Bayesian analysis scheme on a model that couples their Glacial Systems Model with a climate model of intermediate complexity.

**Reply**: Again the reviewer is mis-representing the paper. Even a simple text search of "Bayesian" in that paper will come up blank. No Bayesian analysis is attempted in that paper and such mis-representation is unprofessional.

**Reviewer Point P 2.14** — In their current state, ice sheet models do not have the predictive power to precisely reconstruct ice sheet history.

**Reply**: Nor will they have such ability for precise reconstruction in the foreseeable future, thus the need for structural uncertainty assessment.

**Reviewer Point P 2.15** — The story-line of this paper should have been that it demonstrated that it is possible to grow an ice sheet rapidly after inception started, and that the Eurasian and North American ice sheets have different sensitivities to external forcing. That, to me, would be a more useful application of this modelling exercise, as it tells us something interesting about the climate system without over interpreting the results in terms of how the ice sheets incepted.

**Reply**: The "should have been" story-line is part of the actual story-line as eg the conclusions of the cited paper state "The EA ice sheet is more sensitive to orbital forcing and ensemble parameter values".

**Reviewer Point P 2.16** — The Bayesian analysis technique generally uses Latin hypercube sampling in order to select the values of the parameters that are varied in the modelling experiments (lines 1164-1168)

**Reply**: Incorrect as stated. Is the reviewer conflating MCMC sampling via emulators with the exploratory Latin Hypercube ensemble described in 1164-1168? Furthermore, the example history matching framework in A1 is not Bayesian, though it can be used as a "stepping stone towards a complete Bayesian inversion".

**Reviewer Point P 2.17** — This presents what I consider the biggest weakness of large parameter studies that are used for Bayesian analysis. In a large ensemble with many parameters varied in

tandem, it is not possible to clearly tell why one simulation fails while others succeed. This makes storytelling difficult, and reduces the utility of the model to tell us something about the behaviour of the ice sheet. It is much easier to tell a story with an ice sheet model by holding most variables as constant, and varying a small number of variables in a controlled way. In that way, we know exactly how sensitive the outcome of the experiment is to a parameter.

**Reply**: The reviewer is conflating different research goals into a single method. If the modeller is aiming to understand the role of specific parameters, than sensitivity analysis is an appropriate tool. If the modeller wants to understand process interactions, then sensitivity experiments are appropriate. But if the modeller is making a claim about last ice sheet or paleo climate evolution, then one needs to address uncertainties, and given the complexities of the paleo ice and climate system, a handful of model parameters makes that very hard to do.

**Reviewer Point P 2.18** — 6 In modelling, there is always subjectivity

The main appeal of applying the Bayesian analysis approach for ice sheet modelling is that it gives the illusion of objectivity by assigning a probability value to every decision (section 2.1). Since the experiment allows the parameters for each ensemble member to be selected at random from the experimenter's probability range, it removes responsibility of the outcomes from the experimenter. For practical reasons, there is always going to be a certain limits to how wide of a range of values that can be used in a Bayesian analysis modelling study. I will use the North American deglacial study by Tarasov et al. (2012) as an example.

**Reply**: Again, we wonder if the reviewer has actually endeavoured to fully read our submission. Unlike conventional frequentist statistics, Bayesian inference is much more explicitly cognizant of the judgement aspect of statistical inference. In section 2.1 we state "No matter what interpretation of probability one chooses, the assignment of probabilities require judgements. To be testable and potentially falsifiable, these judgements must be made and treated in a rigorous and self-consistent way." In section 2.3 we also state "This judgement aspect has often been a target by critics of Bayesian approaches, with a usual focus on the specification of the prior. This focus has no clear justification as judgements are required for all aspects of the inferential process and not just the 225 initial specification of the prior. But this holds true for any statistical inference including those by frequentist approaches".

**Reviewer Point P 2.19** — In Tarasov et al. (2012), they varied 39 different components of the ice sheet model, ...

**Reply**: At this point, the reviewer's obsession on a more than 10 year old (2012) paper is getting obnoxious, especially when some relevant points are being mis-represented. The GLAC-1D product of that paper is deprecated. And the stated limitations the reviewer raises in this section about GLAC-1D (limitations which were raised explicitly in that old 2012 paper) as well as a number of others are being addressed in ongoing work. LT's struggles with confident Bayesian inference are one reason the current submission recommends history matching or variants thereof.

**Reviewer Point P 2.20** — 7 Using the right tool for the job

In my methodology, I use an ice sheet model with perfectly plastic ice and assumes the ice sheet is equilibrium (Gowan et al., 2016a). This model requires just three inputs: the ice margin at a specified time slice, a model of the basal shear stress, and GIA deformed topography.

**Reply**:  The claim of "just 3 inputs" is highly misleading.  Each of those inputs is a field (varying in space and time) that has associated uncertainties.  Furthermore, the static perfectly plastic approximation is not at all appropriate for any region with ice streams (much of the southern North American ice sheet margin sector, Hudson Strait,...).  The static equilibrium approach that the method entails raises questions of what predictive/retrodictive value it has especially for times prior to last glacial maximum when available paleo constraints are very sparse in time and space.  How are the errors from the large approximations addressed?

**Reviewer Point P 2.21**  —  When my reconstruction was tested with a lake filling algorithm, it successfully captured the geometry Lake Agassiz during the periods that had strandline data (Hinck et al., 2020).  Only a couple of dozens of iterations were needed to tune the ice sheet reconstruction to fit the strandline data (in combination with many other GIA constraints).

**Reply**:  Since you are making a claim of a "reconstruction", how are you specifying the relationship between your reconstruction and past ice sheet evolution?  Or is this just a curve fitting exercise?

**Reviewer Point P 2.22**  —  If you are interested in finding the geometry of Lake Agassiz to make estimates of its volume to say, figure out how much water could have potentially drained out to disrupt Atlantic circulation at the start of the Younger Drays (Broecker et al., 1989), a dynamic ice sheet model would not be an appropriate choice, no matter how many ensemble members are used.  The climate forcing and/or calving in the dynamic ice sheet model would have to be manually manipulated to do it, defeating the purpose of the Bayesian framework that Tarasov and Goldstein are proposing.

**Reply**:  So now the reviewer has gone from the topic of uncertainty assessment to the topic of the appropriate choice of ice sheet model.  Again the reviewer is missing the point, that no matter what type of model is used, the relationship between the model and the physical system has to be specified. What is the reader supposed to make of the volume of water drained the reviewer generated in their modelling? Is the reviewer claiming it was the exact amount, within 50%, within 5000%? Without that specification and justification thereof, the results are meaningless to most except those who understand the intricacies of the physical system and model uncertainties. Furthermore, for the context of actually inferring the physical runoff, the reviewer's approach necessarily ignores (since it's not computed by their modelling approach), the significant contributions to discharge from ongoing surface melt and meltwater drainage and lacustrine ice calving.

   The reviewer also chooses to not mention that Tarasov et al (2012) describe and use a nudging methodology to address the issue they raise in the text block cited by the reviewer "Given the partially lobate structure of the geologically inferred ice margin, as well as the high sensitivity of ice margin location …  it is unlikely that any glacial systems model will ever freely approach inferred margin chronologies to the degree required for accurate modeling of proglacial lakes".

**Reviewer Point P 2.23**  —  It is better to use a model where the margin location is strictly defined.

**Reply**:  Even when the margin location is not precisely known and may have been quite dynamic?

**Reviewer Point P 2.24**  —  8 The data are never perfect … For these reasons, I tend to judge the fit of a the data using a simple "consistent or not" metric, because developing an framework for

inclusion in a more complicated statistical model is not clear. .. The "consistent or not" assessment has served me well in evaluating my models. A more complex statistical model is not needed.

**Reply**: It sounds like the reviewer is asking the reader to blindly accept their judgement of "consistent or not" (as well as "served me well") instead of specifying what it actually means. If the relationship between a datum and the actual physical system, can't be quantitatively specified in some meaningful manner, than again what meaning/value does the datum have in the context of the actual physical system? How is another scientist supposed to assess their judgement of "consistent or not"?

**Reviewer Point P 2.25** — So, instead, I think it is fine to assess the models in a less rigorous (and definitely less computationally intensive) way rather than attempting to create some probability distribution that is going to be hand-wavey.

**Reply**: The reviewer is has quite obviously missed one of the key points of our submission : "However implementation of standard Bayesian inference for complex simulators is a challenging and potentially non-robust endeavour....No matter how the sampling is carried out, the result can be highly sensitive to the exact specification of the error model and therefore the likelihood..As such, inferences for say a most likely ice sheet history will have limited meaning contingent on a large set of assumptions. ..For many contexts, a more limited product than a rigorous posterior inference may have adequate 735 utility and can be much more robust".

**Reviewer Point P 2.26** — I am confident that the general history of the ice sheet is correct.

**Reply**: Again, what do you mean by this beyond empty words? How is a reader supposed to interpret what your results mean in the context of the actual last glacial cycle?

**Reviewer Point P 2.27** — Although it is stated in this paper that GLAC-1D is an ice sheet reconstruction, that is not strictly true. The ensemble average will not fit any particular observation that the modelling exercise used to evaluate the individual simulations, and it is not glaciologically consistent.

**Reply**: Again the reviewer is making false claims. The GLAC-1D chronologies provided long ago to the community are from individual ice sheet model runs. The chronologies are identified by the actual run numbers (e.g. nn9927 for one of the North American chronologies) and it is even stated in Tarasov et al (2012): "The single run (nn9927, with detailed plots and tabulated summary characteristics in the tertiary supplement).

**Reviewer Point P 2.28** — I'd argue that it would have been better to use this exercise to pick a few simulations that performed well, and make those available.

**Reply**: That is what was done. But given finite computational resources and depending on the exact purposes of the PMIP intercomparison it might have have made more sense to have some of those resources used for model runs with bounding ice sheet chronology boundary conditions instead of the best fit.

**Reviewer Point P 2.29** — Tarasov and Goldstein complain about this (lines 62-69), but I think it would be more productive to be attentive to the realities and needs of other modelling groups rather than lecturing them about the need for ensemble studies to produce an uncertainty range.

**Reply**: More borderline ad hominem attacks based on false claims as detailed above.

**Reviewer Point P 2.30** — 10 Telling a story

Tarasov and Goldstein are of the opinion that storytelling alone is not an adequate way to use ice sheet models (lines 38-48).

**Reply**: No. That is nowhere stated nor intended. On the contrary we state "Story-telling, or in more usual terminology, hypothesis creation/elaboration, is a central part of science'. Logically one can't test a hypothesis without first creating it. One of our complaints though is the mis-representation of hypothesis creation as hypothesis testing.

**Reviewer Point P 2.31** — Most modellers fully understand the limits of what their models can predict.

**Reply**: The reviewer has already provided a counter example in their claim of inferring Lake Agassiz discharge. We do state in our introduction "It is also our own experience that modellers who know their models well are often the most skeptical about their model results". And no matter what the modeller understands, this doesn't necessarily mean the reader will understand it without clear meaningful communication of the uncertainties involved.

**Reviewer Point P 2.32** — 11 Ethical modelling 11.1 Carbon Footprint

Due to the discovery of bugs, I had to run these simulations a few times. Each glacial cycle simulation took roughly one week to complete on 144 processors, while the idealized simulations too between one and two days... Imagine running thousands of model experiments and finding a bug that invalidates the results.

**Reply**: The logic is getting pretty desperate here. Why run such an expensive glaciological model thousands of times? LT uses glaciological models that span the range of 5 hours (low resolution) to 1 week (high resolution) using a single compute core. Or by the reviewer's logic, is a current generation earth systems model that takes hundreds of compute cores for a single simulation and that takes weeks to months of run time unethical especially given that the model likely includes bugs? Modellers face a trade-off between fewer runs with computational more expensive (and hopefully more accurate) high resolution complex models versus lower resolution and simpler models that enable larger ensembles for the same amount of compute resources. Currently few resources are being applied to the further development of computationally cheaper models and to the development of methods to efficiently synthesize the results of both cheaper models and expensive models.

**Reviewer Point P 2.33** — Is it practical to develop a sophisticated statistical and modelling framework under these conditions? Do they think that a student or postdoctoral researcher can afford to wait months or years to get enough results to publish something? This is just not feasible under the current funding regime of science. We can only design modelling exercises that can be accomplished under short time-frames. It would be unfair to students and postdoctoral researchers to be forced into a narrow pathway to accomplish their research with little chance to explore.

**Reply**: Why does the reviewer assume that this work has to be done within one group as opposed to being a part of a multi-group collaboration or that every student or post-doc in paleo-modelling must endeavour to do say a full history matching for past ice sheet evolution or climate? If a student is interested in the topic, internal discrepancy assessment for some computationally accessible paleo model,

for example, would already be a useful and achievable project. And how this issue of time required to carry out fits within the "ethical modelling" section title escapes the authors.

**Reviewer Point P 2.34** — From my perspective, there is a still long way to go before dynamic models can reliably be used to precisely reconstruct past climate and ice sheets.

**Reply**: Again, reinforcing the need for meaningful uncertainty assessment.

**Reviewer Point P 2.35** — Using a single, hand tuned reconstruction, I found that the far-field sea level observations could be matched with a smaller volume ice sheet configuration than previously assumed. ...

**Reply**: With a "single hand tuned reconstruction" and no uncertainty assessment, the reviewer's statement offers no meaningful inference about whether past LGM ice volume was less "than previously assumed". If, for instance, the error induced by the reviewer's modelling approach is 20 m of sealevel, then their finding is consistent with previous findings.
* * *
**Reviewer 3**

**Reviewer Point P 3.1** — The authors have a clear aim which is to encourage the paleo modelling community to employ a full Bayesian uncertainty quantication framework in the analysis of their (coupled systems of) computer models, and most importantly making inferences for the real world physical system by jointly incorporating all sources of uncertainty, including structural model discrepancy

**Reply**: While the second point is correct, the first point is not. As stated at the very top of general response to all reviewers, we want the target audience to understand what Bayesian inference properly entails and means. But for the stated context, this is currently not an appropriate tool. Instead we are suggesting history matching as a tool that can provide robust meaningful inferences.

**Reviewer Point P 3.2** — The content is there- fore of greatest use to those less familiar with such methods, although could also be used as a reference by those more experienced in statistical uncertainty quantification.

**Reply**: Correct, as we spell out in our Table 3 road-map for various audiences.

**Reviewer Point P 3.3** — Firstly the length of the manuscript is excessive and would require a committed and motivated reader to finish the paper. In addition, the abstract, introduction and conclusion do not necessarily provide an adequate summary of the main methods and the consequences of (none) implementation.

**Reply**: We know it's long, but much of this length has come from ongoing efforts to reach the broad target audience. For many readers, our road-map (Table 3) would only entail reading about half of the text. As for the structure, this is obviously not the typical scientific paper, so the role of the abstract, intro, and conclusions are a bit different, especially given the challenge of keeping this paper from getting any longer. And we are not intending to be prescriptive with respects to solutions and approaches, just

about issues that need to be addressed for meaningful scientific inference about the world around us. We do feel the series of questions in the concluding paragraph of the conclusions summarizes the latter. We are unclear of what specifically the reviewer believes is lacking from the introduction.

**Reviewer Point P 3.4** — For any reader with even a moderate understanding or familiarity of Bayesian statistics, this treatment seems unnecessary. For example, sections 2.1 and 2.2 could be much briefer with the reader referred to an introduction to probability theory undergraduate textbook or online resource where necessary, rather than repeating the material within this manuscript.

**Reply**: Again we refer the reviewer to our Table 3 road-map. Most of our intended audience has essentially no understanding of Bayesian statistics. And for those that do, the road-map clearly identifies what sections can be skipped.

**Reviewer Point P 3.5** — The discussion of MCMC in section 2.7 is not the main purpose of this work, hence it could also be condensed without detriment to this manuscript by referring the reader to other papers, textbooks and resources where desired.

**Reply**: One aim of this submission was to be as self-contained as possible for the reader not interested in implementation, especially those with limited or no statistical background. Forcing such a reader to jump around to other texts will likely mean that most such readers will stop reading. However, we do agree that MCMC is not a required component for the main text (beyond brief mention) and will either relegate it to appendix or drop it.

**Reviewer Point P 3.6** — Throughout the manuscript the authors incorporate numerous examples, both toy models, and from across paleo climate and ice-sheet modelling, but also in relation to present and future modelling efforts. Whilst these provide clarity to the meaning of the statistical framework, in particular, relating this to the analysis of paleo computer models, it also seems like an attempt to conform to the scope of Climate of the Past.

**Reply**: The examples were chosen to impart understanding to the target audience (all interested in the inference of paleo ice and or climate evolution), which is basically much (if not most) of the Climate of the Past readership. A number of them were added based on efforts by LT to make the text more comprehensible to members of LT's research group (of paleo ice and/or climate modellers). They were not chosen to fit into any journal guideline.

**Reviewer Point P 3.7** — In many instances these examples seem unnecessarily long and are often vague in their description without any actual results, for example, lines 399-429, in the discussion of internal model discrepancy.

**Reply**: We do not see what is vague about the indicated lines 399-429. This is fairly precisely what LT has done in his group in ongoing work that will soon be submitted for publication. Again we remind reviewer, that to date, no one in the paleo ice or climate modelling fields has published an internal discrepancy assessment, and LT (in the paleo modelling field for over 27 years) strongly suspects that most paleo modellers would not know how to do this or even what it is.

**Reviewer Point P 3.8** — Greater clarity of exposition could be achieved by taking inspiration from Vernon et al. (2018) and restructuring the manuscript, only using short examples where absolutely

necessary. Firstly, I would suggest combining sections 2 and 3 to describe the framework including: prior and model specification; emulation (see comments in section 2.3); uncertainty quantification; history matching and the need for simulations to bound reality; and uncertainty quantification in making predictions/retrodictions.

**Reply**: We respectfully disagree with this suggestion. Section 2 provides a Bayesian conceptual framework, that the paleo modelling community is far from meaningfully implementing. Section 3 introduces a tractable non-Bayesian framework that can be implemented in current modelling work.

**Reviewer Point P 3.9** — The paper should finish by briefly highlighting what further steps are required by the paleo modelling community, rather than the extensive section 4 which seems to repeat many of the points found in sections 2 and 3.

**Reply**: As mentioned above, parts of section 4 (specifically 4.1 Ensuring uncertainty is addressed in model-based studies and 4.2 Addressing uncertainty in data-based studies) take on some of the more traditional role of the Conclusions section, thus the repetition. None of the subsections in section 4 are long, (with "4.7 a community methodology research 1030 and development agenda" at 1.5 pages being the longest.)

**Reviewer Point P 3.10** — Given their importance within the presented Bayesian framework as a fast statistical approximation to the simulator's output(s) for as yet unevaluated paper settings along with a corresponding statement of the uncertainty, I believe that emulators warrant a more detailed exposition within the main body of paper, rather than leaving it to Appendix A6.

**Reply**: Given that the large majority of the intended audience won't know what a Gaussian process is, we respectfully disagree. Beyond our brief description in the main text, we do not see a useful analogue such as we presented for MCMC that is accessible to the broad target audience. The appendix is meant for those who are specifically interested in implementation, which is a relatively small subset of the target audience.

**Reviewer Point P 3.11** — It is not guaranteed that the reader will know what is an emulator, or more specifically, a Regression Stochastic Process Emulator (RSPE) as it is termed in this manuscript. It is therefore necessary to expand on the discussion currently provided in Appendix A6 with further mathematical details of their formulation, as in Vernon et al. (2010), highlighting the range of possible choices to encapsulate the possible model output behaviour, for example, see Rasmussen et al. (2006) for an in-depth discussion of Gaussian Processes (GPs). Within this manuscript there is a focus on emulating outputs one-by-one and treating them independently which for many applications is a suitable and justifiable approach. I would recommend that the authors also reference that there exists numerous multi-variate emulation techniques such as: separable GP emulation (Conti et al. (2010)); the outer-product emulator which extends the separability assumption onto the regression components (Rougier (2008) and Rougier et al. (2009)); non-separable emulators (Fricker et al. (2013)); parallel partial GPs (Gu et al. (2016)); and through the use of basis representations of multivariate outputs (Higdon et al. (2008) and Salter et al. (2019, 2022)). It is unnecessary to provide further methodological details in this manuscript. A further minor point regarding the approach to emulation described in lines 500-502; it would help to provide the name for this technique as multilevel, multi-scale or multi-delity emulation, in order to aid the reader in identifying other literature or code implementa- tions.

**Reply**: We will add most or all of the suggested references to the revised submission.

**Reviewer Point P 3.12** — Practical implementation of the described framework is important and the authors therefore signpost the reader to the recently released hmer R package. It would also be useful to mention the accompanying website, https://hmer-package.github.io/ website/index.html, which provides detailed tutorials, as well as links to published research using the package. For GP emulation, see the RobustGaSP R package, https: //cran.r-project.org/web/packages/RobustGaSP/index.html, and the associated papers (Gu et al. (2016, 2019)). In addition, many within the paleo modelling community use Python as their main programming language, hence it would be of use to suggest similar Python modules such as GPy for GP emulation, https://sheffieldml. github.io/GPy/, which also includes tutorials.

**Reply**: We thank the reviewer for all the above links and will add them to the text.

**Reviewer Point P 3.13** — It would also add to the content of section 2.4 to comment on biases in observational errors and structural model discrepancy and how these should be accounted for within the error model

**Reply**: Good point, and will do.

**Reviewer Point P 3.14** — A comment should also be included regarding any sources of uncertainty exhibiting a parameter dependency, with this being particularly relevant for (internal) structural model discrepancy. In equation 14, the authors should comment on the implications of whether the model M is deterministic or stochastic. This is not immediately clear when going from the first to the second line of this equation.

**Reply**: We already state "It may well be that the structure of the internal discrepancy is non-Gaussian and/or has significant dependence on ensemble parameters. For both of these cases, a more generalized statistical model is required to represent it (c.f. appendix A6 on emulation)." as well as "Emulators can also emulate models that have stochastic components. When internal discrepancy has spatial, temporal, and/or ensemble parametric dependence, an emulator can therefore offer an efficient representation. Such emulation also may enable the introduction of ensemble parameters (i.e. thereby 1475 subject to history matching and/or posterior inference) to set the structure or amplitude of the stochastic noise used to assess the internal discrepancy."
    We will make clear that model M is deterministic in the revisions.

**Reviewer Point P 3.15** — The authors include the following overly strong statement about structural model discrepancy: Within most (if not all) scientific disciplines, the tendency to date has been to effectively ignore this source of uncertainty" (lines 340-341). Whilst it is true that model discrepancy is often overlooked (or given minimal treatment) within the (paleo) climate and ice-sheet modelling literature, there exist examples across a range of applications where model discrepancy is explicitly assessed, for example: in cosmology (Vernon et al. (2010)); epidemiology (Andrianakis et al. (2015, 2017)); and in climate modelling (Edwards et al. (2019)).

**Reply**: Unless the reviewer can provide two scientific fields/disciplines in the natural sciences for which even a tenth of the modelling papers published in the last 6 years adequately address model discrepancy, we stand by the above statement. Model discrepancy is almost always overlooked in the

paleo modelling community (beyond perhaps throw-away statements such as "our results are subject to model uncertainties"). For the paleo ice sheet modelling context (for which internal discrepancy assessment is much more computationally accessible), only two past studies to date (as listed in Table 1) have even addressed it to a limited extent, none have adequately.

We will see what useful and relevant examples of internal discrepancy assessment can be cited beyond those that already are (external discrepancy already has relevant cited examples) is useful and we will add a couple of citations.

**Reviewer Point P 3.16** — In particular, it is necessary to greatly condense the description of the framework to the core elements, with additional information provided for those interested through supplementary material and via references to other resources that already provide an appropriate coverage of the facet.

**Reply**: As we describe in our general reply to reviewers, we do not see how the content can be further usefully condensed without losing much of the target audience or making the text even more "dense". We have already placed the material of value mostly for those interested in implementation in the appendix.

**Reviewer Point P 3.17** — The exposition would be greatly helped by including a substantive example applying this framework to paleo models, along with appropriate code, which will also motivate and demonstrate to the reader how such an analysis can be practically implemented.

**Reply**: This is a whole other paper in itself. LT is working on a write-up of working that addresses much of this for glacial cycle ice sheet modelling contexts.